# REMAINING-DATA-FREE MACHINE UNLEARNING BY SUPPRESSING SAMPLE CONTRIBUTION

**Xinwen Cheng**[1] **Zhehao Huang**[1] **Wenxing Zhou**[1] **Zhengbao He**[1]
**Ruikai Yang**[1] **Yingwen Wu**[1] **Xiaolin Huang\***[‡,2]
[1]Institute of Image Processing and Pattern Recognition,
School of Automation and Intelligent Sensing, Shanghai Jiao Tong University
[2]Shanghai Key Laboratory of Flexible Medical Robotics
{xinwencheng, kinght_h, apricvivi, lstefanie}@sjtu.edu.cn
{ruikai.yang, yingwenwu, xiaolinhuang}@sjtu.edu.cn

## ABSTRACT

Machine unlearning (MU) aims to remove the influence of specific training samples from a well-trained model, a task of growing importance due to the "right to be forgotten." The unlearned model should approach the retrained model, where forgetting data do not contribute to the training process. Therefore, unlearning should withdraw their contribution from the pre-trained model. However, quantifying and disentangling sample's contribution to overall learning process is highly challenging, leading most existing MU approaches to adopt other heuristic strategies such as random labeling or knowledge distillation. These operations inevitably degrade model utility, requiring additional maintenance with remaining data. To advance MU towards better utility and efficiency for practical deployment, we seek to approximate sample contribution with only the pre-trained model. We theoretically and empirically reveal that sample's contribution during training manifests in the learned model's increased sensitivity to it. In light of this, we propose MU-Mis (Machine Unlearning by Minimizing input sensitivity), which directly suppresses the contribution of forgetting data. This straightforward suppression enables MU-Mis to successfully unlearn without degrading model utility on the remaining data, thereby eliminating the need for access to the remaining data. To the best of our knowledge, this is the first time that a remaining-data-free method can perform on par with top performing remaining-data-dependent methods. The code is released in https://github.com/poppopbean0903/MU-Mis.

## 1 INTRODUCTION

Deep neural networks (DNNs) are revealed to store information of training data (Feldman, 2020; Feldman & Zhang, 2020; Tian et al., 2025) and such information could be reproduced by privacy attacks (Shokri et al., 2017; Zhu et al., 2019), raising data privacy concerns. The "right to be forgotten" (Regulation, 2018) is introduced to safeguard user privacy, which entails ensuring that the DNN performs as if the data were never involved in the training.

While retraining from scratch would ideally achieve this, it is often infeasible due to the high cost of training DNNs. This has motivated the study of *"Machine Unlearning"* (MU) (Cao & Yang, 2015), which fine-tunes the *pre-trained model* to approach the *retrained model* as closely as possible. The essential distinction in pre-trained and retrained model lies in the contribution of forgetting data, whose role shift from "contributors" that affect parameter updates in the pre-trained model to "bystanders" that exert no influence in the retrained model. Therefore, unlearning should aim to withdraw their contribution to the learning process.

However, identifying such a contribution is highly challenging. Learning is a dynamic process that gradually remembers and assimilates data, while unlearning, which is the reverse process that gradually removes data information, is achieved by backtracking the training trajectory to withdraw

---
*Corresponding author

historical gradients in early study (Graves et al., 2021; Thudi et al., 2022). Nevertheless, such tracking not only contradicts the efficiency demands of unlearning but also yields limited effectiveness due to the incrementality of training (Wang et al., 2024b).

Consequently, most existing MU methods circumvent the difficulty of estimating sample contribution through other heuristics. A common strategy is to introduce confusion, *e.g.*, random relabeling (Golatkar et al., 2020; Graves et al., 2021; Fan et al., 2024b) or knowledge distillation from useless teacher (Chundawat et al., 2023; Kurmanji et al., 2023). However, these approaches suffer from several limitations: *(i)* such confusion causes *catastrophe unlearning* (Wang et al., 2024b) or *over-forgetting* (He et al., 2025), *i.e.*, severe degradation of model utility on the remaining data; *(ii)* the degradation in turn necessitates costly maintenance using the remaining data, thereby substantially *undermining MU efficiency*; *(iii)* the remaining data are not always accessible in practice. These limitations collectively underscore the importance of moving beyond heuristic confusion strategies and developing more principled unlearning mechanisms to advance MU toward higher utility and efficiency. Therefore, although quantifying sample contribution is inherently challenging, in this paper, we make efforts to ground unlearning in a precise characterization of sample's contribution.

Instead of accumulating the historical contributed gradient update during training, we identify the clue of contribution directly from the derivative of the training algorithm w.r.t a training sample. The learning process is a mapping by the training algorithm $\mathcal{A}$ from the training set $\mathcal{D} = \{(\boldsymbol{x}_i, \boldsymbol{y}_i)\}$ to a learned function $f$: denoted as $f = \mathcal{A}(\mathcal{D})$. Therefore, the training sample $\boldsymbol{x}_i$ contributes to the output: $\partial \mathcal{A} / \partial \boldsymbol{x}_i \neq 0$ while a sample out of the training set does not. A simple yet enlightening example lies in the support vector machine (Cortes & Vapnik, 1995; Christmann & Steinwart, 2008), where only the training data can act as support vectors that impact the decision boundary. Thus, withdrawing the sample contribution can be achieved by suppressing $\partial \mathcal{A} / \partial \boldsymbol{x}_i$.

The main challenge is that $\mathcal{A}$ corresponds to a dynamic training process without a closed-form expression. To address this, we theoretically illustrate that $\partial \mathcal{A} / \partial \boldsymbol{x}_i$ could be approximated by the learned model's sensitivity to its input $\boldsymbol{x}$, i.e. $\partial f(\boldsymbol{x}) / \partial \boldsymbol{x}$ with $f = \mathcal{A}(\mathcal{D})$ in *Section 3.2*. To derive a principled and optimization-friendly guideline aligned with the behavior of a retrained model, we delve deeper into the input sensitivity across different logits. Our empirical investigations under the machine learning (*Section 3.3*) and machine unlearning (*Section 3.4*) scenarios reveal that a sample's contribution manifests as disproportionately higher input sensitivity of the target logit relative to irrelevant logits. In light of this finding, we propose **MU-Mis** (Machine Unlearning by Minimizing Input Sensitivity), which suppresses sample contribution by reducing the sensitivity disparity between the target and non-target logits to the forgetting data.

We evaluate MU-Mis on 3 standard unlearning tasks across 6 datasets, benchmarking against 6 competitive remaining-data-dependent unlearning methods and 4 existing remaining-data-free baselines. The results demonstrate that MU-Mis achieves effective unlearning while preserving model utility on the remaining data **without utilizing them**, performing on par with SoTA remaining-data-dependent approaches and outperforming all remaining-data-free methods significantly, with the added advantage of notable computational efficiency. Moreover, due to its principled forgetting mechanism, MU-Mis exhibits stable and effective behavior in sequential unlearning, whereas existing methods are disclosed to exhibit several deficiencies. Collectively, these results underscore the practicality and reliability of MU-Mis for real-world deployment.

Our key contributions can be summarized as follows:

❶ We theoretically and empirically reveal that a sample's contribution is reflected in the amplified sensitivity gap between the target logit and irrelevant logits, enabling the identification of sample contribution with only the pre-trained model.

❷ Based on the above analysis and findings, we propose MU-Mis, which suppresses the sample's contribution by minimizing the sensitivity magnitude gap for the forgetting data.

❸ Comprehensive experiments demonstrate the effectiveness and efficiency of MU-Mis. To our best knowledge, it is the first time that a remaining-data-free method can perform on par with top performing remaining-data-dependent methods.

## 2 RELATED WORK

Machine unlearning (MU) (Xu et al., 2024; Bourtoule et al., 2021) aims to remove the influence of specific training data from a pre-trained model to mitigate privacy risks. MU is commonly divided into *exact* and *approximate* settings (Shaik et al., 2023). Exact MU approaches the *parameters* of the retrained model and provides statistical guarantees of privacy risk(Guo et al., 2020; Suriyakumar & Wilson, 2022; Neel et al., 2021; Giordano et al., 2019), whereas approximate MU targets the *output distribution* of the retrained model. In this paper, we concentrate on approximate unlearning, as it is more practical in large-scale models and situations with limited time and resources.

**MU via gradient-based updates.** One straightforward way to retrieve sample contribution is to keep and utilize the historical information (*e.g.*, checkpoints and gradients) during training. Amnesiac (Graves et al., 2021) withdraw gradient updates of related batches, and DeltaGrad (Wu et al., 2020) utilize intermediate checkpoints and quasi-newton method for rapid retraining. Memory concerns arising from storing historical information motivate estimating sample contribution from learned model via influence function (Guo et al., 2020). However, inverse Hessian is computationally prohibitive for DNNs, prompting various approximation methods tailored for unlearning (Peste et al., 2021; Mehta et al., 2022; Meng et al., 2022). However, influence function is revealed to be fragile in DNNs (Basu et al., 2020; Bae et al., 2022; Hammoudeh & Lowd, 2024) due to its reliance on convexity and optimality assumptions, accounting for their failure to preserve utility on remaining data (Wu et al., 2022). Regrettably, all the existing DNN data-influence estimators (Pruthi et al., 2020; Chen et al., 2021) require retracing training trajectories, therefore cannot be readily optimized and applied to MU. In this paper, we shift sample contribution from parameter space to function space, *i.e.*, $\Delta w$ to $\partial A/\partial x$, and demonstrate that sample's contribution would reflect in pre-trained model's sensitivity to itself, opening up a new perspective for measuring sample contribution in DNNs.

**MU via loss guided re-optimization.** Above gradient-based unlearning methods suffer from practical limitations for DNNs. Generally, practical MU unlearn by fine-tuning the model to optimize a proposed loss. The loss typically follows two design ideas: *(i)* make model's behavior on the forgetting data similar to that on unseen data through knowledge distillation (Chundawat et al., 2023; Lin et al., 2023; Kurmanji et al., 2023) or label confusion (Graves et al., 2021; Fan et al., 2024b), and *(ii)* suppress parameters that are responsible for the forgetting set (Liu et al., 2024; Foster et al., 2024a; Fan et al., 2024b). However, for lack of identifying "what to unlearn", these methods unlearn either in an "impair-then-repair" regime (Tarun et al., 2023) or introduce tailored mechanisms (Hoang et al., 2024; Fan et al., 2024b) to mitigate collateral damage. In contrast, we pursue a more principled forgetting operation by explicitly identifying sample contributions, eliminating ad-hoc compensation.

**Remaining-data-free MU.** Developing remaining-data-free methods aligns more closely with the essence and practical demands of MU, given the limited accessibility of retained data and MU efficiency in practice. JiT (Foster et al., 2024a) smooths model outputs around forgetting data by minimizing local Lipschitz value, while SCAR (Bonato et al., 2024) distills from the pre-trained model and utilizes Out-of-distribution (OOD) data to preserve utility. However, both methods have a clear performance gap to SoTA remaining-data-dependent methods, and SCAR still relies on additional OOD data. We achieve a more nuanced removal by identifying sample contribution.

## 3 MACHINE LEARNING, MACHINE UNLEARNING AND INPUT SENSITIVITY

### 3.1 PROBLEM FORMULATION

**Machine Learning (ML)** is to learn a mapping from the input space $\mathcal{X}$ to the output space $\mathcal{Y}$, denoted as the function $f(\cdot) : \mathcal{X} \to \mathcal{Y}$. As we mainly focus on classification models, the output of $f$ is C-dimensional in a C-category classification model. Learning is performed by a *training algorithm* $\mathcal{A}$, which generally takes in a training dataset $\mathcal{D}$ and returns the *learned function* $f$, *i.e.*, the outcome of $\mathcal{A}$ varies with different training datasets. To investigate sample-wise influence on the learning process, we consider $\mathcal{A}$ in a broader sense and distinguish different training processes by the *training dataset* $\mathcal{D} = \{(\boldsymbol{x}_i, \boldsymbol{y}_i)\}_{i=1}^m$. That is to say, we have a family of the training algorithm $\mathcal{A}_{\mathcal{D}}$ and each one is a multivariate function that takes all the samples $\{\boldsymbol{x}_j \in \mathcal{X}\}$ as input, regardless of whether they are in the training dataset $\mathcal{D}$. Therefore, the output of $\mathcal{A}_{\mathcal{D}}$ does not vary with each input variable, but only varies with the change of the training data $\boldsymbol{x}_i \in \mathcal{D}$, and makes no response to the change of samples out of the training set.

**Machine Unlearning (MU)** is to remove the influence of *forgetting data* $\mathcal{D}_f \subset \mathcal{D}$ from the *pre-trained model* $w_p$, while preserving model utility on the *remaining data* $\mathcal{D}_r = \mathcal{D}\backslash\mathcal{D}_f$. The learned function $f$ is parameterized by parameters $w \in \mathbb{R}^d$ with input variable $\boldsymbol{x}$, *i.e.*, instantiated as $f(\boldsymbol{x}; w)$. A good approximate unlearning mechanism should efficiently and effectively transform $w_p$ into a *sanitized model* $w_u$, such that the output distribution of $w_u$ closely matches *retrained model* $w_r$.

**Remark on notation.** To facilitate the understanding of the objectives in our analysis, we only bold the input variables of the training algorithm $\mathcal{A}_\mathcal{D}$ and learned function $f(\boldsymbol{x})$, which are respectively $\boldsymbol{x_i}$ and $\boldsymbol{x}$ in the following analysis.

## 3.2 THEORETICAL ANALYSIS CONNECTING SAMPLE CONTRIBUTION AND INPUT SENSITIVITY

As previously discussed, machine unlearning is to withdraw sample's contribution to the learning process, and an efficient unlearning method should explore the contribution directly from the pre-trained model. To detach per-sample contribution with the pre-trained model, we propose to identify the clue of contribution from the derivative of training mapping $\mathcal{A}_\mathcal{D}$ to training sample $\boldsymbol{x_i}$, i.e. $\partial\mathcal{A}_\mathcal{D}/\partial\boldsymbol{x_i}$. Recall that $\mathcal{A}_\mathcal{D}$ is determined by the training dataset $\mathcal{D} = \{(\boldsymbol{x_i}, \boldsymbol{y_i})\}_{i=1}^m$ and outputs the learned function $f(\boldsymbol{x}; w_p)$. Then $\partial\mathcal{A}_\mathcal{D}/\partial\boldsymbol{x_i}$ is to compute $\partial f(\boldsymbol{x}; w_p)/\partial\boldsymbol{x_i}$. However, there is no explicit expression for this derivative. Therefore, in this part, we reflect on the learning dynamics to seek a surrogate with the pre-trained model. Figure 1 provides an overview of the key objectives investigated in our following analysis.

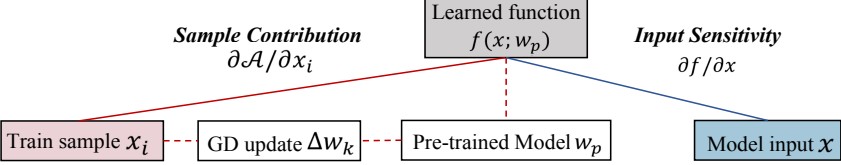

Figure 1: **A brief overview of the theoretical connection between sample's contribution and a pre-trained model's input sensitivity.** The dashed lines illustrate how the influence of a training sample propagates through gradient updates to the pre-trained model. When a sample participates in training, the gradient it contributes induces an update of the model in function space, which inherently increases the learned function's sensitivity to that sample's input.

**Gradient Descent (GD).** After $T$ iterations training updates in the parameter space, we have pre-trained model parameter $w_p = w_0 + \sum_{k=1}^T \Delta w_k$, where $w_0$ is randomly initialized model parameters and $\Delta w_k = w_{k+1} - w_k$ is the $k^{\text{th}}$ parameter update. Specifically, when training loss $\mathcal{L}$ and gradient descent with step size $\eta$ are used, we have

$$\Delta w_k = -\eta \sum_{i=1}^m \frac{\partial\mathcal{L}(\boldsymbol{x_i})}{\partial w}\Big|_{w=w_k} = -\eta \sum_{i=1}^m \frac{\partial f(\boldsymbol{x_i}; w)}{\partial w}\Big|_{w=w_k} \frac{\partial\mathcal{L}(\boldsymbol{x_i})}{\partial f}.$$

**Function space update induced by GD.** Viewing machine learning from the function space with first-order Taylor expansion on parameters, correspondingly we have $f = f_0 + \sum_{k=1}^T \Delta f_k$, where $f_0 = f(\boldsymbol{x}, w_0)$ is initial function and $\Delta f_k$ is induced by parameter update $\Delta w_k$. The evolution in function induced by parameter update is:

$$\Delta f_k(\boldsymbol{x}; w) \approx \frac{\partial f(\boldsymbol{x}; w)}{\partial w}\Big|_{w=w_k}^\top \Delta w_k = -\eta \sum_{i=1}^m \frac{\partial f(\boldsymbol{x}; w)}{\partial w}\Big|_{w=w_k}^\top \frac{\partial f(\boldsymbol{x_i}; w)}{\partial w}\Big|_{w=w_k} \frac{\partial\mathcal{L}(\boldsymbol{x_i})}{\partial f}.$$

**Learned function.** To better explain the idea, we make simplifications: *(i)* Note that $\frac{\partial f(\cdot; w)}{\partial w}\big|_{w=w_k}$ is the mapping from model input $\boldsymbol{x}$ to the induced backpropagation gradient with parameters $w_k$. We abbreviate this mapping as $g_k(\boldsymbol{x}) : \mathcal{X} \to \mathbb{R}^{d\times C}$ and its derivative to input $\boldsymbol{x}$ as $g'_k$, where $d$ is total number of model parameters. *(ii)* In classification problem with cross-entropy loss as $\mathcal{L}$, we have $\frac{\partial\mathcal{L}(\boldsymbol{x_i})}{\partial f} = e_c - p(\boldsymbol{x_i})$, where $e_c$ is a one-hot vector with only $c^{\text{th}}$ element equals to 1, and $p$ is the probability vector of $\boldsymbol{x_i}$. The final learned function $f$ is

$$f(\boldsymbol{x}; w_p) = f(\boldsymbol{x}; w_0) + \sum_{k=1}^T \Delta f_k(\boldsymbol{x}, w) = f(\boldsymbol{x}; w_0) - \eta \sum_{k=1}^T \underbrace{g_k^\top(\boldsymbol{x})}_{(1)} \sum_{i=1}^m \underbrace{g_k(\boldsymbol{x_i})(e_c - p(\boldsymbol{x_i}))}_{(2)}.$$

Notice that term (1) is related to the **forward inference process** while term (2) is related to the **machine learning process**. Derivative of $f$ w.r.t $\boldsymbol{x}$ indicates how the prediction of $f$ varies with its input $\boldsymbol{x}$ at inference time, while derivative w.r.t $\boldsymbol{x_i}$ indicates how the learned function $f$ varies when the training sample $\boldsymbol{x_i}$ varies. The former implies **the learned model's sensitivity to its input**, and the latter is **the training sample's influence on learning**. Next, we take the derivative of $f$ w.r.t $\boldsymbol{x}$ and $\boldsymbol{x_i}$ respectively to view their relationship. Note that $p$ is a probability vector determined by $\boldsymbol{x_i}$. Due to softmax activation, we consider $p(\boldsymbol{x_i})$ hardly changes around $\boldsymbol{x_i}$, and omit its derivative term w.r.t $\boldsymbol{x_i}$. The difference in mapping $g_k$ when $\boldsymbol{x_i}$ changes is also omitted.

$$
\begin{cases}
\dfrac{\partial f(\boldsymbol{x}; w_p)}{\partial \boldsymbol{x_i}} = -\eta \sum_{k=1}^{T} \underbrace{g_k'(\boldsymbol{x_i}) g_k(\boldsymbol{x})(e_c - p(\boldsymbol{x_i}))}_{=:\mathcal{C}_k(\boldsymbol{x}, \boldsymbol{x_i})}, \\
\dfrac{\partial f(\boldsymbol{x}; w_p)}{\partial \boldsymbol{x}} = \dfrac{\partial f(\boldsymbol{x}; w_0)}{\partial \boldsymbol{x}} - \eta \sum_{k=1}^{T} \sum_{i=1}^{m} \underbrace{g_k'(\boldsymbol{x}) g_k(\boldsymbol{x_i})(e_c - p(\boldsymbol{x_i}))}_{=:\mathcal{S}_k(\boldsymbol{x}, \boldsymbol{x_i})}.
\end{cases}
\tag{1}
$$

**Input sensitivity of learned function reflects sample contribution.** $\mathcal{C}_k(\boldsymbol{x}, \boldsymbol{x_i})$ determines the prediction change on $\boldsymbol{x}$ when $\boldsymbol{x_i}$ changes, and $\mathcal{S}_k(\boldsymbol{x}, \boldsymbol{x_i})$ stands for the part of model's sensitivity to $\boldsymbol{x}$ contributed by training sample $\boldsymbol{x_i}$. Note that $\frac{\partial f(\boldsymbol{x_i}; w_p)}{\partial \boldsymbol{x_i}}$ is similar to the definition of memorization, which is framed as *self-influence* (Feldman, 2020; Feldman & Zhang, 2020). To be more specific, memorization of a sample is defined as the prediction difference in itself when training with or without it. Similarly, the self-influence here is the prediction difference on $\boldsymbol{x_i}$ when it slightly changes, i.e. $\frac{\partial f(\boldsymbol{x_i}; w_p)}{\partial \boldsymbol{x_i}}$. Thus we consider $\frac{\partial f(\boldsymbol{x_i}; w_p)}{\partial \boldsymbol{x_i}}$ as the reflection of sample $\boldsymbol{x_i}$'s contribution. From the formulation, we have $\mathcal{S}_k(\boldsymbol{x_i}, \boldsymbol{x_i}) = \mathcal{C}_k(\boldsymbol{x_i}, \boldsymbol{x_i})$. For a specific training sample $\hat{x} \in \mathcal{D}$, the learned model's sensitivity to it can be further decomposed as

$$
\begin{aligned}
\frac{\partial f(\boldsymbol{x}; w_p)}{\partial \boldsymbol{x}}\Big|_{\boldsymbol{x}=\hat{x}} &= \frac{\partial f(\boldsymbol{x}; w_0)}{\partial \boldsymbol{x}}\Big|_{\boldsymbol{x}=\hat{x}} - \eta \sum_{k=1}^{T} \sum_{i=1}^{m} \mathcal{S}_k(\hat{x}, \boldsymbol{x_i}) \\
&= \frac{\partial f(\boldsymbol{x}, w_0)}{\partial \boldsymbol{x}}\Big|_{\boldsymbol{x}=\hat{x}} - \eta \sum_{k=1}^{T} \left[ \mathcal{S}_k(\hat{x}, \hat{x}) + \sum_{\tilde{x} \in \mathcal{D}/\hat{x}} \mathcal{S}_k(\hat{x}, \tilde{x}) \right] \\
&= \underbrace{-\eta \sum_{k=1}^{T} \mathcal{S}_k(\hat{x}, \hat{x})}_{\text{Contribution Term}} + \underbrace{\frac{\partial f(\boldsymbol{x}, w_0)}{\partial \boldsymbol{x}}\Big|_{x=\hat{x}} - \eta \sum_{k=1}^{T} \sum_{\tilde{x} \in \mathcal{D}/\hat{x}} \mathcal{S}_k(\hat{x}, \tilde{x})}_{\text{Residual Term}}.
\end{aligned}
\tag{2}
$$

The randomly initialized function $f_0$ is generally quite insensitive to input change. Thus, the first term of the above residual term is very small. The second term is related to the correlation between the gradient on $\tilde{x}$ and the sensitivity of the gradient on $\hat{x}$. We use a simple MLP model to illustrate the insight of $\mathcal{S}_k(\hat{x}, \tilde{x}) << \mathcal{S}_k(\hat{x}, \hat{x})$ with $\hat{x} \neq \tilde{x}$ in Appendix D. Therefore, the residual term is relatively smaller than the contribution term. In summary, the contribution of a training sample to the training process would be approximately reflected in the pre-trained model's output sensitivity to the sample.

**Empirical validation.** We validate the contribution to learning by comparing $\|\nabla_{\boldsymbol{x}} f\|_F$ of the training data before and after training in Figure 2. In a randomly initialized model, there is little response to input changes, only about $10^{-4}$. After training, there is a significant order of magnitude growth to $10^3$, indicating an increased attention of the trained model to the training data's variations. This implies that the training data contribute to model performance, and such efforts include promoting the model's sensitivity to them during training.

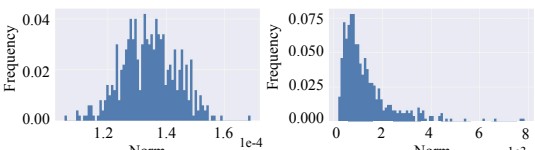

Figure 2: **Input sensitivity $\|\nabla_{\mathbf{x}} f\|_F$ of training data before and after training.** Left: In randomly initialized model $w_0$. Right: In well-trained model $w_p$. After training, the model exhibits significantly increased sensitivity to the training data, reflecting their contribution during training.

### 3.3 INPUT SENSITIVITY OF THE TARGET AND IRRELEVANT CLASS LOGIT

During training, model predictions on samples are driven toward their correct labels, so sample contributions might differ across logits. To further refine our view of sample contribution, we examine individual logits of $f(\boldsymbol{x}) \in \mathbb{R}^C$ in the following part.

Let $f_c$ denotes the logit output of the target class and $f_{c'}$ denotes the logit output of irrelevant classes. Figure 3 compares distributions between $\|\nabla_{\boldsymbol{x}} f_c\|_F$ and $\frac{1}{C-1}\sum_{c'\neq c}\|\nabla_{\boldsymbol{x}} f_{c'}\|_F$ (denoted as $\|\nabla_{\boldsymbol{x}} f_{c'}\|_F$ for brevity in the following) of training data before and after training. In the **randomly initialized** model, these two quantities are of **comparative magnitude**, but $\|\nabla_{\boldsymbol{x}} f_c\|_F$ becomes **much larger** than $\|\nabla_{\boldsymbol{x}} f_{c'}\|_F$ **after training**. This observation implies that samples contribute to amplifying $\|\nabla_{\boldsymbol{x}} f_c\|_F$ to surpass $\|\nabla_{\boldsymbol{x}} f_{c'}\|_F$ during training, generating a discernible difference in

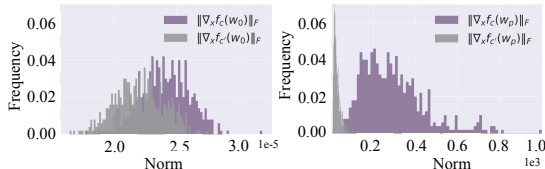

Figure 3: **Input sensitivity** $\|\nabla_{\mathbf{x}} f_c\|_F$ **and** $\|\nabla_{\mathbf{x}} f_{c'}\|_F$ **before and after training.** Left: randomly initialized model $w_0$. Right: well-trained model $w_p$. After training, the gap between target and irrelevant class sensitivities enlarges, providing a clearer signal of the sample's contribution.

whether a sample has been learned. A complementary explanation of this finding comes from the generative view of discriminative models: the softmax-based discriminative classifier is revealed to be implicitly a density model which learns data distribution (Grathwohl et al.; Srinivas & Fleuret, 2021). From this viewpoint, the logits $f(\boldsymbol{x})$ of standard classifiers are un-normalized log-densities, and corresponding input-gradients $\nabla_{\boldsymbol{x}} f_i(\boldsymbol{x})$ are log-gradients of a class-conditional density model. In other words, we have $\nabla_{\boldsymbol{x}} \log p_\theta(\boldsymbol{x}|y=i) = \nabla_{\boldsymbol{x}} f_i(\boldsymbol{x})$ in the classification model, providing a rationale for the observed discrepancy.

## 3.4 INPUT SENSITIVITY OF SAMPLES PRESENT AND ABSENT IN TRAINING

For effective unlearning, the optimization objective should accurately steer the pre-trained model toward the retrained model. To validate that the theoretically grounded sensitivity gap provides a reliable measure of sample contribution to guide unlearning, we empirically examine the input sensitivity of forgetting data under MU scenarios (introduced in Section 5.1 and Appendix F.1).

For each forgetting sample, we compute the difference $\Delta$ between the retrained and the pre-trained model's sensitivity to it, where the sensitivity including $\|\nabla_{\boldsymbol{x}} f_c\|_F, \|\nabla_{\boldsymbol{x}} f_{c'}\|_F$ and $\|\nabla_{\boldsymbol{x}} f_c\|_F - \|\nabla_{\boldsymbol{x}} f_{c'}\|_F$. Aiming for a light-weight unlearning algorithm, we prefer an optimization direction rather than modeling a distribution or specifying a target value for each sample. Hence, we focus on the *sign* of $\Delta$ and count the ratio of rise and fall of $\Delta$ to examine the overall trend in Figure 4.

From left to right in Figure 4, $\Delta$ is the sample-wise difference between the retrained and pre-trained model on $\|\nabla_{\boldsymbol{x}} f_c\|_F, \|\nabla_{\boldsymbol{x}} f_{c'}\|_F$ and $\|\nabla_{\boldsymbol{x}} f_c\|_F - \|\nabla_{\boldsymbol{x}} f_{c'}\|_F$. For each quantity, there is a consistent trend across different unlearning settings. Generally, $f_c$ of the retrained model exhibits lower sensitivity and $f_{c'}$ exhibits higher sensitivity to the forgetting data than the pre-trained model. And their sensitivity magnitude gap is consistently smaller in the retrained model across different settings. Therefore, the sensitivity magnitude gap faithfully reflects the behavior of the retrained model and thus serves as a reliable objective to guide unlearning.

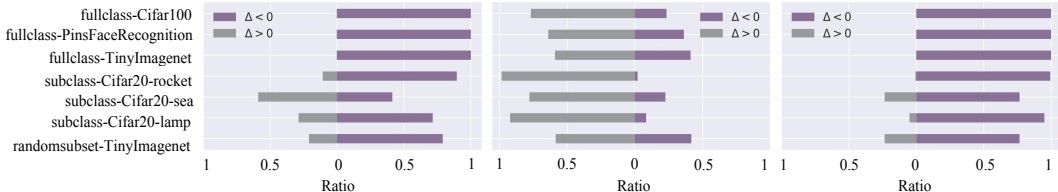

Figure 4: **Ratio of input sensitivity difference $\Delta$ rise and fall of the forgetting data under different unlearning settings.** From left to right, $\Delta$ is the sample-wise difference between the retrained and pre-trained model on $\|\nabla_{\boldsymbol{x}} f_c\|_F, \|\nabla_{\boldsymbol{x}} f_{c'}\|_F$ and $\|\nabla_{\boldsymbol{x}} f_c\|_F - \|\nabla_{\boldsymbol{x}} f_{c'}\|_F$. Sample's contribution to input sensitivity includes promoting $\|\nabla_{\boldsymbol{x}} f_c\|_F$ and suppressing $\|\nabla_{\boldsymbol{x}} f_{c'}\|_F$, thereby enlarging the magnitude gap $\|\nabla_{\boldsymbol{x}} f_c\|_F - \|\nabla_{\boldsymbol{x}} f_{c'}\|_F$.

## 4 PROPOSED METHOD

### 4.1 MU-MIS: MACHINE UNLEARNING BY MINIMIZING INPUT SENSITIVITY

In the above section, we theoretically and empirically derived an optimizable and lightweight approximation of sample contributions from the perspective of input sensitivity, showing that they manifest as disproportionately higher sensitivity of the target logit relative to irrelevant logits.

In light of this finding, we propose to withdraw the sample's contribution by reducing such enhancement on the sensitivity magnitude gap. Minimizing this loss guides the pre-trained model to roll back $\|\nabla_x f_c\|_F$ and pick up $\|\nabla_x f_{c'}\|_F$. Mathematically, our proposed unlearning loss is:

$$\mathcal{L}(\mathcal{D}_f; w) = \frac{1}{N_f} \sum_{x_f \in \mathcal{D}_f} (\|\nabla_{\boldsymbol{x}} f_c(x_f, w)\|_F^2 - \|\nabla_{\boldsymbol{x}} f_{c'}(x_f, w)\|_F^2) \quad (3)$$

where $N_f$ is number of the forgetting data, $c$ represents the target class of sample $x$ and $c' \neq c$ denotes an irrelevant class. For each forgetting sample, a new $c'$ is randomly selected every time the loss is computed.

**Algorithm 1** MU-Mis: Machine Unlearning by Minimizing Input Sensitivity

---
**Input:** Forgetting data $\mathcal{D}_f$; Pre-trained model weights $w_p$; Learning rate $\eta$; Stopping threshold ratio $\delta$.
   *# Initialization*
1: $w_0 \leftarrow w_p, \epsilon \leftarrow \infty$
   *# Iterative optimization*
2: **repeat**
3:    **for** each forgetting sample $x \in \mathcal{D}_f$ **do**
4:       Randomly select an irrelevant class $c' \neq c$
5:    **end for**
6:    Compute loss $\mathcal{L}$ according to equation 3
7:    Update $w_{t+1} \leftarrow w_t - \eta \nabla \mathcal{L}$
8:    Update $\epsilon \leftarrow \min(\epsilon, \|\nabla_{\mathbf{x}} f_{c'}(x, w_t)\|_F)$
9: **until** $\|\nabla_{\mathbf{x}} f_{c'}(x, w_t)\|_F > \epsilon$ **and** $\frac{\|\nabla_{\mathbf{x}} f_{c'}(x, w_t)\|_F}{\|\nabla_{\mathbf{x}} f_{c'}(x, w_0)\|_F} > \delta$
**Output:** Updated model weights $w_t$

---

**Stopping Guideline.** To ensure a practical deployment of MU-Mis, we design a stopping rule for terminating optimization once the withdrawal is completed. Empirical analysis in Appendix E reveals a consistent trend of metrics during our optimization: as the MU-Mis loss decreases, forgetting accuracy (FA) drops steadily, while the accuracies on retained (RA) and test data (TA) initially decline slightly and then grow with the recovery of irrelevant-class logit sensitivity. Crucially, RA approaches the retrained model when this sensitivity returns to its initial level. Therefore, we introduce a threshold ratio $\delta$ to govern the termination of unlearning. This criterion ensures that optimization halts when irrelevant-class sensitivity is sufficiently restored. The overall algorithm is outlined in Algorithm 1.

## 5 EXPERIMENTS

### 5.1 EXPERIMENT SETUPS

**Tasks, Datasets and Models.** We evaluate unlearning across 3 settings: full-class (CIFAR-100 (Krizhevsky et al., 2009), PinsFaceRecognition (Burak, 2020), and Tiny ImageNet (Le & Yang, 2015)), sub-class (CIFAR-20 (Krizhevsky et al., 2009)), and random-subset (CIFAR-10 (Krizhevsky et al., 2009) and SVHN (Netzer et al., 2011)). ResNet-18 (He et al., 2016) is adopted as the default backbone, and we additionally evaluate under ViT (Dosovitskiy et al., 2021) to highlight the efficiency of remaining-data-free methods. Beyond unlearning utility, we assess the resilience of unlearning methods by executing multiple full-class and sub-class unlearning requests iteratively.

**Evaluation Metrics.** MU methods should be assessed from three aspects (Xu et al., 2024): *utility*, *privacy*, and *efficiency*. For *utility*, we compute forgetting data accuracy (**FA**), remaining data accuracy (**RA**), and test data accuracy (**TA**) of the unlearned model. The average gap (**Avg. Gap**) between the retrained model and the unlearned model across above 3 accuracy-related metrics are computed to illustrate the utility disparity. We compute the train (**FGTA**) and valid (**FGVA**) accuracy on the forgotten classes in sequential unlearning. Regarding the *privacy* guarantee, we use 2 complementary membership inference attack (**MIA**) methods, MIA-Entropy (Chundawat et al., 2023) and MIA-SCRUB (Kurmanji et al., 2023) to probe the remaining information of the forgetting data. For *efficiency*, we provide the run time efficiency (**RTE**) in **seconds** to indicate timeliness.

**Baselines.** We compare against 8 remaining-data methods: Bad Teacher(BT) (Chundawat et al., 2023), Fine-tune(FT) (Warnecke et al., 2023), SCRUB (Kurmanji et al., 2023), SSD (Foster et al., 2024b), DUCK (Cotogni et al., 2023), SalUn (Fan et al., 2024b), MUNBa (Wu & Harandi, 2025) and LoTus (Spartalis et al., 2025), as well as 4 remaining-data-free methods: RL (Golatkar et al., 2020),NG (Thudi et al., 2022), JiT (Foster et al., 2024a) , SCAR (Bonato et al., 2024). Notably, unlike SCAR, our method requires no auxiliary OOD data. Further details on sequential unlearning settings, metrics, and baselines are provided in Appendix F.1.

## 5.2 Unlearning Utility

Table 1: Performance overview for **full class** unlearning task evaluated on CIFAR-100 and Tiny ImageNet using ResNet-18. This table includes performances of our proposed MU-Mis, 6 remaining-data-dependent and 4 remaining-data-free methods, which are delineated by a horizontal line. The result format is given by $a_{\pm b}$ with mean $a$ and standard deviation $b$ over 5 independent trials. The metric *average gap (Avg. Gap)* is calculated by the average of the performance gaps measured in accuracy-related metrics, including FA, RA and TA. RTE is reported in **seconds**. Values in terms of accuracy-related metrics deviating by more than $5\%$ from the retrain model are highlighted in red.

| Method | CIFAR-100 | | | | | | Tiny ImageNet | | | | | |
|---|---|---|---|---|---|---|---|---|---|---|---|---|
| | RA | FA | TA | Avg. Gap↓ | MIA | RTE | RA | FA | TA | Avg. Gap↓ | MIA | RTE |
| Pretrain | 76.41 | 79.69 | 76.47 | 26.84 | 95.80 | 10880 | 65.85 | 62.00 | 65.50 | 21.03 | 93.59 | 13600 |
| Retrain | $76.52_{\pm0.27}$ | $0.00_{\pm0.00}$ | $75.76_{\pm0.24}$ | 0.00 | $2.87_{\pm0.46}$ | 7432 | $65.36_{\pm0.03}$ | $0.00_{\pm0.03}$ | $64.90_{\pm0.03}$ | 0.00 | $4.80_{\pm0.04}$ | 10367 |
| BT | $76.67_{\pm0.03}$ | $0.00_{\pm0.00}$ | $76.02_{\pm0.03}$ | 0.14 | $0.00_{\pm0.00}$ | 32 | $64.90_{\pm0.01}$ | $0.00_{\pm0.00}$ | $64.53_{\pm0.01}$ | 0.28 | $0.00_{\pm0.00}$ | 240 |
| FT | $76.67_{\pm0.21}$ | $0.28_{\pm0.62}$ | $75.88_{\pm0.22}$ | 0.19 | $0.28_{\pm0.00}$ | 250 | $64.16_{\pm0.26}$ | $0.00_{\pm0.00}$ | $63.87_{\pm0.22}$ | 0.74 | $4.40_{\pm0.58}$ | 262 |
| SCRUB | $76.81_{\pm0.04}$ | $0.00_{\pm0.00}$ | $76.02_{\pm0.04}$ | 0.18 | $5.57_{\pm0.34}$ | 124 | $65.06_{\pm0.04}$ | $0.00_{\pm0.00}$ | $64.69_{\pm0.03}$ | 0.17 | $14.60_{\pm0.52}$ | 860 |
| SSD | $76.27_{\pm0.00}$ | $0.00_{\pm0.00}$ | $75.49_{\pm0.00}$ | 0.17 | $0.00_{\pm0.00}$ | 26 | $65.58_{\pm0.00}$ | $0.00_{\pm0.00}$ | $65.19_{\pm0.00}$ | 0.17 | $0.00_{\pm0.00}$ | 59 |
| DUCK | $75.82_{\pm0.18}$ | $0.20_{\pm0.45}$ | $75.13_{\pm0.17}$ | 0.51 | $0.00_{\pm0.00}$ | 100 | $64.97_{\pm0.14}$ | $0.00_{\pm0.00}$ | $64.61_{\pm0.14}$ | 0.23 | $2.60_{\pm0.46}$ | 55 |
| SalUn | $76.63_{\pm0.03}$ | $1.20_{\pm0.45}$ | $75.85_{\pm0.03}$ | 0.47 | $0.00_{\pm0.00}$ | 254 | $65.21_{\pm0.10}$ | $0.00_{\pm0.00}$ | $64.88_{\pm0.10}$ | 0.06 | $4.40_{\pm0.40}$ | 2630 |
| MUNBa | $74.09_{\pm0.11}$ | $0.00_{\pm0.00}$ | $73.40_{\pm0.12}$ | 1.60 | $9.30_{\pm0.18}$ | 217 | $64.22_{\pm0.14}$ | $0.00_{\pm0.00}$ | $63.88_{\pm0.15}$ | 0.72 | $7.80_{\pm0.16}$ | 897 |
| LoTus | $76.48_{\pm0.08}$ | $5.00_{\pm0.02}$ | $75.87_{\pm0.08}$ | 1.72 | $0.00_{\pm0.00}$ | 140 | $65.02_{\pm0.10}$ | $0.00_{\pm0.00}$ | $64.65_{\pm0.11}$ | 0.20 | $0.00_{\pm0.00}$ | 182 |
| NG | $69.76_{\pm0.01}$ | $0.00_{\pm0.00}$ | $69.23_{\pm0.01}$ | 4.43 | $0.00_{\pm0.00}$ | 2 | $59.62_{\pm0.00}$ | $0.00_{\pm0.00}$ | $59.26_{\pm0.00}$ | 3.79 | $1.80_{\pm0.00}$ | 3 |
| RL | $65.98_{\pm0.12}$ | $5.22_{\pm0.45}$ | $65.52_{\pm0.11}$ | 8.66 | $0.00_{\pm0.00}$ | 12 | $53.41_{\pm0.00}$ | $0.00_{\pm0.00}$ | $53.04_{\pm0.00}$ | 7.94 | $2.00_{\pm0.00}$ | 10 |
| SCAR | $71.33_{\pm0.12}$ | $5.61_{\pm0.89}$ | $70.66_{\pm0.14}$ | 5.29 | $13.28_{\pm0.67}$ | 367 | $59.98_{\pm0.06}$ | $0.00_{\pm0.00}$ | $59.62_{\pm0.06}$ | 3.55 | $0.67_{\pm0.12}$ | 1052 |
| JiT | $65.44_{\pm0.14}$ | $3.00_{\pm0.76}$ | $64.87_{\pm0.13}$ | 8.32 | $4.44_{\pm0.30}$ | 15 | $53.82_{\pm0.09}$ | $0.00_{\pm0.00}$ | $53.16_{\pm0.08}$ | 7.76 | $5.29_{\pm0.25}$ | 5 |
| MU-Mis | $76.42_{\pm0.07}$ | $0.00_{\pm0.00}$ | $75.64_{\pm0.07}$ | **0.07** | $0.00_{\pm0.00}$ | 30 | $64.95_{\pm0.00}$ | $0.00_{\pm0.00}$ | $64.85_{\pm0.00}$ | 0.15 | $0.20_{\pm0.00}$ | 83 |

Table 2: Performance overview for **sub-class** unlearning task evaluated on 'Rocket' and 'Sea' (where the retrain model exhibits different degrees of generalization ability on the unlearned sub-class) of CIFAR-20 using ResNet-18. The content format follows Table 1.

| Method | Rocket | | | | | | Sea | | | | | |
|---|---|---|---|---|---|---|---|---|---|---|---|---|
| | RA | FA | TA | Avg. Gap↓ | MIA | RTE | RA | FA | TA | Avg. Gap↓ | MIA | RTE |
| Pretrain | 85.26 | 80.73 | 85.21 | 26.53 | 92.89 | 6910 | 85.09 | 97.66 | 85.21 | 5.94 | 91.81 | 6910 |
| Retrain | $84.85_{\pm0.09}$ | $2.69_{\pm0.45}$ | $84.07_{\pm0.10}$ | 0.00 | $12.06_{\pm0.75}$ | 4298 | $84.60_{\pm0.22}$ | $80.93_{\pm2.20}$ | $84.61_{\pm0.19}$ | 0.00 | $51.61_{\pm3.60}$ | 4298 |
| BT | $85.24_{\pm0.02}$ | $2.80_{\pm0.45}$ | $84.36_{\pm0.02}$ | 0.26 | $0.00_{\pm0.00}$ | 27 | $82.51_{\pm0.00}$ | $81.00_{\pm0.00}$ | $82.63_{\pm0.00}$ | 1.38 | $15.00_{\pm0.00}$ | 47 |
| FT | $82.70_{\pm0.19}$ | $4.20_{\pm1.30}$ | $81.97_{\pm0.12}$ | 1.92 | $5.40_{\pm1.04}$ | 138 | $82.36_{\pm0.29}$ | $88.00_{\pm1.41}$ | $82.43_{\pm1.60}$ | 3.83 | $58.08_{\pm1.79}$ | 417 |
| SCRUB | $84.73_{\pm0.13}$ | $5.80_{\pm1.30}$ | $83.84_{\pm0.13}$ | 1.15 | $13.28_{\pm0.02}$ | 113 | $84.86_{\pm0.10}$ | $88.17_{\pm1.72}$ | $84.86_{\pm0.13}$ | 2.58 | $57.07_{\pm1.71}$ | 113 |
| SSD | $84.23_{\pm0.05}$ | $2.60_{\pm0.89}$ | $83.35_{\pm0.06}$ | 0.48 | $3.76_{\pm0.36}$ | 18 | $84.79_{\pm0.00}$ | $78.00_{\pm0.00}$ | $84.61_{\pm0.00}$ | 1.24 | $8.00_{\pm0.00}$ | 7 |
| DUCK | $82.09_{\pm0.33}$ | $19.4_{\pm3.28}$ | $81.43_{\pm0.35}$ | 7.37 | $32.84_{\pm1.57}$ | 58 | $80.95_{\pm0.19}$ | $66.45_{\pm2.30}$ | $80.77_{\pm0.19}$ | 7.34 | $54.92_{\pm2.29}$ | 68 |
| SalUn | $84.82_{\pm0.06}$ | $2.99_{\pm1.25}$ | $84.00_{\pm0.05}$ | **0.13** | $0.00_{\pm0.00}$ | 1042 | $82.85_{\pm0.00}$ | $81.00_{\pm0.00}$ | $83.10_{\pm0.00}$ | 1.11 | $13.40_{\pm0.00}$ | 63 |
| MUNBa | $81.43_{\pm0.13}$ | $7.00_{\pm0.05}$ | $80.80_{\pm0.12}$ | 3.67 | $7.20_{\pm0.14}$ | 362 | $80.64_{\pm0.14}$ | $84.00_{\pm0.15}$ | $80.66_{\pm0.13}$ | 3.66 | $60.00_{\pm0.20}$ | 564 |
| LoTus | $35.93_{\pm0.21}$ | $39.00_{\pm0.61}$ | $36.04_{\pm0.20}$ | 44.42 | $18.60_{\pm0.18}$ | 105 | $73.12_{\pm0.15}$ | $81.00_{\pm0.16}$ | $73.39_{\pm0.14}$ | 7.59 | $61.20_{\pm0.19}$ | 16 |
| NG | $62.84_{\pm5.66}$ | $5.67_{\pm4.08}$ | $62.48_{\pm5.59}$ | 15.52 | $72.70_{\pm21.80}$ | 4 | $80.95_{\pm0.00}$ | $75.00_{\pm0.00}$ | $80.84_{\pm0.02}$ | 4.45 | $60.00_{\pm0.00}$ | 3 |
| RL | $60.89_{\pm1.96}$ | $6.52_{\pm1.07}$ | $60.50_{\pm2.01}$ | 17.11 | $3.70_{\pm6.51}$ | 5 | $80.48_{\pm0.02}$ | $77.00_{\pm0.00}$ | $80.34_{\pm0.02}$ | 4.11 | $48.70_{\pm0.11}$ | 3 |
| SCAR | $76.49_{\pm0.12}$ | $43.81_{\pm4.44}$ | $76.26_{\pm0.23}$ | 19.09 | $28.04_{\pm1.67}$ | 442 | $76.30_{\pm0.15}$ | $77.40_{\pm2.71}$ | $76.12_{\pm0.19}$ | 6.77 | $51.84_{\pm1.98}$ | 434 |
| JiT | $59.15_{\pm0.05}$ | $4.00_{\pm0.00}$ | $58.60_{\pm0.05}$ | 17.49 | $29.03_{\pm0.20}$ | 4 | $51.48_{\pm0.04}$ | $7.20_{\pm1.10}$ | $51.04_{\pm0.04}$ | 46.81 | $32.20_{\pm0.24}$ | 4 |
| MU-Mis | $84.28_{\pm0.18}$ | $2.91_{\pm1.02}$ | $83.50_{\pm0.19}$ | 0.49 | $0.07_{\pm0.25}$ | 21 | $84.35_{\pm0.03}$ | $81.00_{\pm2.95}$ | $84.33_{\pm0.05}$ | **0.20** | $1.25_{\pm1.85}$ | 21 |

**MU-Mis outperforms existing remaining-data-free methods significantly and remains highly competitive with SoTA remaining-data-dependent methods.** Table 1 and Table 2 correspond to MU performances on full-class and sub-class unlearning respectively. More experiment results are referred to Appendix G.1. In terms of unlearnig *utility*, MU-Mis achieves the smallest Avg. Gap in full-class-CIFAR-100, full-class-PinsFaceRecognition, sub-class-Sea and sub-class-Lamp unlearning, outperforming all the baseline methods. From the highlighted values in red in the tables, we could see that existing remaining-data-free methods suffer from poor utility preservation. From Table A8, we could see that there is a clear gap between MU-Mis and RUM in when removing mixture of different memorization level samples. But surprisingly, MU-Mis indicates a lowest KL divergence to the retrained model in the forgetting data, indicating a more principled removal than SalUn and RUM. Overall, MU-Mis surpasses strong remaining-data-dependent methods in full-class and sub-class unlearning, falls short of the RUM in the particularly challenging random-subset setting. But importantly, MU-Mis outperforms all existing remaining-data-free methods by a substantial margin across all scenarios. In terms of *privacy*, MIA-Entropy indicates the residual membership of the forgetting data and MIA-SCRUB indicates non-membership of the forgetting data in the unlearned model. We can see that MIA-Entropy remain consistently low and MIA-SCRUB remain consistently to the retrained model in Table A11 cross 3 tasks, collectively demonstrating a successful privacy protection of MU-Mis. In addition to resolving the issue of constrained access to the remaining data, our remaining-data-free method also offers a notable advantage in MU efficiency. In unlearning a full class Tiny ImageNet, MU-Mis is up to $30\times$ faster than SalUn, with only 0.09 higher Avg. Gap.

**Efficiency advantage is more pronounced on larger scale models.** Table 1 shows the performance when unlearning a full class of Tiny ImageNet under ViT (Dosovitskiy et al., 2021). MU-Mis outperforms other remaining-data-free methods [1] significantly and performs comparably with the most competitive method SalUn in terms of model utility and privacy. The efficiency advantage of MU-Mis becomes

Table 3: Performance overview for **full class** unlearning task evaluated on **Tiny ImageNet** using **ViT**. RTE is reported in **minute**.

| Methods | RA | FA | TA | Avg. Gap $\downarrow$ | MIA | RTE (min) |
|---|---|---|---|---|---|---|
| Pretrain | 84.21 | 87.5 | 84.23 | 30.44 | 95.40 | - |
| Retrain | $86.35_{\pm 0.17}$ | $0.00_{\pm 0.00}$ | $85.92_{\pm 0.14}$ | 0.00 | $8.80_{\pm 0.26}$ | - |
| Salun | $83.94_{\pm 0.16}$ | $0.00_{\pm 0.00}$ | $83.49_{\pm 0.14}$ | 1.88 | $0.00_{\pm 0.00}$ | 81 |
| NG | $63.12_{\pm 0.00}$ | $0.00_{\pm 0.00}$ | $62.88_{\pm 0.00}$ | 15.28 | $0.00_{\pm 0.00}$ | 0.21 |
| RL | $67.69_{\pm 0.00}$ | $0.00_{\pm 0.00}$ | $67.43_{\pm 0.00}$ | 12.38 | $0.00_{\pm 0.00}$ | 0.15 |
| MU-Mis | $82.13_{\pm 0.24}$ | $0.00_{\pm 0.00}$ | $82.17_{\pm 0.23}$ | 2.69 | $0.00_{\pm 0.00}$ | 3 |

markedly pronounced: the unlearning time is reduced from more than **1 hour** to **3 minutes**. We also evaluate subclass-CIFAR20-sea unlearning under ViT and show the results in Table A10, where MU-Mis exhibits the best Avg.Gap and is $20\times$ faster than SalUn.

## 5.3 UNLEARNING RESILIENCE: SEQUENTIAL MACHINE UNLEARNING

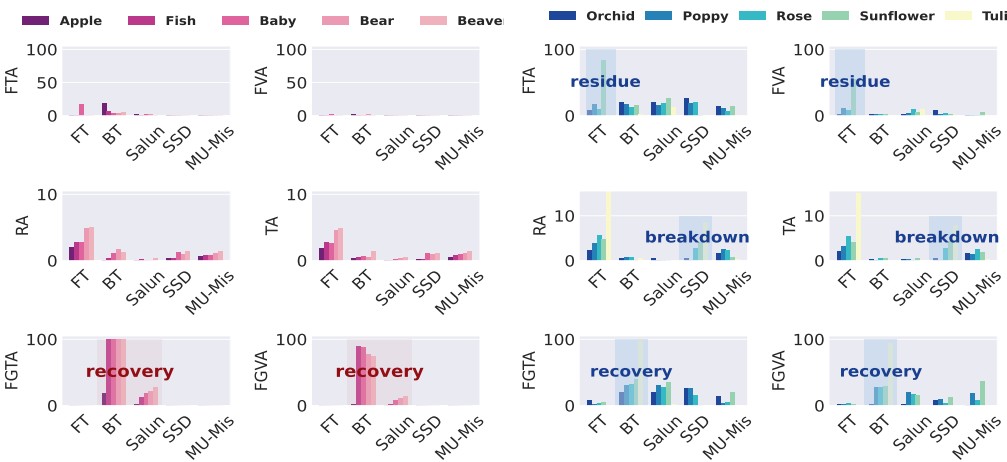

Figure 5: **Disparities in accuracy-related metrics between the unlearned model and the retrained model for full class and sub-class sequential unlearning.** Left: Iteratively unlearns 5 distinct full classes of CIFAR-100. Right: Iteratively unlearns 5 sub-classes of the same super-class 'Flower'.

**Sequential unlearning requires principled unlearning mechanisms.** In practice, unlearning requests may arrive sequentially, requiring multiple executions of the unlearning method. Wang et al. (2024a) point out that sequential unlearning greatly challenges the memorization management ability of unlearning methods due to underlying associations among unlearned classes. The sequentially unlearned model might break down due to disordered forgetting operation, exposing its accumulated effects on model knowledge. Therefore, to highlight the importance of principled forgetting, we perform sequential unlearning. We examine the impact of subsequent requests on previous unlearning efforts and present the disparities between the unlearned model and the retrain model at each iteration in accuracy-related MU metrics in Figure 5. For detailed experiment settings, refer to Appendix F.1.

**Dificiencies in existing MU methods.** From Figure 5, we can see that there are 3 kinds of deficiencies in existing SoTA MU methods:

*(i) Performance Recovery.* The performance on the forgotten classes stages a recovery in BT and Salun unlearned model, indicated by the above zero FGTA and FGVA. This suggests that retargeting model's outputs of the forgetting data does not completely remove associated knowledge, posing a substantial risk since the concealed information might still be exploited by privacy attackers.

*(ii) Knowledge Residue.* High disparity of FTA and FVA in sub-class task indicates that FT method, which relies on "catastrophic forgetting" (Kirkpatrick et al., 2017) to unlearn, fails to unlearn effectively in sub-class task due to the resemblance between the forgetting and remaining data.

*(iii) Utility Breakdown.* In sub-class task, SSD exhibits a marked decline in utility after the last unlearning request, demonstrated by the final RA of 76.33%. In contrast, RA in the retrained model

---

[1]We failed to identify effective hyper-parameters for JiT and SCAR for this experiment.

and MU-Mis unlearned model are respectively 84.83% and 84.59%. Such a plummet implies a potential risk of model utility breakdowns when the magnitude of parameters is continuously scaled.

**Resilient performance of MU-Mis to sequential unlearning requests.** To facilitate an intuitive assessment in terms of utility and resilience, we compute the utility Avg. Gap and resilience Avg. Gap for each iteration in Figure 6. The utility Avg. Gap is averaged over FTA, FVA, RA and TA, and the resilience Avg. Gap is averaged over FGTA and FGVA. From Figure 6, it is evident that MU-Mis and SSD are significantly better than BT, FT, and Salun, demonstrating a notably small disparity to the retrained model regarding both the utility and resilience Avg. Gap across the full class and sub-class tasks. Importantly, MU-Mis achieves these results without relying on the remaining data, which are required by SSD.

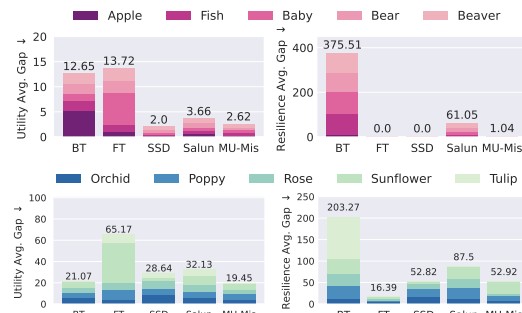

Figure 6: **Overview of utility Avg. Gap and resilience Avg. Gap during full class (upper) and sub-class (bottom) sequential unlearning.**

**Minimal KL divergence of MU-Mis from retrain model during sequential unlearn.** Beyond model predictions, we further provide the empirical KL divergence (introduced in Appendix F.2) between the unlearned model and the retrained model's output distributions during sequential requests in Figure 7. It is evident that MU-Mis exhibits the lowest KL divergence from the retrained model throughout both the full-class and sub-class sequential unlearning processes.

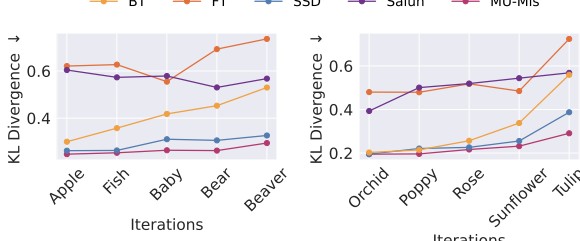

Figure 7: **KL divergence between outputs of unlearned model and retrained model during full class (left) and sub-class (right) sequential unlearning.**

**Summary.** In general, MU-Mis stands out with its comprehensive capabilities in terms of unlearning utility, unlearning resilience as well as output indistinguishability, while current SoTA MU methods are disclosed to exhibit limitations and deficiencies in certain aspects. Their inappropriate or inadequate unlearning approaches undermine their reliability and applicability in practical scenarios.

## 5.4 SUPPLEMENTARY EXPERIMENTS AND ANALYSES

We provide the following experiments and analyses for completeness in Appendix: *(i)* ablation study of MU-Mis in Appendix G.3; *(ii)* a hyper-parameter sensitivity analysis showing stability of MU-Mis in Appendix G.4; *(iii)* visualizations of attention map confirming effectiveness of MU-Mis in Appendix G.5; *(iv)* an empirical analysis attributing the effectiveness of MU-Mis to the orthogonality of input sensitivity gradients among samples in Appendix H; *(vi)* A comprehensive analysis covering the theoretical link between sensitivity gaps and loss curvature, empirical signatures of sensitivity across memorization and influence levels, and the broader role of unlearning in shaping memorization, generalization, and sample contribution in Appendix I.

## 6 CONCLUSION

There are 3 main challenges in machine unlearning: *the stochasticity of training*, *incrementality of training*, and *catastrophe of unlearning* (Wang et al., 2024b). We address incrementality by quantifying sample contribution through the lens of input sensitivity. Building on this, our proposed MU-Mis achieves effective and efficient unlearning without compromising model utility, alleviating catastrophic unlearning. Experiments validate the superiority of this principled forgetting mechanism. Overall, MU-Mis is well-grounded, lightweight and remaining-data-free, offering a practical and competitive alternative to existing unlearning methods. Furthermore, we highlight in Appendix I that there is a profound connection between input sensitivity view and machine unlearning, which we believe is an interesting direction to further improve remaining-data-free unlearning in the most challenging random subset scenario.

ACKNOWLEDGMENTS

The authors thank the anonymous reviewers for their insightful comments and constructive suggestions, which have helped improve the quality of this manuscript. The research leading to these results has received funding from the National Natural Science Foundation of China (No. 62376155) and the AI for Science Program, Shanghai Municipal Commission of Economy and Informatization (No. 2025-GZL-RGZN-BTBX-02026).

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

APPENDIX

# A    ETHICS STATEMENT

This work studies machine unlearning (MU), motivated by the "right to be forgotten," with the goal of enhancing user privacy and data protection. All experiments are conducted on publicly available datasets and standard benchmark models; no sensitive or personally identifiable information is used. While unlearning techniques could in principle be misused to manipulate model behavior, our focus is on strengthening trust and accountability in machine learning systems. We believe this work contributes positively to the development of privacy-preserving and ethically responsible AI.

# B    REPRODUCIBILITY STATEMENT

We include anonymized supplementary materials containing the complete algorithm implementations for executing all experiments. We provide detailed experimental settings, hyperparameters, datasets, and evaluation metrics in our Appendix to ensure reproducibility.

# C    THE USE OF LLMS

Large language models (LLMs) were employed solely as auxiliary writing tools. Their usage was strictly limited to surface-level assistance, including grammar correction, stylistic polishing, clarity improvement, and formatting consistency. LLMs were not involved in formulating research ideas, designing methods, conducting analyses, interpreting results, or drawing conclusions. At no stage were LLMs used to generate original content, experimental designs, or theoretical claims. All text segments refined with LLM assistance were subsequently reviewed, validated, and, where necessary, rewritten by the authors to ensure technical accuracy and precision of expression. The authors bear full responsibility for the final presentation and content of this paper. This disclosure is made in accordance with conference guidelines on LLM usage to ensure transparency and research integrity.

# D    A TOY EXAMPLE COMPLEMENTING SAMPLE CONTRIBUTION DERIVATION

We use a simple MLP model to illustrate the insight of $\mathcal{S}_k(\hat{x}, \tilde{x}) << \mathcal{S}_k(\hat{x}, \hat{x})$ with $\hat{x} \neq \tilde{x}$. Assume the $l^{th}$ layer output of model is $x^l = \phi(\theta^l x^{l-1})$, where $\theta^l$ refers to $l^{th}$ layer parameter and $\phi$ refers to activation function. Then,

$$g_k = \frac{\partial f_k}{\partial \theta^l} = \frac{\partial f_k}{\partial (\theta^l x^{l-1})} x^{l-1^T}, \tag{A1}$$

$$g_k' = \frac{\partial f_k}{\partial \theta^l \partial x} = \frac{\partial f_k}{\partial (\theta^l x^{l-1})} \frac{\partial x^{l-1^T}}{\partial x}$$

$$= \frac{\partial f_k}{\partial (\theta^l x^{l-1})} \phi'(\theta^{l-1} x^{l-2}) \theta^{l^T} \frac{\partial x^{l-2}}{\partial x}. \tag{A2}$$

Thereby, the inner-dot $g_k'(\hat{x}) g_k(\tilde{x}) \propto \tilde{x}^{l-1} \phi'(\theta^{l-1} \hat{x}^{l-2})$. If ReLU activation is used, where $\phi'(x) = 1$ if $x > 0$ else $\phi'(x) = 0$, $\mathcal{S}_k(\hat{x}, \tilde{x}) \propto g_k'(\hat{x}) g_k(\tilde{x})$ will be quite small. The conclusion here is that the residual term is relatively smaller than the contribution term. Therefore, the contribution of a training sample to the training process would be approximately reflected in the pre-trained model's output sensitivity to the sample.

Additionally, we investigate sensitivity signatures across different samples (*i.e.*, different memorization levels and influence scores) in Appendix I.2. We find that highly memorized samples exhibit smaller sensitivity gap than low and middle memorized sample, and more influential samples exhibit higher sensitivity gap. This empirical evidence further confirms that our proposed sensitivity gap successfully reflect sample contribution and the residual term is not that crucial to some extent.

# E STOPPING GUIDANCE

The optimization should cease once the withdrawal is completed, requiring a stopping guideline for practical use. We monitor both the optimization objective and unlearning metrics during minimizing the MU-Mis loss Eq.(3) in Figure A1, using the example of unlearning with ResNet-18 on fullclass-CIFAR100-rocket. In Figure A1, different colors represent different learning rates and the purple dashed line represents the accuracy of the retrained model. A consistent trend on accuracy change during unlearning is observed across different learning rates: as the optimization of MU-Mis loss progresses, the accuracy of the forgetting data (FA) gradually decreases, the accuracy of the remaining (RA) and test data (TA) first decrease slightly and then grow up with the recovery of $\|\nabla_x f_{c'}(x, w_p)\|_F$.

Notably, when $\|\nabla_x f_{c'}(x, w)\|_F$ recovers close to the level of its initial value $\|\nabla_x f_{c'}(x, w_p)\|_F$, RA approaches the retrained model across different learning rates. There exists a clear relationship between the unlearning progress and model performance, allowing for effective unlearning by stopping the optimization timely. To this end, we introduce a stopping threshold ratio $\delta$ to regulate the time of stopping. We record the minimal value of irrelevant class logit sensitivity as $\epsilon$ and terminate unlearning process when $\|\nabla_x f_{c'}(\mathcal{D}_f, w)\|_F > \epsilon$ and $\frac{\|\nabla_x f_{c'}(\mathcal{D}_f, w)\|_F}{\|\nabla_x f_{c'}(\mathcal{D}_f, w_p)\|_F} > \delta$.

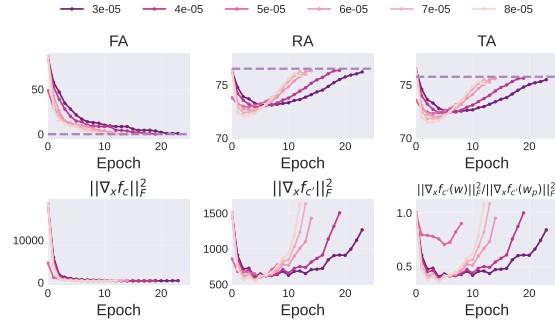

Figure A1: Accuracy and optimization objective during fullclass-CIFAR100-rocket unlearning with different learning rates on ResNet-18. FA decreases gradually, RA and TA first drop slightly and then rise with the recovery of $\|\nabla_x f_{c'}(x, w)\|_F$. The endpoint of each curve corresponds to the time when $\|\nabla_x f_{c'}\|_F$ exceeds 90% of its initial value.

# F EXPERIMENT DETAILS

## F.1 EXPERIMENT SETTING

**Tasks, Datasets and Models.** We investigate 3 kinds of unlearning tasks in supervised image classification scenarios, including forgetting a full class, a sub-class under a super-class, and a random subset. We evaluate full class unlearning on CIFAR-100 (Krizhevsky et al., 2009), PinsFaceRecognition (Burak, 2020), and Tiny ImageNet (Le & Yang, 2015), sub-class unlearning on three sub-classes of CIFAR-20 (Krizhevsky et al., 2009), random subset unlearning on CIFAR-10 (Krizhevsky et al., 2009) and SVHN (Netzer et al., 2011). We perform sequential unlearning by iteratively unlearning a full and a sub-class to evaluate the algorithm's robustness to privacy onion effect (Carlini et al., 2022). For iterative full class MU, we iteratively unlearn 5 distinct classes (label 0-4, corresponding to "Apple", "Fish", "Baby", "Bear" and "Beaver") of CIFAR-100 in line with Wang et al. (Wang et al., 2024a). For sub-class setting, we iteratively unlearn 5 sub-classes ("orchid", "poppy", "rose", "sunflower", "tulip") under the same superclass "flower" of CIFAR-20. We use ResNet-18 (He et al., 2016) for all the above experiments. To further indicate the significant efficiency advantage of our remaining-data-free method, we perform full class unlearning on Tiny-ImageNet and sub-class unlearning on CIFAR-20 with ViT.

**Evaluation Metrics.** MU methods should be assessed from three aspects: *utility*, *privacy*, and *efficiency*(Xu et al., 2024). Beyond that, in practice, where the unlearning requests are made constantly, the unlearning *resilience* should be assessed, i.e. subsequent unlearning should not spoil previous unlearning efforts. For *utility*, we compute forgetting data accuracy (**FA**), remaining data accuracy (**RA**), and test data accuracy (**TA**) of the unlearned model. FA and RA are computed on the valid set in class-wise unlearning and on the train set in random subset unlearning. We compute the average gap (**Avg. Gap**) between the retrained model and the unlearned model on accuracy-related metrics, including FA, RA and TA to illustrate the utility disparity. In terms of *resilience*, we evaluate on sequential unlearning tasks. We compute the train (**FGTA**) and valid (**FGVA**) accuracy on the forgotten classes and quantify the unlearning resilience with the average of their disparity to the retrained model (**Resilience Avg. Gap**). To further examine the indistinguishability between the

retrained and unlearned model, we compute the **KL divergence** between their output distributions over the entire dataset. The retrained model is an oracle of approximate MU, therefore, above disparity metrics should be as small as possible. Regarding the *privacy* guarantee, we use membership inference attack (**MIA**) (Chundawat et al., 2023) to probe the remaining information of the forgetting data. The MIA success rate indicates how many samples in $\mathcal{D}_f$ are predicted as membership samples of the unlearned model. From a privacy perspective, a lower MIA value implies less information leakage in the unlearned model and is preferred (Liu et al., 2024). For *efficiency*, we provide the run time efficiency (**RTE**) in **seconds** to indicate timeliness.

**Baselines.** We compare our method along with 6 baselines which utilize the remaining data, as well as 4 remaining-data-free methods. The 6 baselines include Bad Teacher (**BT**) (Chundawat et al., 2023),Finetune (**FT**) (Warnecke et al., 2023), SCalable Remembering and Unlearning unBound (**SCRUB**) (Kurmanji et al., 2023), Selective-Synaptic-Dampening (**SSD**) (Foster et al., 2024b), Distance-based Unlearning via Centroid Kinematics (**DUCK**) (Cotogni et al., 2023), Saliency-based unlearning (**SalUn**) (Fan et al., 2024b), Large-Scale Machine Unlearning with a Taste of Uncertainty (**LoTUS**) (Spartalis et al., 2025), Machine Unlearning via Nash Bargaining (**MUNBa**) (Wu & Harandi, 2025). We also add a strong remaining-data-dependent method Refined-Unlearning Meta-algorithm (**RUM**) (Zhao et al., 2024) for the most challenging random subset setting. The 4 remaining-data-free methods include Random Labeling (**RL**) (Golatkar et al., 2020), Negative Gradient (**NG**) (Thudi et al., 2022), Just in Time unlearning (**JiT**) (Foster et al., 2024a) and Selective-distillation for Class and Architecture-agnostic unleaRning (**SCAR**) (Bonato et al., 2024).

The detailed method of each baseline is as the following:

- **BT** (Chundawat et al., 2023): Bad Teacher transfers knowledge from useful and useless teachers for the remaining data and the forgetting data. The code source is `https://github.com/if-loops/selective-synaptic-dampening`.

- **FT** (Warnecke et al., 2023): Finetune optimizes the pre-trained model with the remaining data, unlearning relying on "catastrohpic forgetting". The code source is `https://github.com/if-loops/selective-synaptic-dampening`.

- **SCRUB** (Kurmanji et al., 2023): SCRUB aims to push outputs of the student model (the unlearned model) away from the teacher model (the pre-trained model) to distill knowledge. This is achieved by first performing several max-steps (distill the knowledge) and then perform several min-steps (regain performance on the remaining data with cross-entropy loss). The code source is `https://github.com/meghdadk/SCRUB`.

- **SSD** (Foster et al., 2024b): SSD uses the Fisher information matrix to assess parameter importance and suppress parameters that are important to the forgetting data while less important to the remaining data. The code source is `https://github.com/if-loops/selective-synaptic-dampening`.

- **DUCK** (Cotogni et al., 2023): DUCK employs metric learning to guide the removal of samples matching the nearest incorrect centroid in the embedding space. The code source is `https://github.com/OcraM17/DUCK`.

- **SalUn** (Fan et al., 2024b): SalUn computes weight saliency map to enable the most important weights for the forgetting data. The code source is `https://github.com/OPTML-Group/Unlearn-Saliency`.

- **LoTUS** (Spartalis et al., 2025):LoTUS performs large-scale machine unlearning by estimating and propagating uncertainty to guide parameter update suppression, enabling scalable forgetting without relying on remaining data. The source code is `https://github.com/sohomghosh/LoTUS`.

- **MUNBa** (Wu & Harandi, 2025): MUNBa formulates machine unlearning as a Nash bargaining problem and jointly optimizes forgetting and retention objectives to balance utility preservation and effective removal. The source code is `https://github.com/OPTML-Group/MUNBa`.

- **RUM** (Zhao et al., 2024): RUM analyzes fundamental factors that impact unlearning difficulty (e.g., embedding-space entanglement between forgotten and retained data, and memorization levels), and proposes a meta-algorithm that partitions the forget set into homogeneous subsets and applies per-subset unlearning to improve performance. The source code is `https://github.com/kairanzhao/RUM`.

- **NG** (Thudi et al., 2022): Negative gradient computes several steps of gradient ascent with the forgetting data. The source code is https://github.com/jbonato1/scar.
- **RL** (Golatkar et al., 2020): Random label relabels the forgetting data with randomly assigned class and fine-tune the model with computed cross-entropy loss. The source code is https://github.com/jbonato1/scar.
- **SCAR** (Bonato et al., 2024): SCAR utilizes Out-of-distribution (OOD) data as a surrogate for the remaining data and distills the knowledge of the original model into the unlearned model to preserve model utility. The source code is https://github.com/jbonato1/scar.
- **JiT** (Foster et al., 2024a): JiT smooths the model output around the forgetting data by minimizing the local Lipschitz constant. The source code is https://github.com/jwf40/Information-Theoretic-Unlearning.

## F.2 KL Divergence

The KL divergence between two distributions is:

$$D_{\mathrm{KL}}(p_z(w_r)||p_z(w_u)) = \int p_z(w_r) \log[p_z(w_r)/p_z(\theta)]\mathrm{d}\mathcal{D} \tag{A3}$$

We calculate empirical KL divergence with the entire dataset (including both the train and valid set). We first collect the predicted class probabilities from both the unlearned and retrained models of each sample, then we compute the output KL divergence as follows:

$$D_{\mathrm{KL}} = \frac{1}{N} \sum_{i=1}^{N} \sum_{c=1}^{C} p_c(x_i; w_r) \log \frac{p_c(x_i; w_r)}{p_c(x_i; w_u)}, \tag{A4}$$

where $N$ is total number of dataset, $C$ denotes the total number of classes, $w_u$ is the unlearned model parameter and $w_r$ is the retrained model parameter. $p_c(x_i, w)$ represents the $c$-th posterior probability of $i$-th sample in model $w$.

## F.3 Training Details

For **ResNet-18**, training uses SGD with a momentum of 0.9, weight decay of $5 \times 10^{-4}$, and batch size of 128 with a learning rate initialized at 0.1. The learning rate decays at 60,120,160 by 0.1 with a total of 200 epochs.

For **ViT**, we initialize with model pre-trained on ImageNet provided by torchvision. Then we randomly initialize the last fully connected layers and train it with SGD with a momentum of 0.9, weight decay of $5 \times 10^{-4}$, batch size 64, constant learning rate $\eta = 0.1$ for 10 epochs. All the experiments are conducted on a single RTX 4090.

## F.4 Hyper-parameters

For **MU-Mis**, we optimize the pretrained model under **model.eval()** mode with **vanilla SGD** without momentum for ResNet-18 and Adam for ViT. For only the forgetting data are used in MU-Mis, we must freeze the batch norm layers to avoid spoiling the remaining data. We use batch size of 256 for MU-Mis across all the experiments with ResNet-18 and batch size of 32 with ViT. In random subset unlearning, the target class $c'$ is fixed as 1 to facilitate unlearning. We report the learning rate $\eta$ and stopping threshold $\delta$ used in different settings in Table A1 for reproducibility.

For each baseline, We perform grid search to find the best hyper-parameters in each setting. The hyper-parameter sweep range for each method is presented in Table A4. The hyper-parameters used for all the methods in sequential full class and sub-class tasks are shown in Table A2 and Table A3. For all the methods, we fix batch size as 256 for ResNet-18 and 64 for ViT unless otherwise stated in hyper-parameter ranges. For BT, we use constant learning rate. For fine-tune based methods, e.g. FT, as well as SCRUB and SalUn, we use cosine scheduler. We fix temperature= 1, alpha = 0.5, gamma = 0.99, weight decay = $5 \times 10^{-4}$ for SCRUB. We fix weight decay = $5 \times 10^{-4}$ for DUCK and SCAR.

**Reweighted Variant of MU-Mis Loss.** In practice, the TC (Target Class) and OC (Other Classes) terms in Eq. (3) can be re-weighted to enhance the optimization process of the MU-Mis loss. The ablation study in Table A12 implies that the TC term promotes the unlearning and the OC term contributes to the recovery of performance on the remaining data. As shown in Figure A1, TC term is much larger than OC and dominants the MU-Mis loss at the beginning of optimization. To facilitate the unlearning and subsequent recovery, we could amplify the significance of the TC term when its magnitude is bigger than OC term, and conversely prioritize the OC term when the TC term's influence diminishes in comparison. The parameter $\tau$ serves as an indicator of this dynamic re-weighting switch, while $\kappa$ quantifies the strength of such adjustment. Mathematically, the reweighted MU-Mis loss has the following formulation:

$$\mathcal{L}(w, \mathcal{D}_f) = \frac{1}{N} \sum_{\mathbf{x} \in \mathcal{D}f} [\alpha_c \cdot \|\nabla_{\mathbf{x}} f_c(w, \mathbf{x})\|_F^2 - \alpha_{c'} \cdot \|\nabla_{\mathbf{x}} f_{c'}(w, \mathbf{x})\|_F^2], \tag{A5}$$

where

$$\alpha_c = (\kappa - 1)\tau + 1,$$
$$\alpha_{c'} = (1 - \kappa)\tau + \kappa,$$
$$\tau = \mathbb{I}(\|\nabla_{\mathbf{x}} f_c(w, \mathbf{x})\|_F^2 > \|\nabla_{\mathbf{x}} f_{c'}(w, \mathbf{x})\|_F^2).$$

Thereby, we have

$$\begin{cases} \alpha_c = \kappa, \alpha_{c'} = 1 \text{ if } \|\nabla_{\mathbf{x}} f_c(\mathbf{x})\|_F^2 \geq \|\nabla_{\mathbf{x}} f_{c'}(\mathbf{x})\|_F^2, \\ \alpha_c = 1, \alpha_{c'} = \kappa \text{ if } \|\nabla_{\mathbf{x}} f_c(\mathbf{x})\|_F^2 < \|\nabla_{\mathbf{x}} f_{c'}(\mathbf{x})\|_F^2. \end{cases}$$

The original MU-Mis loss Eq. (3) corresponds to $\kappa = 1$. We apply this strategy in sub-class sequential MU task and Tiny-ImageNet dataset. We set $\kappa = 80$ for iterations $1 \sim 3$ and $\kappa = 1$ for iterations $4 \sim 5$ in sequential sub-class, and $\kappa = 80$ for Tiny-ImageNet.

Table A1: Hyperparameters of MU-Mis h(learning rate $\eta$ and stopping threshold ratio $\delta$).

| Setting | $\eta$ | $\delta$ |
|---|---|---|
| fullclass-CIFAR-100 | $7 \times 10^{-5}$ | 1.45 |
| fullclass-PinsFaceRecognition | $4 \times 10^{-4}$ | 0.68 |
| fullclass-Tiny-ImageNet | $4 \times 10^{-6}$ | 1.1 |
| subclass-CIFAR-20-rocket | $3 \times 10^{-5}$ | 3.00 |
| subclass-CIFAR-20-sea | $2 \times 10^{-5}$ | 0.93 |
| subclass-CIFAR-20-lamp | $5 \times 10^{-5}$ | 1.80 |
| fullclass-Tiny-ImageNet-ViT | $5 \times 10^{-4}$ | 1.03 |

Table A2: Hyperparameters for full class sequential unlearning in Fig. 5.

| Methods | Hyperparameters |
|---|---|
| Retrain | epoch = 200, lr = 0.1, milestones = [60, 120, 160]. |
| BT | epoch = 10, lr = $\{5, 5, 5, 1, 5\} \times 10^{-5}$, temperature scalar = $\{3, 1, 1, 1, 5\}$. |
| FT | epoch = 10, lr = $10^{-1}$ for all iterations. |
| SSD | dampening constant $\lambda = \{1, 1, 1, 1, 0.1\}$, selection weight $\alpha = \{95, 70, 50, 70, 80\}$. |
| SalUn | epoch = 10, lr = $10^{-3}$, threshold = 0.6 for all iterations. |
| MU-Mis | epoch = 50, lr = $\{2, 1, 1, 0.5, 0.8\} \times 10^{-4}$ |

Table A3: Hyperparameters for subclass sequential unlearning in Fig. 5.

| Methods | Hyperparameters |
|---|---|
| Retrain | epoch = 200, lr = 0.1, milestones = [60, 120, 160]. |
| BT | epoch = $\{5, 10, 5, 10, 5\}$, lr = $\{0.5, 0.5, 1, 1, 5\} \times 10^{-5}$, temperature scalar = $\{5, 5, 5, 3, 1\}$. |
| FT | epoch = 20, lr = $10^{-1}$ for all iterations. |
| SSD | dampening constant $\lambda = \{1, 0.1, 0.1, 1, 1\}$, selection weight $\alpha = \{71, 100, 90, 87, 85\}$. |
| SalUn | epoch = 10, lr = $10^{-3}$, threshold = 0.6 for all iterations. |
| MU-Mis | epoch = 30, lr = $\{5, 1, 0.1, 3, 3\} \times 10^{-6}$, stopping threshold $\delta = \{1.4, 1.4, 0.95, 10, 10\}$. |

Table A4: Hyper-parameters range overview for different methods in all the experiments.

| Methods | Hyperparameters |
|---|---|
| BT | epoch $\in \{1, 3, 5, 10\}$, 
 lr $\in \{10^{-1}, 10^{-2}, 10^{-3}, 10^{-4}, 10^{-5}, 5 \times 10^{-5}, 5 \times 10^{-6}, 10^{-6}\}$, 
 temperature scalar $\in \{1, 3, 5\}$. |
| FT | epoch $\in \{5, 10, 15, 20\}$, 
 lr $\in \{10^{-1}, 10^{-2}, 10^{-3}\}$. |
| SSD | dampening constant $\lambda \in \{0.1, 0.5, 0.9, 1\}$, 
 selection weight $\alpha \in \{1, 5, 10, 20, 25, 30, 40, 50, 60, 70, 80, 90, 100\}$. |
| SCRUB | epoch $= 10$, 
 lr $\in \{10^{-1}, 5 \times 10^{-2}, 10^{-2}, 5 \times 10^{-3}, 10^{-3}, 5 \times 10^{-4}, 10^{-4}, 5 \times 10^{-5}, 10^{-5}\}$, 
 max step $\in \{2, 3, 5, 8\}$. |
| SalUn | epoch $\in \{10, 20\}$, 
 lr $\in \{10^{-2}, 10^{-3}, 5 \times 10^{-4}, 5 \times 10^{-5}, 10^{-5}, 5 \times 10^{-6}, 10^{-6}\}$, 
 threshold $\in \{0.2, 0.3, 0.4, 0.5, 0.6, 0.7, 0.8, 0.9\}$. |
| DUCK | lr $\in \{5 \times 10^{-2}, 10^{-2}, 5 \times 10^{-3}, 10^{-3}, 10^{-4}\}$, 
 $\lambda_1, \lambda_2 \in \{0.5, 1, 1.2, 1.5, 2, 3, 5\}$. |
| NG | epoch $\in \{1, 3, 5, 10, 15, 20, 25, 30\}$, 
 lr $\in \{10^{-1}, 10^{-2}, 10^{-3}, 10^{-4}, 10^{-5}\}$. |
| RL | epoch $\in \{1, 3, 5, 10, 15, 20, 25, 30\}$, 
 lr $\in \{10^{-1}, 10^{-2}, 10^{-3}, 10^{-4}, 10^{-5}\}$. |
| JiT | dampening constant $= 1$, 
 lr $\in [10^{-3}, 10^{-6}]$, 
 lipschitz weight $\alpha \in [0, 1]$ . |
| SCAR | lr $\in \{10^{-2}, 5 \times 10^{-3}, 10^{-3}, 5 \times 10^{-4}, 10^{-4}, 5 \times 10^{-5}, 5 \times 10^{-6}, 10^{-6}\}$, 
 batch size $\in \{256, 512, 1024\}$, 
 temperature $\in \{1, 3, 5\}$, 
 $\lambda_1, \lambda_2 \in \{1, 1.5, 3, 5\}$. |
| LoTUS | lr $\in \{10^{-5}, 5 \times 10^{-5}, 10^{-4}, 5 \times 10^{-4}, 10^{-3}, 5 \times 10^{-3}, 10^{-2}, 5 \times 10^{-2}, 10^{-1}\}$, 
 epochs $\in \{5, 10, 15, 20\}$, 
 $\alpha \in \{2, 4, 6, 8, 16\}$. |
| MUNBa | lr $\in \{10^{-5}, 5 \times 10^{-5}, 10^{-4}, 5 \times 10^{-4}, 10^{-3}, 5 \times 10^{-3}, 10^{-2}, 5 \times 10^{-2}, 10^{-1}\}$, 
 epochs $\in \{5, 10, 15, 20, 25, 30, 40\}$. |

# G ADDITIONAL EXPERIMENT RESULTS

## G.1 UNLEARNING UTILITY

### G.1.1 FULL CLASS MU ON PINSFACERECOGNITION

Results of full class unlearning on PinsFaceRecognition dataset are presented in Table A5. MU-Mis exhibits the smallest Avg.Gap to the retrained model, alongside a low MIA susceptibility. SCRUB and Salun exhibit a notably high MIA score, demonstrating a high risk of privacy leakage.

Table A5: Performance overview for **full class** unlearning (including MU-Mis and 6 baselines) evaluated on PinsFaceRecognition with ResNet-18. The content format follows Table 1.

| Method | RA | FA | TA | Avg. Gap ↓ | MIA | RTE |
|---|---|---|---|---|---|---|
| Pretrain | 93.49 | 100 | 93.59 | 34.54 | 100.00 | 13144 |
| Retrain | $93.06_{\pm0.21}$ | $0.00_{\pm0.00}$ | $92.25_{\pm0.32}$ | 0.00 | $0.00_{\pm0.00}$ | 11400 |
| BT | $92.69_{\pm0.01}$ | $0.00_{\pm0.00}$ | $91.48_{\pm0.01}$ | 0.26 | $0.00_{\pm0.00}$ | 36 |
| FT | $93.99_{\pm0.09}$ | $8.78_{\pm1.34}$ | $92.94_{\pm0.10}$ | 3.47 | $0.00_{\pm0.00}$ | 146 |
| SCRUB | $92.82_{\pm0.02}$ | $0.00_{\pm0.00}$ | $91.61_{\pm0.02}$ | 0.29 | $19.63_{\pm0.28}$ | 112 |
| SSD | $93.41_{\pm0.00}$ | $0.00_{\pm0.00}$ | $92.17_{\pm0.00}$ | 0.14 | $0.00_{\pm0.00}$ | 8 |
| DUCK | $92.17_{\pm0.11}$ | $0.00_{\pm0.00}$ | $91.06_{\pm0.11}$ | 0.69 | $0.00_{\pm0.00}$ | 64 |
| SalUn | $93.28_{\pm0.05}$ | $0.62_{\pm0.00}$ | $92.12_{\pm0.06}$ | 0.34 | $54.22_{\pm1.06}$ | 154 |
| MunBa | $91.44_{\pm0.12}$ | $1.63_{\pm0.02}$ | $90.27_{\pm0.11}$ | 1.74 | $0.00_{\pm0.00}$ | 522 |
| LoTus | $92.87_{\pm0.08}$ | $0.00_{\pm0.00}$ | $91.75_{\pm0.09}$ | 0.23 | $0.00_{\pm0.00}$ | 374 |
| MU-Mis | $92.98_{\pm0.06}$ | $0.00_{\pm0.00}$ | $92.13_{\pm0.04}$ | **0.07** | $0.00_{\pm0.00}$ | 24 |

### G.1.2 SUB CLASS MU ON CIFAR-20-LAMP

We evaluate sub-class unlearning on CIFAR-20-Lamp as presented in Table A6. The FA of the retrained model is 11.31, indicating certain generalization capability on the unlearned class. MU-Mis exhibits the smallest Avg.Gap to the retrained model.

Table A6: Performance overview for **sub-class** unlearning (including proposed MU-Mis and 6 baselines) evaluated on lamp of CIFAR-20 using ResNet-18. The content format follows Table 1.

| Method | RA | FA | TA | Avg. Gap ↓ | MIA | RTE |
|---|---|---|---|---|---|---|
| Pretrain | 85.31 | 74.22 | 85.21 | 21.30 | 92.82 | 6910 |
| Retrain | $85.12_{\pm0.22}$ | $11.31_{\pm1.60}$ | $84.40_{\pm0.20}$ | 0.00 | $7.06_{\pm0.11}$ | 4298 |
| BT | $85.52_{\pm0.04}$ | $10.00_{\pm0.00}$ | $84.84_{\pm0.04}$ | 0.72 | $0.00_{\pm0.00}$ | 29 |
| FT | $82.47_{\pm0.15}$ | $14.00_{\pm2.19}$ | $81.90_{\pm0.18}$ | 1.97 | $2.80_{\pm0.36}$ | 128 |
| SCRUB | $82.17_{\pm0.68}$ | $19.00_{\pm4.74}$ | $81.60_{\pm0.70}$ | 4.48 | $26.20_{\pm4.18}$ | 113 |
| SSD | $84.56_{\pm0.00}$ | $15.00_{\pm0.00}$ | $83.84_{\pm0.00}$ | 1.60 | $0.60_{\pm0.00}$ | 18 |
| DUCK | $83.25_{\pm0.31}$ | $31.02_{\pm2.74}$ | $82.69_{\pm0.34}$ | 7.75 | $27.68_{\pm5.15}$ | 68 |
| SalUn | $84.44_{\pm0.05}$ | $13.70_{\pm1.66}$ | $83.74_{\pm0.04}$ | 1.24 | $1.68_{\pm0.17}$ | 1007 |
| MunBa | $81.17_{\pm0.15}$ | $17.00_{\pm0.10}$ | $80.69_{\pm0.14}$ | 4.45 | $5.00_{\pm0.12}$ | 676 |
| LoTus | $27.44_{\pm0.22}$ | $12.00_{\pm0.08}$ | $27.40_{\pm0.20}$ | 38.45 | $33.60_{\pm0.18}$ | 14 |
| MU-Mis | $84.36_{\pm0.51}$ | $11.70_{\pm1.29}$ | $83.66_{\pm0.50}$ | **0.63** | $0.00_{\pm0.00}$ | 10 |

### G.1.3 RANDOM SUBSET UNLEARNING

We evaluate random subset unlearning on CIFAR-10 and SVHN as presented in Table A6. We could see that it is much more challenging for remaining-data-free methods to preserve model utility in this setting, for forgetting and remaining data are highly entangled in this scenario. Across CIFAR-10 and SVHN, MU-Mis exhibits consistently low MIA value, demonstrating a good privacy preservation.

**Comparison with RUM in random subset setting.** RUM(Zhao et al., 2024) provides a careful analysis of what makes unlearning hard and applies tailored unlearning strategies to each group of homogeneous subsets (i.e., by memorization level), achieving excellent performance in especially challenging random-subset setting. Therefore, we follow the experiment of RUM (i.e., Table 1 in its original paper, where random subset is consist of 3000 samples of high, middle, low memorization levels in CIFAR-10.) We compare performances of RUM, SalUn and MU-Mis in the following Table A8. We could see that MU-Mis still outperforms GA, but there is a clear performance gap between MU-Mis and RUM, indicating a room for further improvement.

Table A7: Performance overview for **random subset (10%)** unlearning task evaluated on forgetting 10% CIFAR-10 and SVHN using ResNet-18. The content format follows Table 1.

| Method | CIFAR-10 | | | | | | SVHN | | | | | |
|---|---|---|---|---|---|---|---|---|---|---|---|---|
| | RA | FA | TA | Avg. Gap↓ | MIA | RTE | RA | FA | TA | Avg. Gap↓ | MIA | RTE |
| Pretrain | 100.00 | 99.96 | 94.68 | 1.84 | 92.64 | 14322 | 100.00 | 100.00 | 96.48 | 1.65 | 83.11 | 16904 |
| Retrain | $100.00_{\pm0.00}$ | $94.51_{\pm0.16}$ | $94.75_{\pm0.11}$ | 0.00 | $84.77_{\pm11.13}$ | 12800 | $100.00_{\pm0.00}$ | $95.12_{\pm0.12}$ | $96.40_{\pm0.14}$ | 0.00 | $81.24_{\pm11.13}$ | 13890 |
| BT | $99.64_{\pm0.01}$ | $93.06_{\pm0.06}$ | $93.01_{\pm0.04}$ | 1.18 | $7.20_{\pm0.00}$ | 50 | $98.73_{\pm0.01}$ | $96.71_{\pm0.03}$ | $95.01_{\pm0.01}$ | 1.42 | $22.32_{\pm0.00}$ | 215 |
| FT | $100.0_{\pm0.00}$ | $94.73_{\pm0.13}$ | $93.17_{\pm0.25}$ | 0.51 | $68.66_{\pm0.00}$ | 395 | $100.0_{\pm0.00}$ | $96.38_{\pm0.03}$ | $96.87_{\pm0.09}$ | 0.54 | $80.72_{\pm0.54}$ | 805 |
| SCRUB | $100.00_{\pm0.00}$ | $95.79_{\pm0.50}$ | $93.43_{\pm0.10}$ | 0.87 | $77.56_{\pm0.93}$ | 207 | $95.40_{\pm1.18}$ | $94.79_{\pm1.00}$ | $94.94_{\pm0.83}$ | 1.98 | $23.15_{\pm2.62}$ | 104 |
| SSD | $99.99_{\pm0.00}$ | $99.98_{\pm0.02}$ | $94.66_{\pm0.01}$ | 1.86 | $92.54_{\pm0.00}$ | 18 | $98.67_{\pm0.39}$ | $98.63_{\pm0.43}$ | $96.59_{\pm0.11}$ | 1.52 | $84.23_{\pm0.77}$ | 55 |
| DUCK | $98.04_{\pm0.05}$ | $97.90_{\pm0.21}$ | $92.38_{\pm0.06}$ | 2.57 | $90.59_{\pm0.29}$ | 21 | $96.16_{\pm0.20}$ | $94.86_{\pm0.39}$ | $95.53_{\pm0.25}$ | 1.51 | $30.46_{\pm9.28}$ | 156 |
| SalUn | $100.0_{\pm0.00}$ | $94.31_{\pm0.03}$ | $93.19_{\pm0.03}$ | 0.59 | $26.52_{\pm0.00}$ | 247 | $99.77_{\pm0.00}$ | $94.50_{\pm0.05}$ | $95.85_{\pm0.01}$ | 0.47 | $19.92_{\pm0.00}$ | 409 |
| MunBa | $100.00_{\pm0.09}$ | $94.57_{\pm0.10}$ | $93.19_{\pm0.08}$ | **0.54** | $58.20_{\pm0.18}$ | 420 | $99.73_{\pm0.05}$ | $96.46_{\pm0.06}$ | $96.84_{\pm0.05}$ | 0.68 | $66.70_{\pm0.20}$ | 583 |
| LoTus | $96.90_{\pm0.12}$ | $96.73_{\pm0.08}$ | $90.55_{\pm0.11}$ | 3.17 | $56.10_{\pm0.19}$ | 172 | $94.48_{\pm0.14}$ | $94.31_{\pm0.10}$ | $86.65_{\pm0.13}$ | 5.36 | $78.80_{\pm0.15}$ | 35 |
| NG | $96.46_{\pm0.23}$ | $96.15_{\pm0.35}$ | $90.52_{\pm0.22}$ | 3.13 | $88.21_{\pm0.32}$ | 25 | $98.98_{\pm0.02}$ | $98.98_{\pm0.15}$ | $96.56_{\pm0.01}$ | 1.53 | $84.53_{\pm0.26}$ | 25 |
| RL | $96.17_{\pm0.28}$ | $93.99_{\pm0.51}$ | $88.29_{\pm0.34}$ | 4.27 | $83.11_{\pm0.50}$ | 26 | $98.66_{\pm0.08}$ | $98.53_{\pm0.10}$ | $96.02_{\pm0.02}$ | 1.56 | $73.94_{\pm0.28}$ | 25 |
| JiT | $95.45_{\pm1.92}$ | $95.46_{\pm1.81}$ | $89.63_{\pm1.90}$ | 3.54 | $86.00_{\pm0.32}$ | 255 | $96.30_{\pm0.64}$ | $96.26_{\pm0.74}$ | $95.14_{\pm0.89}$ | 1.88 | $62.78_{\pm24.32}$ | 333 |
| SCAR | $98.63_{\pm0.13}$ | $98.64_{\pm0.08}$ | $92.43_{\pm0.13}$ | 2.61 | $48.36_{\pm0.35}$ | 197 | $96.34_{\pm0.54}$ | $96.46_{\pm0.69}$ | $92.39_{\pm0.70}$ | 2.86 | $55.20_{\pm2.86}$ | 158 |
| MU-Mis | $97.76_{\pm0.03}$ | $97.43_{\pm0.04}$ | $91.50_{\pm0.03}$ | 2.80 | $33.04_{\pm0.28}$ | 116 | $95.48_{\pm0.00}$ | $95.50_{\pm0.03}$ | $94.22_{\pm0.00}$ | 2.36 | $26.11_{\pm0.00}$ | 116 |

**The complementary roles of RUM, SalUn, and MU-Mis.** Nonetheless, it is worth noting that RUM, SalUn, and our MU-MIS focus on different aspects: **RUM** answers **how samples** (low → middle → high memorization) should be scheduled and treated during unlearning, **SalUn** identifies which parameters (gradient-based saliency map) should be updated, and **MU-Mis** figures out **what information** (sample contribution) should be removed. The three aspects are all important and numerically has their own advantages, e.g., RUM is perfect in random subset unlearning and MU-Mis work well without using remaining data. Therefore, RUM, SalUn and MU-Mis are not competing in the same design space, and are naturally complementary and can be combined. And we are very happy to see performance of RUM and especially the combination of RUM and MU-Mis. Combining RUM could improve performance of MU-Mis to some extent, but it still under-perform RUM by a noticeable margin on model utility.

**MU-Mis exhibit lowest KL-divergence on forget set to the retrain model than SalUn and RUM.** Furthermore, we examine the KL divergence between MU-Mis and retrained model in Table A8. Surprisingly, we find that MU-Mis achieves the smallest KL-divergence to the retrain model in forget set across different methods. We attribute this to their fundamentally different unlearning mechanisms: RUM (inherited from fine-tuning and SalUn) relies on random relabeling or catastrophic forgetting. As discussed in our paper, random relabeling might not provide a principled way to align the output distribution of the retrain model on forgetting data, leading to a sub-optimal output distribution. Also, compared with RL, MU-Mis does improve existing RDF method in random subset unlearning. The overall KL divergence of RL is 4.3, while that of MU-Mis is 0.65, which is an substantial improvement, indicating that MU-Mis produces an output distribution much closer than other RDF methods. Consequently, although MU-Mis still underperforms remaining-data-dependent methods such as RUM+SalUn in terms of model utility, we think its lower KL divergence on the forgetting data is a meaningful step that advances unlearning towards a more faithful/reasonable unlearning.

**Challenges of remainig-data-free methods in random subset removal.** As is investigated in Figure A4 in Appendix H, we could see that intra-class samples assemble highly similar gradients. Moreover, many existing remaining-data-dependent approaches are still suffering from utility degradation in this setting. Therefore, it is important to acknowledge the daunting challenges such a vision of remaining-data-free method in random subset setting faces. Although not perfect, but MU-Mis has advanced remaining-data-free method in this challenging scenario. Therefore, we remain hopeful about the vision of a perfectly RDF method that preserves model utility under random subset settings and we believe a better location of sample contribution might offer a promising path.

Table A8: Unlearning performance and KL divergence metrics of SalUn, RUM and MU-Mis on CIFAR-10 with ResNet-18 following Zhao et al. (2024).

| Method | FA | RA | TA | Avg. Gap↓ | ToW | MIA | MIA.GAP | ToW_MIA | KL_Forget | KL_Test | KL_Retain | KL_All |
|---|---|---|---|---|---|---|---|---|---|---|---|---|
| Pretrain | 100.00 | 100.00 | 85.10 | 12.24 | 0.64 | 0.03 | 0.45 | 0.44 | 2.7131 | 0.3040 | 0.0018 | 0.1826 |
| Retrain | $63.93_{\pm0.15}$ | $100.00_{\pm0.00}$ | $84.45_{\pm0.11}$ | 0.00 | $1.00_{\pm0.07}$ | $0.47_{\pm0.05}$ | 0.00 | $1.00_{\pm0.08}$ | 0.0000 | 0.0000 | 0.0000 | 0.0000 |
| SalUn | $73.63_{\pm0.21}$ | $99.99_{\pm0.01}$ | $81.64_{\pm0.18}$ | 4.17 | $0.88_{\pm0.06}$ | $0.79_{\pm0.04}$ | 0.31 | $0.64_{\pm0.07}$ | 0.9941 | 0.3601 | 0.0548 | 0.1175 |
| RUM | $66.40_{\pm0.14}$ | $100.00_{\pm0.00}$ | $84.41_{\pm0.13}$ | **0.84** | **$0.97_{\pm0.05}$** | $0.99_{\pm0.03}$ | 0.52 | $0.48_{\pm0.09}$ | 1.4462 | **0.2770** | **0.0057** | **0.0675** |
| RL | $58.40_{\pm0.24}$ | $62.32_{\pm0.20}$ | $47.81_{\pm0.15}$ | 26.62 | $0.37_{\pm0.04}$ | $0.49_{\pm0.08}$ | 0.02 | $0.34_{\pm0.06}$ | 5.9854 | 4.5657 | 4.1801 | 4.3005 |
| MU-Mis | $66.90_{\pm0.19}$ | $70.48_{\pm0.22}$ | $57.37_{\pm0.16}$ | 19.85 | $0.50_{\pm0.03}$ | $0.42_{\pm0.06}$ | 0.06 | $0.40_{\pm0.04}$ | **0.6525** | 1.1091 | 1.2391 | 0.7509 |
| RUM+MU-Mis | $60.70_{\pm0.23}$ | $77.45_{\pm0.19}$ | $61.18_{\pm0.14}$ | 16.35 | $0.58_{\pm0.08}$ | $0.41_{\pm0.07}$ | 0.06 | $0.56_{\pm0.05}$ | 2.7765 | 1.0060 | 0.6421 | 0.7005 |

### G.1.4 FULL CLASS MU COMPARED WITH JIT WITH VGG16

JiT unlearns by regularizing lipshitz constant around the forgetting data, which might fail on DNNs with batch norm layers. Following the architecture used in the original paper, we compare with it on CIFAR-100-rocket unlearning in Table A9. The results show that JiT exhibits greater efficacy when implemented with VGG-16 as opposed to ResNet-18. However, it still exhibits a noticeable performance disparity when compared to MU-Mis.

Table A9: Performance comparison between MU-Mis and JiT of **full class** unlearning on CIFAR-100 using **VGG-16**.

| Metrics | RA | FA | TA | Avg. Gap↓ | MIA | RTE |
|---|---|---|---|---|---|---|
| Pretrain | 64.72 | 66.75 | 64.76 | 23.64 | 85.20 | - |
| Retrain | $64.96_{\pm0.35}$ | $0.00_{\pm0.00}$ | $63.36_{\pm0.34}$ | 0.00 | $15.7_{\pm1.56}$ | - |
| JiT | $60.98_{\pm0.08}$ | $3.73_{\pm0.00}$ | $60.45_{\pm0.07}$ | 3.21 | $8.02_{\pm0.24}$ | 16 |
| MU-Mis | $64.60_{\pm0.04}$ | $0.00_{\pm0.28}$ | $63.96_{\pm0.04}$ | **0.32** | $3.28_{\pm0.22}$ | 35 |

### G.1.5 SUB-CLASS MU ON CIFAR-20-SEA WITH VIT

We conducted experiments with ViT on subclass-CIFAR20-sea to further demonstrate our effectiveness across different tasks. MU-Mis exhibits the best RA, FA, and TA, and provides advantages in unlearning time.

Table A10: Performance overview for **sub-class** unlearning task evaluated on **Cifar20-Sea** using **ViT**. RTE is reported in **minute**.

| Methods | RA | FA | TA | Avg. Gap ↓ | MIA | RTE (min) |
|---|---|---|---|---|---|---|
| Pretrain | 93.65 | 91.32 | 93.63 | 0.93 | 69.80 | - |
| Retrain | $93.90_{\pm0.12}$ | $88.98_{\pm0.08}$ | $93.84_{\pm0.14}$ | 0.00 | $59.00_{\pm0.23}$ | - |
| SalUn | $94.15_{\pm0.10}$ | $89.29_{\pm0.03}$ | $94.13_{\pm0.08}$ | 0.27 | $62.10_{\pm0.42}$ | 10 |
| MU-Mis | $93.70_{\pm0.02}$ | $88.84_{\pm0.25}$ | $93.67_{\pm0.02}$ | **0.17** | $69.67_{\pm0.31}$ | 0.5 |

### G.2 MEMBERSHIP INFERENCE ATTACK

We examine our privacy leakage with a comparably good unlearning-adapted LiRA proposed by SCRUB Kurmanji et al. (2023). We report SCRUB-LiRA examined results across 3 unlearning settings in Table A11. We could see that in MU-Mis unlearned model, SCRUB-LiRA is quite close to random guessing, indicating that the forgetting data in MU-Mis unlearned model is similar to non-members.

**LiRA-SCRUB and MIA-Entropy is complementary.** The goal of the adversary in LiRA-based MIA is to distinguish "forgotten samples" from unseen (non-member) samples. Therefore, their criteria of successful unlearning is that in the unlearned model, the adversary could not well distinguish forgotten samples from unseen samples, i.e., MIA collapse to random guessing (50%) for random subset unlearning. LiRA-based MIA examines how indistinguishable the forgetting data to the non-members, thereby the closer of SCRUB-MIA value to 50% in random subset setting is the better. While MIA in our paper examines how much forgotten samples are predicted as members, thereby the lower is the better. That is to say, **LiRA-based MIA** examines the extent of **non-membership** of forgetting data, while our **MIA-Entropy** examines the extent of membership, *i.e.*, the residual **membership** signals of forgetting data. Therefore, these 2 MIAs are not contradictory or competing, but complementary, which all support our conclusions.

Table A11: SCRUB-LiRA scores across different unlearning settings.

| Setting | Model | FA (%) | RA (%) | TA (%) | MIA-Entropy | MIA-SCRUB |
|---|---|---|---|---|---|---|
| **Fullclass-CIFAR100-Rocket** | Pretrain | 86.00 | 76.37 | 76.56 | 95.40 | 73.40±2.60 |
| | Retrain | 0.00 | 76.86 | 76.07 | 6.60 | 96.40±1.60 |
| | MU-Mis | 0.00 | 76.37 | 75.71 | 0.00 | 90.40±2.00 |
| **Subclass-CIFAR20-Sea** | Pretrain | 97.00 | 85.10 | 85.14 | 91.80 | 64.40±2.40 |
| | Retrain | 80.00 | 84.85 | 85.00 | 53.00 | 47.40±6.40 |
| | MU-Mis | 80.00 | 84.26 | 84.15 | 0.60 | 53.00±4.60 |
| **Random-CIFAR10-10%** | Pretrain | 99.96 | 100.00 | 94.68 | 92.64 | 56.64±0.60 |
| | Retrain | 94.51 | 100.00 | 94.75 | 84.77 | 49.68±0.30 |
| | MU-Mis | 97.43 | 97.42 | 91.50 | 83.34 | 52.66±0.40 |

### G.3 ABLATION STUDY

#### G.3.1 ABLATION STUDY ON EACH TERM OF MU-MIS

We study the role of each term in our loss through ablation, illustrating with fullclass-CIFAR100-Rocket on ResNet-18. We denote the first term $\|\nabla_x f_c(w, x)\|_F^2$ in our loss Eq. (3) as TC (Target Class) and the second term $\|\nabla_x f_{c'}(w, x)\|_F^2$ as OC (Other Class). In Table A12, we showcase the unlearning performance of decreasing TC, increasing OC, and decreasing

Table A12: Ablation study on each term of our loss in full class (Rocket) unlearning. TC (Target Class) refers to the first term and OC (Other Class) refers to the second term.

| Methods | RA | FA | TA | Avg. Gap ↓ | MIA | RTE |
|---|---|---|---|---|---|---|
| Pretrain | 76.41 | 79.69 | 76.47 | 26.84 | 95.80 | - |
| Retrain | $76.52_{\pm0.27}$ | $0.00_{\pm0.00}$ | $75.76_{\pm0.24}$ | 0.00 | $2.87_{\pm0.46}$ | - |
| TC | $65.41_{\pm0.00}$ | $0.00_{\pm0.00}$ | $64.73_{\pm0.00}$ | 7.38 | $1.40_{\pm0.00}$ | 44 |
| OC | $70.13_{\pm0.41}$ | $74.62_{\pm1.56}$ | $70.21_{\pm0.42}$ | 28.85 | $0.00_{\pm0.00}$ | 25 |
| TC-OC.detach() | $70.24_{\pm0.01}$ | $17.35_{\pm0.04}$ | $69.45_{\pm0.03}$ | 9.98 | $1.60_{\pm0.045}$ | |
| TC - OC | $76.42_{\pm0.07}$ | $0.00_{\pm0.00}$ | $75.64_{\pm0.07}$ | **0.07** | $0.00_{\pm0.00}$ | 30 |

TC - OC (MU-Mis) respectively. We also investigate another variant of MU-Mis: regressing sensitivity norm of TC(target class) term to OC(other class) term. During unlearning, we observe that as $\|\nabla_x f_c\|_F$ decreases, FA decreases gradually. We find that although the gradient of $\|\nabla_x f_{c'}\|_F$ is detached, its norm exhibits slight decrease as well when minimizing. As the sensitivity norm of $f_c$ approaches $f_{c'}$, i.e., the loss approaches 0, FA gradually stops to decrease and converges at 17.35%. Therefore, this variant could unlearn successfully but failed to preserve model utility by regressing sensitivity magnitude of fc to fc'. As demonstrated by Figure A1, when minimizing $\|\nabla_x f_c\|_F$, $\|\nabla_x f_{c'}\|_F$ and retain accuracy (RA) will drop slightly as well, then RA will recover to its original level as the picking up of $\|\nabla_x f_{c'}\|_F$ in our loss. Therefore, the magnitude of $\|\nabla_x f_{c'}\|_F$ is essential for preserving model utility on remaining data (which is also supported by the ablation study in our Table A10). Thus, when regressing $\|\nabla_x f_c\|_F$ to $\|\nabla_x f_{c'}\|_F$, the norm of $\|\nabla_x f_{c'}\|_F$ is not explicitly preserved, and drops during minimizing. Therefore, this regressing variant is a useful diagnose and it confirms that the joint-term optimization in MU-Mis is necessary.

From the ablation study, we know that minimizing the first term obliviates information of the forgetting data, but greatly hurts performance on the remaining data. Solely increasing OC brings a slight accuracy drop on remaining data, but hardly unlearns the forgetting data. While our MU-Mis loss could unlearn effectively meanwhile preserve model utility on the remaining data.

#### G.3.2 PERFORMANCE UNDER SENSITIVITY REGULARIZATION TECHNIQUE

We employed Jacobian regularization Hoffman et al. (2019) to our pre-training and re-training. Jacobian regularization penalizes model's input-output sensitivity around training sample x, i.e., $\|J(x)\| = \|\frac{\partial f(x)}{\partial x}\|_F$ to encourage a smoother decision boundary for better robustness and generalization. We evaluated the unlearning of an full class on the CIFAR-10 dataset by training ResNet18 for 30 epochs with lr = 0.1, bs = 256, regularization coefficient =0.01. The unlearning performance is shown in Table A14, where the first 3 rows are performances on vanilla model (with data augmentation but without Jacobian regularization). In Jacobian regularized model, it seems to be more difficult to fully forget a class by MU-Mis, but it still preserves model utility well without remaining data.

#### G.3.3 ROBUSTNESS OF PERFORMANCE TO SENSITIVITY METRIC.

We investigate performances of MU-Mis with different sensitivity norm choices in Table A13. We can see that the $L_2$ norm yields similar performance to the Frobenius norm (original MU-Mis) and retrain, suggesting that the method is robust as long as the norm aggregates all components of the sensitivity vector. In contrast, norms that emphasize only extreme

Table A13: Performances of MU-Mis under different sensitivity norm choices.

| Methods | RA | FA | TA | Avg. Gap ↓ | MIA |
|---|---|---|---|---|---|
| Pretrain | 76.41 | 79.69 | 76.47 | 26.84 | 95.80 |
| Retrain | $76.52_{\pm0.27}$ | $0.00_{\pm0.00}$ | $75.76_{\pm0.24}$ | 0.00 | $2.87_{\pm0.46}$ |
| L2 | $75.53_{\pm0.05}$ | $1.95_{\pm0.02}$ | $74.89_{\pm0.11}$ | 1.27 | $0.000_{\pm0.00}$ |
| Inf | $50.34_{\pm0.12}$ | $1.95_{\pm0.04}$ | $50.11_{\pm0.14}$ | 17.93 | $0.710_{\pm0.02}$ |
| Frobenius | $76.42_{\pm0.07}$ | $0.00_{\pm0.00}$ | $75.64_{\pm0.07}$ | **0.07** | $0.000_{\pm0.00}$ |

coordinates (e.g., infinity norm) tend to over-penalize the worst-case direction, which empirically hurts model utility. We observed that the model tends to break down even under a very small learning rate (1e-8) for $L_1$ norm, which we attribute this to its non-smooth optimization, which induces abrupt and sparse updates in model outputs.

Table A14: Unlearning performance w/o Jacobian regularization on CIFAR-10-fullclass-0.

| Method | No Jacob Regularization | | | | | Jacob Regularization = 0.01 | | | | |
|--------|-----|-----|-----|-----------|-----|-----|-----|-----|-----------|-----|
| | FA | RA | TA | Avg. Gap↓ | MIA | FA | RA | TA | Avg. Gap↓ | MIA |
| Pretrain | 88.54 | 83.59 | 84.03 | 32.54 | 94.98 | 88.53 | 83.33 | 83.11 | 32.58 | 94.36 |
| Retrain | $0.00_{\pm0.00}$ | $86.11_{\pm0.11}$ | $77.56_{\pm0.15}$ | 0.00 | $9.56_{\pm0.08}$ | $0.00_{\pm0.00}$ | $85.77_{\pm0.23}$ | $77.35_{\pm0.07}$ | 0.00 | $0.11_{\pm0.09}$ |
| MU-Mis | $1.40_{\pm0.35}$ | $84.13_{\pm0.28}$ | $75.98_{\pm0.52}$ | 1.62 | $0.98_{\pm0.12}$ | $5.67_{\pm0.32}$ | $82.37_{\pm0.14}$ | $75.12_{\pm0.07}$ | 3.77 | $0.21_{\pm0.09}$ |

## G.4 HYPER-PARAMETER SENSITIVITY

We show that there is a clear relationship between the norm ratio and model performance during unlearning in Figure A1. The stopping threshold ratio $\delta$ controls the magnitude ratio of final irrelevant class sensitivity norm $\|\nabla_x f_{c'}(x, w)\|_F$ to initial one $\|\nabla_x f_{c'}(x, w_p)\|_F$. In this part, we investigate the sensitivity of unlearning performance to $\delta$ in Figure A2, taking fullclass-CIFAR100-Rocket with ResNet-18 as an example. Different colors represent different $\delta$, ranging from 70% to 100%. The green dashed line indicates the performance of the retrained model. Notably, under a specific learning rate, the Avg. Gap exhibits minimal variation with changes in $\delta$. Generally, MU-Mis demonstrates resilience against variations in hyper-parameters $\delta$. In summary, the hyper-parameter tuning for MU-Mis is effortless, resilient and well-guided. This advantage in hyper-tuning indirectly enhances the efficiency of unlearning and facilitates straightforward application of MU-Mis, making it valuable for practical applications of MU methods.

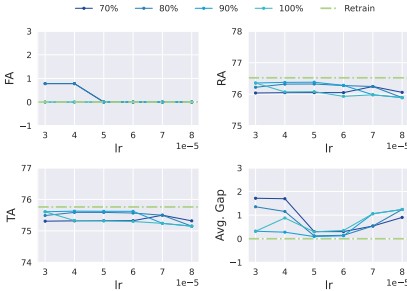

Figure A2: MU-Mis unlearning performance with different stopping threshold ratio under different learning rates on fullclass-CIFAR100-rocket with ResNet-18. Performance varies little with the threshold ratio when the learning rate is fixed.

## G.5 EFFECTIVENESS VISUALIZATION BY ATTENTION MAP

To further investigate and understand the behavior of our unlearned model, we showcase the attention heatmaps (Selvaraju et al., 2017) of models before and after applying MU-Mis on PinsFaceRecognition dataset in Figure A3. For the forgetting data, the original attention concentrates on the faces. After applying MU-Mis, the attention on the faces either disappears or significantly weakens, and is shifted towards the background. For the remaining data, MU-Mis fully maintains previous attention. Notably, an alternative interpretation of input sensitivity is the measurement of how changes in the image influence its model prediction (Smilkov et al., 2017). Our proposed method reduces the target class logit sensitivity to the forgetting data while recovering irrelevant classes', thereby enabling the unlearned model to disregard the semantic information in the forgetting data meanwhile preserve prediction sensitivity and performance on the remaining data.

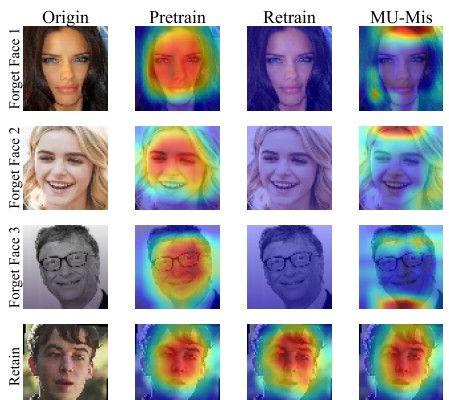

Figure A3: **Visualization of attention maps for the full class unlearning task on PinsFaceRecognition.** MU-Mis distracts attention from forgetting data regions while preserving attention on remaining data.

# H UNDERLYING REASONS FOR MODEL UTILITY PRESERVATION

We suggest the favorable preservation on model utility might be attributed to two factors:

- **Conceptual motivations**: The forgetting operation in MU-Mis differs fundamentally from previous methods. While previous methods involved relabelling the forgetting data (Liu et al., 2024; Foster et al., 2024a; Fan et al., 2024b) or knowledge distillation from a useless teacher (Chundawat et al., 2023; Lin et al., 2023; Kurmanji et al., 2023), which unlearn by introducing incorrect information to spoil original knowledge, MU-Mis unlearns by solely withdrawing the contribution of forgetting data. In Figure A1, we show that RA and TA stay at a high level throughout the withdrawal process.

- **Empirical investigation**: Current gradient-based unlearning methods (Liu et al., 2024; Foster et al., 2024a; Fan et al., 2024b; Wu et al., 2020; Graves et al., 2021; Neel et al., 2021; Thudi et al., 2022) all employ cross-entropy loss gradient $\nabla_\theta \mathcal{L}(w, x)$ to unlearn. However, the intra-class gradients in a well-trained model are quite similar (Papyan, 2020; Papyan et al., 2020). While the gradient of input sensitivity norm w.r.t parameters is double back-propagation (Drucker & Le Cun, 1992), enhancing sample-specificity by first back-propagating to the input samples before reaching the parameters. Our pairwise analysis of cosine similarity among intra-class and inter-class samples, detailed in the following, reveals a distinctive orthogonality in input sensitivity gradients, spontaneously reducing the interference between the forgetting and remaining data.

We calculate the pairwise cosine similarity within a class and between classes of the derivatives of four metrics w.r.t. parameters in a well-trained model in Figure A4 to demonstrate an inherent orthogonality input sensitivity view. They are $\nabla_w \mathcal{L}$, $\nabla_w f_c$, $\nabla_w \|\nabla_x f_c\|_F$, and $\nabla_w \sum_{c' \neq c} \|\nabla_x f_{c'}\|_F$ (denoted as $\nabla_w \|\nabla_x f_{c'}\|_F$ in the following for brevity). The first two metrics are commonly used in current unlearning methods, and the last two are utilized in MU-Mis. Derivatives of all four metrics are approximately orthogonal between samples from different classes. However, intra-class similarities differ across four metrics. We can see that both $\nabla_w \mathcal{L}$ and $\nabla_w f_c$ bear a resemblance within a class, with cosine similarity centering around 0.3 and reaching up to 0.6. While $\nabla_w \|\nabla_x f_c\|_F$ centers around 0.1 with intra-class similarity scarcely exceeding 0.3. **Directly taking the derivative of output w.r.t parameters preserves within-class similarity of the output, but such similarity is reduced when output first takes the derivative to the input and then back to parameters.**

Interestingly, $\nabla_w \sum_{c' \neq c} \|\nabla_x f_{c'}\|_F$ seems to be pairwise similar. We speculate that there are two reasons why this portion of our loss function does not significantly harm the performance of the remaining data. Firstly, as shown in Figure 3, $\sum_{c' \neq c} \|\nabla_x f_{c'}\|_F$ is significantly smaller than $\|\nabla_x f_c\|_F$ on well-trained models. Therefore, in the early stages of optimizing the relative magnitudes of input sensitivities, the contribution of $\nabla_w \sum_{c' \neq c} \|\nabla_x f_{c'}\|_F$ to the optimization direction can be neglected. Secondly, $\sum_{c' \neq c} \|\nabla_x f_{c'}\|_F$ for the forgetting data actually corresponds to the target class of the remaining data. The inter-class similarity of $\nabla_w \sum_{c' \neq c} \|\nabla_x f_{c'}\|_F$ implies that when sensitivity of other irrelevant classes increases, the input sensitivity of the remaining data of the corresponding classes increases as well, thereby preserving their sample contributions.

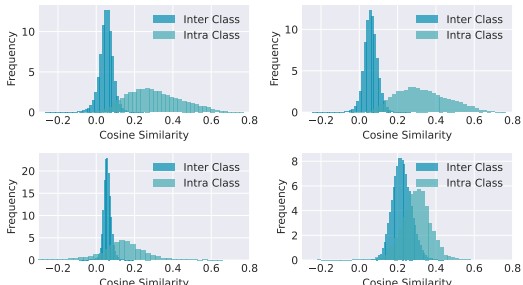

Figure A4: **Inter-class and intra-class cosine similarity of four different metrics w.r.t parameters.** From left to right: $\nabla_w \mathcal{L}$, $\nabla_w f_c$, $\nabla_w \|\nabla_x f_c\|_F$, and $\nabla_w \|\nabla_x f_{c'}\|_F$. Directly taking the derivative of output w.r.t parameters bears a resemblance across samples, but such similarity is reduced when output first takes derivative to input and then back to parameters.

## I DISCUSSION

Beyond the advantages of input sensitivity for measuring sample contribution, we raise the attention that the input sensitivity perspective possesses a profound connection to MU. MU emerges from model's memorization of the training data and seeks to safeguard the privacy of training data. Recent studies by Garg et al. (Garg et al., 2024) and Ravikumar et al. (Ravikumar et al., 2024) reveal the intrinsic relationship between memorization, privacy and sample's input curvature. Also, Mo et al. (Mo et al., 2021) demonstrates that the input sensitivity of model gradient is the underlying cause of information leakage exposed by Model Inversion Attack (MIA) (Zhu et al., 2019; Zhao et al., 2020). Collectively, these works further underscore the inherent advantages of adopting an input sensitivity perspective for machine unlearning.

### I.1 CORRELATIONS BETWEEN SENSTIVITY GAP AND LOSS CURVATURE IN FORMULA

The loss curvature for memorization proxy used in Zhao et al. (2024); Zhao & Triantafillou (2024) is $\text{Curvature}(x) \propto \text{tr}\left(\nabla_x^2 \mathcal{L}(x)\right)$. For input sensitivity of loss, we have

$$\nabla_x \mathcal{L}(x) = (p - e_c)^\top \nabla_x f(x) = (1 - p_c)\nabla_x f_c(x) - \sum_{c' \neq c} p_{c'} \nabla_x f_{c'}(x)$$

$$= (\sum_{c'=1}^{C} p_{c'})\nabla_x f_c(x) - \sum_{c'=1}^{C} p_{c'}\nabla_x f_{c'}(x) = \sum_{c=1}^{C} p_{c'}\left(\nabla_x f_c(x) - \nabla_x f_{c'}(x)\right). \tag{A6}$$

By comparing the formula of loss curvature and sensitivity gap used in MU-Mis, we could see that:

$$\text{tr}\left(\nabla_x^2 \mathcal{L}(x)\right) = \sum_{c=1}^{C} p_{c'} \text{tr}\left(\nabla_x^2 \left(f_c(x) - f_{c'}(x)\right)\right) \text{v.s.} \|\nabla_x f_c(x)\|_F^2 - \|\nabla_x f_{c'}(x)\|_F^2 \tag{A7}$$

While both terms reflects sensitivity gap between target class and irrelevant classes, they refers to different order of gradient. Specifically, **loss curvature** primarily aims at **second-order** sensitivity, while sensitivity gap in **MU-Mis** refers to the **first-order** sensitivity. Mathematically, it seems that there is no straightforward equality or conserved bound that could directly links this two metrics.

### I.2 INPUT SENSITIVITY SIGNATURES ACROSS DIFFERENT SAMPLES

Although there is few clue of mathematical relationship for analysis, we further investigate their correlations by examining the sensitivity signatures across different samples empirically, *i.e*, samples of different memorization/influence levels.

We partition training samples of CIFAR-10 dataset according to their sample-wise memorization score (provided by Feldman (2020); Feldman & Zhang (2020) through training many models on different held-out subsets to measure each sample's self-influence on its own prediction) into low, middle, high memorization levels (following Zhao et al. (2024)). We partition training samples of CIFAR-100 dataset according to their influence score (provided by Feldman (2020); Feldman & Zhang (2020) through training many models on different held-out subsets to quantify each sample's cross-influence on test data) into the same 3 levels. The scores are available at https://github.com/google-research/heldout-influence-estimation.

Interstingly, the distributions of sensitivity gap of different sample groups are shown in Table A15. For memorization level, highly memorized samples exhibit smaller sensitivity gap than low and middle memorized sample. For influence score, more influencial samples exhibit higher sensitivity gap. Importantly, we think this empirical findings provide meaningful support for our use of sensitivity gap as a proxy for sample contribution.

We evaluate MU-Mis when respectively unlearning samples with low, medium, and high memorization levels in Table A16. We could see that removing those highly-memorized samples causes a more substantial utility drop in the remaining data, indicating that the performance of MU-Mis is not uniform across samples of different memorization levels. But maybe this is understandable, to some extent, expected, as unlearning highly entangled and influential samples is intrinsically difficult for

Table A15: Sensitivity gaps across memorization and influence levels on CIFAR-10 and CIFAR-100.

| Level | CIFAR-10 (Memorization Level) | | | | | CIFAR-100 (Influence Level) | | | | |
|-------|------|------|------|------|------|------|------|------|------|------|
| | Mean | Std | 10th | 50th | 90th | Mean | Std | 10th | 50th | 90th |
| Low | 10.7632 | 4.2386 | 6.0814 | 10.0681 | 16.4851 | 36.1209 | 14.7241 | 19.3663 | 34.3506 | 56.0993 |
| Mid | 10.5156 | 5.5244 | 3.9959 | 10.0496 | 17.6092 | 39.3604 | 16.7700 | 21.3345 | 36.4983 | 61.0855 |
| High | 6.7922 | 6.0817 | -0.7257 | 6.6459 | 14.5839 | 42.5803 | 17.1963 | 23.9295 | 39.5377 | 65.7531 |

any unlearning method Fan et al. (2024a); Zhao et al. (2024). We acknowledge this is an important limitation and a promising direction for future work, where more fine-grained unlearning mechanisms could be developed to further improve remaining-data-free unlearning.

Table A16: Performance of MU-Mis when unlearning low, middle, high levels of samples.

| Memorization | Method | FA | RA | TA | Avg. Gap | MIA |
|--------------|--------|------|------|------|----------|------|
| | Original | 100.00 | 100.00 | 85.10 | 0.45 | 0.013 |
| Low | Retrain | 99.83 | 100.00 | 83.93 | 0.00 | 0.049 |
| | MU-Mis | 100.00 | 99.96 | 83.32 | 0.27 | 0.041 |
| | Original | 100.00 | 100.00 | 85.10 | 9.02 | 0.019 |
| Mid | Retrain | 74.40 | 100.00 | 83.63 | 0.00 | 0.539 |
| | MU-Mis | 93.63 | 87.09 | 69.29 | 15.49 | 0.194 |
| | Original | 100.00 | 100.00 | 85.10 | 26.83 | 0.055 |
| High | Retrain | 21.63 | 100.00 | 82.99 | 0.00 | 0.811 |
| | MU-Mis | 46.10 | 66.79 | 57.03 | 27.88 | 0.607 |

### I.3 ESSENTIAL GOAL OF MU IN TERMS OF MEMORIZATION, GENERALIZATION AND SAMPLE CONTRIBUTION

There is an essential question underlying machine unlearning (MU), which might also connect core idea of MU-Mis and RUM :

**(Q1)** What is the correlation between sample contribution and memorization?

**(Q2)** Is unlearning equivalent to alleviating a sample's memorization level?

**Terminology and Conceptual Distinctions.** To clarify the discussion, we first distinguish three closely related but conceptually different notions:

1. **Memorization** is defined as the change in its own prediction when a sample leaves the training set, i.e., self-influence.

2. **Sample influence** is the change on prediction of other data (test data), thereby more lies in cross-influence.

3. **Sample contribution** is the contribution of a training sample to all the model predictions, thereby comprising both the memorization (self-influence) and sample influence (cross-influence).

**RQ1. Memorization level is not proportional to sample contribution.** On the one hand, high memorization can coincide with large contribution, *e.g.*, long-tail but but genuinely informative examples, the model may need to "memorize" them to support generalization. On the other hand, high memorization does not guarantee substantial contribution: a model might over-fit noise, duplicates, or outliers, thereby exhibiting strong memory for those examples even though they contribute little or may harm the remaining data's performance. Conversely, a training example may exert substantial influence on the model's remaining predictions (high contribution) without having been deeply memorized (low memorization).

**RQ2. Minimizing sample contribution is more essential for unlearning than reducing memorization.** Generally, in random subset case, to unlearn an instance, we would aim to alleviate model's memorization of it to prevent privacy leakage exploited by MIA. However, as discussed above, memorization (self-influence) and influence (cross-influence) consist sample contribution together. Therefore, in a broader sense, de-memorization is not that enough for a complete removal, while withdrawing sample contribution is more fundamental.

In light of the effectiveness of MU-Mis in full/sub-class unlearning, where removal is beyond merely reducing memorization, we think MU-Mis is not limited to directly alleviate memorization by optimizing certain curvature-based measures. Rather, we prefer to view MU-Mis as a practical and efficient proxy that could implicitly reduce memorization by suppressing a sample's contribution. Additionally, given the less satisfactory results of MU-Mis in the random-subset unlearning setting, we suspect that a direct optimization of loss curvature (*i.e.*, explicitly minimizing loss curvature around a speicific sample) might be sufficient to prevent privacy leakage and yield better model-utility trade-offs in this scenario.

## J   LIMITATION

Although MU-Mis could achieve comparable performance to SoTA remaining-data-dependent methods, there is a clear room for further improvement in the most challenging unlearning scenario, random subset unlearning. We fully acknowledge that our investigation is quite preliminary, but we believe that the input sensitivity might be a valuable and beneficial perspective for developing remaining-data-free unlearning, which is collectively demonstrated by MU-Mis and RUM, leaving a good starting point for future study.

## K   BROADER IMPACT

From a practical perspective, remaining-data-free unlearning is particularly valuable in real-world scenarios where access to the retained training data is restricted or infeasible. In many industrial systems, data may be distributed across silos, subject to contractual deletion, encrypted under privacy regulations, or too large to reload for repeated maintenance. In such settings, existing unlearning methods that require remaining data for utility recovery impose substantial computational and storage costs, undermining efficiency and scalability. By directly suppressing sample contribution within the pre-trained model, our approach enables effective unlearning without retraining or additional data access, significantly reducing operational overhead and enabling deployment in resource-constrained or compliance-sensitive environments.

Beyond privacy benefits, the key impact of our remaining-data-free design is a qualitative improvement in efficiency. By removing the need to relearn on retained data, our method reduces time overhead compared with existing unlearning approaches, which could make unlearning substantially faster than retraining in practice and enabling responsive deletion in real-world systems.

