# OpenReview forum: "Remaining-data-free Machine Unlearning by Suppressing Sample Contribution"
_ICLR.cc/2026/Conference — ICLR 2026 Poster_

### Official Review · Reviewer_6Bdx · 2025-10-27

**Soundness:** 3
**Presentation:** 2
**Contribution:** 2
**Rating:** 4
**Confidence:** 3

**Summary:**

Overall, this paper focuses on machine unlearning without access to retain set. This paper essentially relies on an observation that, as each sample is used during training, the learned model increases in sensitivity towards it. Specifically, during training of a sample, the difference of gradients between the relevant and irrelevant classes increases. This observation is used to design a loss to suppress the sample contributions, thus achieving unlearning.

**Strengths:**

The paper addresses an important area of machine unlearning which does not use any data from the retain set.

The paper seems sound in the theoretical areas.

Experimental results in Table 1 and Table 2 look convincing.

**Weaknesses:**

I think the intuition of the method is not well expressed. Although the math and equations are there, the intuition is not clear, in the sense that, it should be able to be expressed in a few lines at most. However, the paper does not express these simple concepts well.

In terms of the proposed method, I believe the idea is quite simple. Basically, the sensitivity of each forget sample is computed, with respect to the relevant class and also irrelevant classes. Then, the loss is designed to suppress this difference, thus making the model learn to forget, similar to how a unlearned model would be. However, these concepts have not been explained clearly in the paper.


Importantly, the full related work section has been placed in the Supplementary. I do not feel good about this, since I do not think it is fair for most other works that use space to discuss it in the main paper. In some sense, since reviewers are not obligated to read the supplementary, it can be said to some extent that this paper does not have a proper related work section. All in all, since it is a main and important component of the paper, there should be substantial discussion on related work in the main paper. This is a huge issue for me.

I also feel uncomfortable that the implementation details section is fully in the supplementary.

Section 3.2 which explains the input sensitivity is too long in my opinion. Also, the derivations and math look very similar to influence functions (some example papers shown below). In this long section 3.2, I am unable to grasp the novelty and contributions of these theoretical analyses, how they are different from previous insights, and why they are important.

Efficient Machine Unlearning via Influence Approximation. arXiv 2025

Reliable Active Learning via Influence Functions. TMLR 2023

Influence Selection for Active Learning. ICCV 2021


There is a citation error for two references:

Towards Unbounded Machine Unlearning is a NeurIPS 2023 paper, not 2024.

Is retain set all you need in machine unlearning? Restoring performance of unlearned models with out-of-distribution images is an ECCV 2024 paper, not 2025

The authors can consider reporting results and/or comparing with more recent works such as:

Learning to Unlearn for Robust Machine Unlearning. ECCV 2024

Adversarial Machine Unlearning. ICLR 2025

Decoupled Distillation to Erase: A General Unlearning Method for Any Class-centric Tasks. CVPR 2025

LoTUS: Large-Scale Machine Unlearning with a Taste of Uncertainty. CVPR 2025

MUNBa: Machine Unlearning via Nash Bargaining. ICCV 2025




Minor errors that do not affect the score: Line 105: knowledge -> knowledge

**Questions:**

Please refer to Weaknesses for most questions.

How would the model be applied in cases without classification tasks? E.g., forgetting a single name or data entry.

The method aims to minimize the sensitivity difference between relevant and irrelevant classes. Just curious, what would happen if the gradients for the irrelevant classes are fixed, i.e., without backpropagating gradients, and only the gradient for the relevant class is used? I guess, in some sense, it would tell us if the loss is mainly working on the relevant class or the irrelevant classes.

---

> ### Author Response · Authors · 2025-11-22
> **Part I**
>
> > **W1.** Intuition and idea are not well-expressed.
>
> **R1. Clarification of our intuition and novelty.** We appreciate the reviewer’s request for a clearer and more concise intuition， which helps us recognize our statement and organization could be improved to better convey our idea. We summarize our intuition and novelty in the following to clarify our contribution and arrangment of our contents.
>
> **Motivation.** Our work is motivated by 2 goals:
> 1. Existing unlearning operation is not that essential, causing utility damage when unlearning and requiring additional maintenance with the remaining data. We should examine the forgetting data's sample contribution to enable a more nuanced removal.
> 2. Sample contribution is hard to quantify, we should find a lightweight caculatable and optimizable proxy with only the pre-trained model for unlearning to withdraw.
>
> **Intuition.** Our intuition include 2 parts:
>
> 1. Moving **sample contribution** measurement from existing **parameter** space (i.e., the change in parameters $\Delta w$ when a sample is included or removed, which **influence function** measures) to **function**  space (i.e., $\partial A/\partial x$).
>
> 2. $\partial A/\partial x_i$ could be approximated by the learned model’s sensitivity to its input x, i.e. $\partial f(x)/\partial x$ with f = A(D), achieving light-weight optimization objective.
>
>
> **Novelty.** Our work differs from previous work in the following 4 aspects:
> 1. **An answer of what is learned and what to unlearn.** Our goal of suppressing sample contribution is simple but essential, giving a concrete answer from input sensitivity view. While existing removal operations (e.g., random relabelling, knowledge distillation) work well,  but rarely explicitly what quantity is/should being removed.
>
>
> 2. **Intuition 1 distinguishes our idea from influence function.** Derivation of influence function operates in **parameter space, which typically assumes local optimality, convexity, and access or approximation of the inverse Hessian**. These assumptions make them difficult to apply reliably and efficiently to modern DNNs, and, to our best knowledge, they have not yielded SoTA or remaining-data-free unlearning performance in practice. In contrast, our derivation does not rely on optimality conditions or explicit Hessian inversion. We **linearise the training algorithm in function space** and connect sample’s contribution $\partial A/\partial x$ directly to the pre-trained model’s input sensitivity, which could be directly optimized for DNNs.
>
> 3. **Intuition 2 finds an optimizable and lightweight objective for pratical unlearning optimization.** We fully understand that you feel Sec. 3.2 is too long and not immediately intuitive. This is precisely because the key conclusion of our intuition 2 (link between $\partial A/\partial x_i$ and $\partial f/\partial x$) is conceptually simple but not trivial to justify without some formalism. To aid intuitive understanding, we visualized the relationships among analyzed variables in Figure 2. In our revision, we would add the following explaination *"When a sample participates in training, the gradient it contributes induces an update of the model in function space, which inherently increases the learned function’s sensitivity to that sample’s input."* to the caption of Fig. 2 to further explain the intuition.
>
> 4. **Beyond existing impair-then-repair unlearning regime, towards efficient and practical remaining-data-free unlearning.** Our experiments show that targeting the dominant contribution term is already sufficient to obtain competitive unlearning behaviour in a remaining-data-free setting. Such experiments also provide solid experimal evidence that influence functions and sample contributions are essentially different: none of the influence function based method can achieve satisfactory unlearning performance without using the remianing data.
>
>
> **Summary.** We sincerely hope that the above explanation could clarify or convey the intuitions, distinctions and improvements in our work. In short, our primary contribution lies in introducing a new operational perspective, input sensitivity, to withdraw sample contribution for unlearning. And the derivation of this perpective is indeed non-trivial, which makes it difficult to express in only a few sentences. But this is exactly the complexity and main contribution of our work. The simplicity of obtained objective is indeed benefited from this derivation, and it is exactly what practical unlearning optimization desires. Therefore, we think MU-Mis is simple but essential, which enables us reach remaining-data-free regime, which is quite challenging.

---

> ### Author Response · Authors · 2025-11-22
> **Part II**
>
> > **W2.** Unsatisfying arrangement of related work, theoretical derivation and implementation details.
>
> **Our previous consideration.** When preparing the manuscript, we struggled with how to best allocate the limited space. As explained above, our main contribution lies in establishing a connection between sample contribution and input sensitivity of pre-trained model. Thereby, we thought presenting a formal establishment might help clarify the distinction of our perspective from previous work (i.e., $\partial A/\partial x_i$ vs. $\Delta w$), as well as indicating future directions for further improvement (i.e., the simplifications and omissions during derivation).
>
> **Confusion caused.** Receiving your feedback, we now realize that our choices, especially the placement of the related work (which contains important introduction of influence function) in Appendix, may have caused confusion and inconvenience. We're sincerely sorry that our current arrangment makes you uncomforable.
>
> **Better arrangement according to your feedback.** Thanks for your kind feedback which helps us better decide which part to be emphasized or condensed. We decide to i) shorten Section 3.2/3.3/3.4 a bit and hit the essence, i.e, the motivation and the differnce to influence-function,  more directly;  ii) move related-work section to the main paper; iii) try to extract some important detials into our main (we will try but cannot promise, since more experiments will be included after the rebuttal).

---

> ### Author Response · Authors · 2025-11-22
> **Part III**
>
> > **W3.** Add more SoTA baselines.
>
> **R3. Add 2 recent MU methods,  LoTus [1] and MunBa[2].** Thanks for your thoughtful consideration and recommendation. Among the 5 provided papers, [3] and [4] seem to be not open-sourced, thus we implement LoTus and MunBa to fullfill our experiments. Their performances are shown in Table 1-3. We could see that in full class and sub-class tasks, MU-Mis are still competitve against these 2 competitive methods.
>
> **Table 1. Unlearning Performance of MunBa, LoTus and MU-Mis for full-class unlearning on CIFAR-100, PinsFaceRecognition, and Tiny ImageNet.**
> | Method | CIFAR-100 | | | | | PinsFace | | | | | Tiny ImageNet | | | | |
> | - | - | - | - | - | - | - | - | - | - | - | - | - | - | - | - |
> | | RA | FA | TA | Avg. Gap↓ | MIA | RA | FA | TA | Avg. Gap↓ | MIA | RA | FA | TA | Avg. Gap↓ | MIA |
> | Pretrain | 76.41 | 79.69 | 76.47 | 26.84 | 95.80 | 93.49 | 100.00 | 93.59 | 34.54 | 100.00 | 65.85 | 62.00 | 65.50 | 21.03 | 93.59 |
> | Retrain | 76.52 | 0.00 | 75.76 | 0.00 | 2.87 | 93.06 | 0.00 | 92.25 | 0.00 | 0.00 | 65.36 | 0.00 | 64.90 | 0.00 | 4.80 |
> | MunBa | 74.09 | 0.00 | 73.40 | 1.60 | 9.30 | 91.44 | 1.63 | 90.27 | 1.74 | 0.00 | 64.22 | 0.00 | 63.88 | 0.72 | 7.80 |
> | LoTus | 76.48 | 5.00 | 75.87 | 1.72 | 0.00 | 92.87 | 0.00 | 91.75 | 0.23 | 0.00 | 65.02 | 0.00 | 64.65 | 0.20 | 0.00 |
> | MU-Mis | 76.42 | 0.00 | 75.64 | **0.07** | 0.00 | 92.98 | 0.00 | 92.13 | **0.07** | 0.00 | 64.95 | 0.00 | 64.85 | **0.15** | 0.20 |
>
> **Table 2. Unlearning Performance of MunBa, LoTus and MU-Mis for  sub-class unlearning on Rocket, Sea, and Lamp sub-class of CIFAR-20.**
> | Method | Rocket | | | | | Sea | | | | | Lamp | | | | |
> | - | - | - | - | - | - | - | - | - | - | - | - | - | - | - | - |
> | | RA | FA | TA | Avg. Gap↓ | MIA | RA | FA | TA | Avg. Gap↓ | MIA | RA | FA | TA | Avg. Gap↓ | MIA |
> | Pretrain | 85.26 | 80.73 | 85.21 | 26.53 | 92.89 | 85.09 | 97.66 | 85.21 | 5.94 | 91.81 | 85.31 | 74.22 | 85.21 | 21.30 | 92.82 |
> | Retrain | 84.85 | 2.69 | 84.07 | 0.00 | 12.06 | 84.60 | 80.93 | 84.61 | 0.00 | 51.61 | 85.12 | 11.31 | 84.40 | 0.00 | 7.06 |
> | MunBa | 81.43 | 7.00 | 80.80 | 3.67 | 7.20 | 80.64 | 84.00 | 80.66 | 3.66 | 60.00 | 81.17 | 17.00 | 80.69 | 4.45 | 5.00 |
> | LoTus | 35.93 | 39.00 | 36.04 | 44.42 | 18.60 | 73.12 | 81.00 | 73.39 | 7.59 | 61.20 | 27.44 | 12.00 | 27.40 | 38.45 | 33.60 |
> | MU-Mis | 84.28 | 2.91 | 83.50 | **0.49** | 0.07 | 84.35 | 81.00 | 84.33 | **0.20** | 1.25 | 84.36 | 11.70 | 83.66 | **0.63** | 0.00 |
>
> **Table 3: Unlearning Performance of MunBa, LoTus and MU-Mis for random subset (10$\%$) for CIFAR-10 and SVHN.**
> | Method | CIFAR-10 | | | | | SVHN | | | | |
> | - | - | - | - | - | - | - | - | - | - | - |
> | | RA | FA | TA | Avg. Gap↓ | MIA | RA | FA | TA | Avg. Gap↓ | MIA |
> | Pretrain | 100.00 | 99.96 | 94.68 | 1.84 | 92.64 | 100.00 | 100.00 | 96.48 | 1.65 | 83.11 |
> | Retrain | 100.00 | 94.51 | 94.75 | 0.00 | 84.77 | 100.00 | 95.12 | 96.40 | 0.00 | 81.24 |
> | MunBa | 93.19 | 94.57 | 93.19 | 2.81 | 58.20 | 96.84 | 96.46 | 96.84 | **1.64** | 66.70 |
> | LoTus | 90.55 | 96.73 | 90.55 | 5.29 | 56.10 | 86.66 | 94.31 | 86.65 | 7.96 | 78.80 |
> | MU-Mis | 97.76 | 97.43 | 91.50 | **2.80** | 33.04 | 95.48 | 95.50 | 94.22 | 2.36 | 26.11 |
>
> **Modification of our claim.** In light of the above dicsussion and rapid progress in unlearning algorithms, we fully acknowledge that our original claim that MU-Mis “outperforms SoTA remaining-data-dependent methods” is not well supported. We will revise our wording to state that MU-Mis is “on par with top-performing remaining-data-dependent methods in terms of the utility–privacy–efficiency tradeoff, while being remaining-data-free”. Especially, we will explicitly state that MU-Mis underperforms RUM and some other remaining-data-dependent methods in the random subset regime.
>
> **Implementation remarks.** To avoid under-estimating the performance of both LoTUS and MUNBa, we followed and slightly extended the hyper-parameter ranges in their original works, conducting a grid-search to ensure fair comparison. Specifically, we sweep learning rates in [1e-5, 5e-5, 1e-4, 5e-4, 1e-3, 5e-3, 1e-2, 5e-2, 1e-1] and epochs in [5, 10, 15, 20] for both methods. For LoTUS, we sweep alpha in [2, 4, 6, 8, 16] and employ Gumbel-softmax as the probability transformer. All other components strictly follow their original implementations. For MunBa, we extend unlearning epochs to 25, 30, 40 for PinsFaceRecognition and sub-class for a complete unlearning. We provide exact forget accuracy (FA) in the retrain model as teacher validation accuracy for LoTus, as in the sub-class setting, FA in the retrain model is neither 0 nor the validation accuracy of the pre-trained model.  We would report performances with mean and std across 5 independent runs in our revised manuscripts. **If there is any concern regarding our hyper-parameter implementation or search procedure, we sincerely welcome feedback and are willing to promptly correct any issue to ensure accurate reproduction.**

---

> ### Author Response · Authors · 2025-11-22
> **Part IV**
>
> > **W4.** Reference error and typos.
>
> **R4.Great thanks and we will re-check and correct.** Thank you for carefully catching these issues. We will carefully re-check all references and typography and correct any inconsistencies to ensure accuracy and clarity.
>
>
> > **Q1.** Performance beyond classification tasks, e.g., forgetting a single name or data entry.
>
> **RQ1. Conceptually feasible but requiring a more concrete task.** Thank you very much for your thoughtful suggestion about extending our method to forgetting individual entries. Our current formulation is indeed built around classification task, but we believe that the general intuition $\partial f/\partial x$ increases after training holds true across different tasks. This intuition manifests as an enlarged sensitivity gap between different class logits specifically in classification tasks.
>
> However, we think generalizing this idea to non-classification settings might require a nontrivial redesign of the loss and additional task-specific benchmarks, which might go a little beyond the scope of this submission. Therefore, after careful consideration, we preferred not to pursue this particular experiment at this stage. We're sorry for this and we appreciate your understanding, but we remain very open to further exploration if there is a specific task/benchmark.
>
>
> > **Q2.** Performance of regressing sensitivity norm of TC(target class) term to OC(other class) term.
>
> **RQ2. Successful unlearnng but failed preservation of model utility by regressing sensitivity magnitude of $f\_c$ to $f\_{c'}$.** Thank you for your insightful question, which provides another variant of withdrawing sample contribution by minimizing $( || \nabla\_x f\_{c}||\_F - ||\nabla\_x f\_{c'}||\_F\text{.detach()})^2$. (As we understand it?)
>
> **Freezing gradient of $f\_{c'}$ fails to preserve model utility.** During unlearning, we observe that as $||\nabla\_x f\_{c}||\_F$ decreases, FA decreases gradually. We find that although the gradient of $||\nabla\_x f\_{c'}||\_F$ is detached, its norm exhibits slight decrease as well when minimizing (which is similarly reported in our Figure A1 in Appendix F). As the sensitivity norm of $f\_c$ approaches $f\_{c'}$, i.e., the loss approaches 0, FA gradually stops to decrease and converges at 17.35$\%$. The final performance is reported in Table 2, with TC-OC.detach() refers to your mentioned variant. We attach the ablation study in our Table A10 for better understanding roles of each term in our loss.
>
>
>
> **Table 2. Ablation study on each term of our loss in full class (Rocket) unlearning. TC (Target Class) refers to the
> first term and OC (Other Class) refers to the second term.**
>
> | Methods       | RA               | FA               | TA               | Avg. Gap↓ | MIA               |
> | ------------- | ---------------- | ---------------- | ---------------- | --------- | ----------------- |
> | Pretrain      | 76.41            | 79.69            | 76.47            | 26.84     | 95.80             |
> | Retrain       | 76.52 $\pm$ 0.27 | 0.00 $\pm$ 0.00  | 75.76 $\pm$ 0.24 | 0.00      | 2.87 $\pm$ 0.46   |
> | TC            | 65.41 $\pm$ 0.00 | 0.00 $\pm$ 0.00  | 64.73 $\pm$ 0.00 | 7.38      | 1.40 $\pm$ 0.00   |
> | OC            | 70.13 $\pm$ 0.41 | 74.62 $\pm$ 1.56 | 70.21 $\pm$ 0.42 | 28.85     | 0.000 $\pm$ 0.000 |
> | TC-OC.detach()  | 70.24 $\pm$ 0.01 | 17.35 $\pm$ 0.04 | 69.45 $\pm$ 0.03 | 9.98      | 0.016 $\pm$ 0.045 |
> | TC-OC(MU-Mis) | 76.42 $\pm$ 0.07 | 0.00 $\pm$ 0.00  | 75.64 $\pm$ 0.07 | **0.07**  | 0.000 $\pm$ 0.000 |
>
> **Explanation of failure of this variant.** As demonstrated by Figure A1 in Appendix F, when minimizing $||\nabla\_x f\_{c}||\_F$, $||\nabla\_x f\_{c'}||\_F$ and retain accuracy (RA) will drop slightly as well, then RA will recover to its original level as the picking up of $||\nabla\_x f\_{c'}||\_F$ in our loss. Therefore, the magnitude of $||\nabla\_x f\_{c'}||\_F$ is essential for preserving model utility on remaining data (which is also supported by the ablation study in our Table A10). Thus, when regressing $||\nabla\_x f\_{c}||\_F$ to $||\nabla\_x f\_{c'}||\_F$, the norm of $||\nabla\_x f\_{c'}||\_F$ is not explicitly preserved, and drops during minimizing. Your proposed variant is a useful diagnose and it confirms that our joint‐term optimization is necessary.
>
>
>
> > **References:**
>
> > [1] LoTUS: Large-Scale Machine Unlearning with a Taste of Uncertainty, CVPR, 2025
>
> > [2] MUNBa: Machine Unlearning via Nash Bargaining, ICCV, 2025.
>
> > [3] Adversarial Machine Unlearning. ICLR. 2025.
>
> > [4] Decoupled Distillation to Erase: A General Unlearning Method for Any Class-centric Tasks. CVPR. 2025

---

### Official Review · Reviewer_pcTz · 2025-10-29

**Soundness:** 3
**Presentation:** 3
**Contribution:** 2
**Rating:** 2
**Confidence:** 4

**Summary:**

The paper addresses the issue of improving unlearning methods using a new per-example signal and developing a method that does not require access to remain data (RD), only to the forget set (FS).
Specifically, it proposes MU-Mis, and revolves around a specific per-example signal, namely that of the sensitivity difference between the example’s target-class logit and non-target logits. MU-Mis for each FS example shrinks this gap.
The method optimizes a fairly simple objective over only FS, ie no RD access).
The paper presents extensive experiments with CIFAR-100, Tiny-ImageNet, and CIFAR-20 with ResNet-18 and ViT models, using a comprehensive set of metrics. The central claim is that MU-Mis performs on par or better than MU methods that access RD, while being faster.

**Strengths:**

1. Simple, nice idea revolving around the notion of per-example sensitivity of FS examples, with a plan to efficiently compute it.

2. No RD access required.

3. Comprehensive evaluation and results/discussion.

4. Competitive performance vs 'older' baselines; often comparable to SalUn, with large speedups in some settings.

**Weaknesses:**

1. Incorrect/unsupported claims with respect to achieving same or even better performance than MU methods accessing RD.

This claim is made using rather older methods. MU-Mis is shown to be comparable to Salun (which is typically either the best or 2nd best performer to MU-Mis.

The authors appear to be unaware of the paper by Zhao et al (NeurIPS 24 - https://arxiv.org/abs/2406.01257) which actually is shown to significantly outperform Salun (on datasets and models also tested in this paper). Specifically, Zhao et al show that their method, coined RUM, can encapsulate Salun (and several other MU methods) and significantly improve their performance. Hence,  RUM-Salun should be expected to also significantly outperform MU-Mis, no?. The authors should address this concern and test whether this expectation is valid and perhaps revisit their claims if so, At any rate, that method appears to be a much stronger baseline than Salun and should be included here.

Interestingly, RUM appears to utiize a signal that is very similar to the sensitivity one used in this paper, namely the per-example memorization of FS examples (which the authors acknowledge to be akin to their sensitivity signal in this paper). So the link between the two approaches is very interesting and should be discussed.

However, Zhao et al go further, utilizing different MU methods for examples with different memorization levels. And this approach (let's call it full RUM) appears to significantly outperform even RUM-Salun... So, again, the claims should be revisited in light of these methods. And full RUM should be included as a baseline/competing algorithm.

And RUM appears to be able to use known memorization proxies, which are very efficient to compute - see the companion RUM paper (https://arxiv.org/html/2410.16516v2).

2. The MIA method used in the paper may be rather simplistic, and related discussions on MIA results do not really help MU evaluation.

The paper by Kurmanji et al (cited in the paper) actually put forth a LiRA-like strong MIA for MU. So (i) this should be acknowledged and justified and (ii) this or other similar strong-grade MIA attacks should be used. Or maybe you should use the MIA also used in the RUM papers for consistent comparisons?

3. On a related issue, the claim that "lower MIA score is better" seems counter-intuitive.

As the authors clearly state, performance should be viewed in relation to Retrain. So MIA scores of an MU method should be 'normalized' w.r.t. those of Retrain, no? The notion of MIA-gap (e.g., MIA_Retrain vs MIA_MU-Mis) in the RUM and other papers has been used for this reason. This should better be used here as well (with an appropriate MIA).

4. The paper may/will benefit by incorporating more advanced baselines, in addition to RUM-Salun and full RUM. A highly interesting set of MU methods were proposed for the NeurIPS MU competition - the paper https://arxiv.org/abs/2406.09073 presents a rich set of such methods which notably outperformed well known published algorithms. Other MU methods have emerged more recenlty as well.

**Questions:**

Please address the above weaknesses.
Especially the ones wrt RUM-Salun and full RUM, and wrt the MIA-related issues.

Overall, the idea is clever and not requiring RD access is highly desirable.
So, I remain open to improving my score if, for example, it is shown that the proposed method continues to outperform SOTA RD-based MU methods and if results for better MIA scores show excellent performance.

---

> ### Author Response · Authors · 2025-11-22
> **Part I**
>
> > **W1&W4.** Incorrect/Unsupported claims with respect to the performance compared with MU methods accessing RD. And full RUM and **more advanced baselines (W4)** should be included as a baseline/competing algorithm.
>
> **R1&R4. MU-Mis remain competitive with 2 newly added baselines in full/sub-class, while underperforming remaining-data-dependent methods in random subset scenario.** Thank you for your thoughtful consideration and a nice recommendation of RUM (Zhao et al. NeurIPS 2024) [1]. RUM provides a careful analysis of what makes unlearning hard and applies tailored unlearning strategies to each group of homogeneous subsets (i.e., by memorization level), achieving excellent performance in especially challenging random-subset setting. We greatly appreciate this line of work and fully agree that RUM sets a very strong standard for remaining-data-dependent unlearning, particularly for random subset forgetting.
>
> **The complementary roles of RUM, SalUn, and MU-Mis.** The overall aim of RUM, SalUn, and our MU-MIS is same: relieve the damage on remaining data when unlearing, but they focus on different aspects: RUM answers **how samples** (low -> middle -> high memorization) should be scheduled and treated during unlearning, SalUn identifies **which parameters** (gradient-based saliency map) should be updated, and MU-Mis figures out **what information** (sample contribution) should be removed. The three aspects are all imporant and numerically has their own advantages, e.g., RUM is perfect in random subset unlearning and MU-Mis work well without using remaining data.Therefore, **RUM, SalUn and MU-Mis are not competing in the same design space, and are naturally complementary and can be combined.**
>
> **Performances of MU-Mis compared with RUM in random subset unlearning.** Following experiment setup (including MIA implementation) of Table 1 in original paper of [1], we compare performances of RUM, SalUn and MU-Mis in the following Table 1. As you may have observed, on random subset unlearing tasks (Table A7 in our paper),  MU-Mis exhibits a bit worse performance than remaining-data-dependent methods (with the benefit of remianing-data-free is still there). Thus, we are very happy to see performance of RUM and especially the combination of RUM and MU-Mis. Combining RUM could improve performance of MU-Mis to some extent, but it still under-perform RUM by a noticeable margin on model utility. In our revised manuscript, we will add RUM and RUM+MU-Mis to our random subset unlearning and include discussion on the different aspects of RUM, Salun, MU-Mis to encourge better combinations and improvements.
>
> **Table 1. Unlearning performance of SalUn, RUM and MU-Mis on CIFAR-10 with ResNet-18.**
> | Method   | FA     | RA     | TA    | Avg.Gap | ToW    | MIA    | MIA.GAP | ToW_MIA |
> | -------- | ------ | ------ | ----- | ------- | ------ | ------ | ------- | ------- |
> | pretrain | 100.00 | 100.00 | 85.10 | 12.24   | 0.64   | 0.03   | 0.45    | 0.44    |
> | retrain  | 63.93  | 100.00 | 84.45 | 0.00    | 1.00   | 0.47   | 0.00    | 1.00    |
> | SalUn    | 73.63  | 99.99  | 81.64 | 4.17    | 0.88   | 0.79   | 0.31    | 0.64    |
> | RUM      | 66.40  | 100.00 | 84.41 | 0.84    | 0.97   | 0.99   | 0.52    | 0.48    |
> | RL       | 58.40  | 62.32  | 47.81 | 26.62   | 0.37   | 0.49   | 0.02    | 0.34    |
> | MU-Mis   | 66.90  | 70.48  | 57.37 | 19.85   | 0.50   | 0.42   | 0.06    | 0.40    |
> | RUM+MU-Mis | 60.70  | 77.45  | 61.18 | 16.35   | 0.58   | 0.41   | 0.06    | 0.56    |
>
> **MU-Mis remain competitive with newly added baselines LoTus [2] and MunBa [3] (Table 2-4 are referred to Part II due to space constraints.)** For RUM primarily focuses on random subset setting, we additionally incorporate 2 recent remaining-data-dependent methods, LoTus and MUNBa (whose codes are publicly available among the 5 competitive MU methods recommended by Reviewer 6Bdx), as new baselines. The results are shown in Table 2-4. Generally, MU-Mis remains competitive with these 2 remaining-data-dependent methods in the full and sub-class unlearning scenarios, without utilizing the remaining data.

---

> ### Author Response · Authors · 2025-11-22
> **Part II  (Continuing our responses R1&R4 to W1&W4 in Part I)**
>
> **Table 2. Unlearning Performance of MunBa, LoTus and MU-Mis for full-class unlearning on CIFAR-100, PinsFaceRecognition, and Tiny ImageNet.**
> | Method | CIFAR-100 | | | | | PinsFace | | | | | Tiny ImageNet | | | | |
> | - | - | - | - | - | - | - | - | - | - | - | - | - | - | - | - |
> | | RA | FA | TA | Avg. Gap↓ | MIA | RA | FA | TA | Avg. Gap↓ | MIA | RA | FA | TA | Avg. Gap↓ | MIA |
> | Pretrain | 76.41 | 79.69 | 76.47 | 26.84 | 95.80 | 93.49 | 100.00 | 93.59 | 34.54 | 100.00 | 65.85 | 62.00 | 65.50 | 21.03 | 93.59 |
> | Retrain | 76.52 | 0.00 | 75.76 | 0.00 | 2.87 | 93.06 | 0.00 | 92.25 | 0.00 | 0.00 | 65.36 | 0.00 | 64.90 | 0.00 | 4.80 |
> | MunBa | 74.09 | 0.00 | 73.40 | 1.60 | 9.30 | 91.44 | 1.63 | 90.27 | 1.74 | 0.00 | 64.22 | 0.00 | 63.88 | 0.72 | 7.80 |
> | LoTus | 76.48 | 5.00 | 75.87 | 1.72 | 0.00 | 92.87 | 0.00 | 91.75 | 0.23 | 0.00 | 65.02 | 0.00 | 64.65 | 0.20 | 0.00 |
> | MU-Mis | 76.42 | 0.00 | 75.64 | **0.07** | 0.00 | 92.98 | 0.00 | 92.13 | **0.07** | 0.00 | 64.95 | 0.00 | 64.85 | **0.15** | 0.20 |
>
> **Table 3. Unlearning Performance of MunBa, LoTus and MU-Mis for  sub-class unlearning on Rocket, Sea, and Lamp sub-class of CIFAR-20.**
> | Method | Rocket | | | | | Sea | | | | | Lamp | | | | |
> | - | - | - | - | - | - | - | - | - | - | - | - | - | - | - | - |
> | | RA | FA | TA | Avg. Gap↓ | MIA | RA | FA | TA | Avg. Gap↓ | MIA | RA | FA | TA | Avg. Gap↓ | MIA |
> | Pretrain | 85.26 | 80.73 | 85.21 | 26.53 | 92.89 | 85.09 | 97.66 | 85.21 | 5.94 | 91.81 | 85.31 | 74.22 | 85.21 | 21.30 | 92.82 |
> | Retrain | 84.85 | 2.69 | 84.07 | 0.00 | 12.06 | 84.60 | 80.93 | 84.61 | 0.00 | 51.61 | 85.12 | 11.31 | 84.40 | 0.00 | 7.06 |
> | MunBa | 81.43 | 7.00 | 80.80 | 3.67 | 7.20 | 80.64 | 84.00 | 80.66 | 3.66 | 60.00 | 81.17 | 17.00 | 80.69 | 4.45 | 5.00 |
> | LoTus | 35.93 | 39.00 | 36.04 | 44.42 | 18.60 | 73.12 | 81.00 | 73.39 | 7.59 | 61.20 | 27.44 | 12.00 | 27.40 | 38.45 | 33.60 |
> | MU-Mis | 84.28 | 2.91 | 83.50 | **0.49** | 0.07 | 84.35 | 81.00 | 84.33 | **0.20** | 1.25 | 84.36 | 11.70 | 83.66 | **0.63** | 0.00 |
>
> **Table 4: Unlearning Performance of MunBa, LoTus and MU-Mis for random subset (10$\%$) for CIFAR-10 and SVHN.**
> | Method | CIFAR-10 | | | | | SVHN | | | | |
> | - | - | - | - | - | - | - | - | - | - | - |
> | | RA | FA | TA | Avg. Gap↓ | MIA | RA | FA | TA | Avg. Gap↓ | MIA |
> | Pretrain | 100.00 | 99.96 | 94.68 | 1.84 | 92.64 | 100.00 | 100.00 | 96.48 | 1.65 | 83.11 |
> | Retrain | 100.00 | 94.51 | 94.75 | 0.00 | 84.77 | 100.00 | 95.12 | 96.40 | 0.00 | 81.24 |
> | MunBa | 93.19 | 94.57 | 93.19 | 2.81 | 58.20 | 96.84 | 96.46 | 96.84 | **1.64** | 66.70 |
> | LoTus | 90.55 | 96.73 | 90.55 | 5.29 | 56.10 | 86.66 | 94.31 | 86.65 | 7.96 | 78.80 |
> | MU-Mis | 97.76 | 97.43 | 91.50 | **2.80** | 33.04 | 95.48 | 95.50 | 94.22 | 2.36 | 26.11 |
>
>
>
> **Performance summary of added experiments.** We have included two new unlearning methods and compared with RUM for random subset tasks. The conclusion concindes with previous: (1) MU-Mis is signficantly better than any other remaining-free mehtods; (2) MU-Mis is better than remaining-data-dependent methods we have tested on full and sub-class unlearning; (3) MU-Mis under-performs some strong remaining-data-dependent methods, like Salun and RUM in random subset setting.
>
> **Modification of our claim.** In light of the above dicsussion and rapid progress in unlearning algorithms, we fully acknowledge that our original claim that MU-Mis “outperforms SoTA remaining-data-dependent methods” is not well supported. We will revise our wording to state that MU-Mis is “on par with top-performing remaining-data-dependent methods in terms of the utility–privacy–efficiency tradeoff, while being remaining-data-free”. Espeically, we will explicitly state that MU-Mis underperforms RUM and some other remaining-data-dependent methods in the random subset regime.
>
> **Summary and perspective on remaining-data-free MU.** We are grateful to the reviewers for helping us refine our claims and baselines. We also wish to emphasize （without diminishing the importance of RUM and other RD-based methods）that remaining-data-free unlearning is itself an important but challenging direction for advancing MU. MU-Mis is an admittedly preliminary attempt in this direction and remains far from perfect, but we hope it can be seen as a modest yet meaningful step forward, in our view, provides an encouraging indication that remaining-data-free MU is indeed attainable, as long as we keep working towards more precisely targeting what to forget (as in MU-Mis) and which/how to update/remove (as in SalUn and RUM). Therefore, we respectfully hope that our progress, despite its limitations, can be recognized and supported as part of the broader development of this field.

---

> > ### Comment · Reviewer_pcTz · 2025-11-26
> >
> > Regarding W1 and W4:
> >
> > Thank you for your additional experiments - much appreciated.
> > My reading of the results is different, however. We agree that RUM appears to be much better than MU-Mis (and the other methods). (BTW, there is strong evidence that SalUn performs badly for ToWMIA for bigger datasets and models!).
> > I did not understand why you did not run RUM for class/subclass unlearning? Is there something that prevents RUM from being used for this scenario?
> >
> > I do agree that MU-Mis appears to be better than the other baselines (not RUM) for (sub)class unlearning. **But this is the easiest by far unlearning scenario**. Pretrained models have managed to have robust distinguishing separation lines between different classes. So removing one class will not affect utility on the others. And this is where the "remaining-data free" MU approaches are expected to do well.
> >
> > One could/should argue that we need a solution that performs excellently for all cases - especially for the most difficult instance-based unlearning (arbitrary subsets of classes).

---

> > > ### Author Response · Authors · 2025-11-28
> > > **Further Response to W1 and W4 (Part II)**
> > >
> > > > 3. Expectation of a solution that performs excellently for all cases, especially for the most difficult instance-based unlearning (arbitrary subsets of classes).
> > >
> > > **Improvement of MU-Mis over existing RDF methods exist for all unlearning tasks, including random subset.** Surely, we sincerely agree with you that the random-subset unlearning should not be excluded. In fact, MU-Mis is designed to identify samples' contribution, which is for general machine unlearning tasks and could be applied to class, sub-class, and random subset scenarios. In all these cases, MU-Mis shows improvement from all remaining-data-free methods. In (sub)class unlearning tasks, MU-Mis could even outperform metods that access remaining data. For random subset task, we acknowledge there is a performance gap compared to the best remaining-data-dependent methods. But we do hope that the gap to the best remaining-data-dependent methods in random subset setting will not lower your judgment on our improvement from all remaining-data-free methods for all scenarios.
> > >
> > >
> > > **Challenge of Reamining-data-dependent for RDF methods in random subset setting.** We remain hopeful about the vision of a perfectly RDF method that preserves model utility under random subset settings. At the same time, we think it is also important to acknowledge the daunting challenges such a vision faces. In fact, before it is truly achieved, we are even not confident whether this goal is achievable, i.e., it may be fundamentally unrealistic for an RDF method to consistently match the performance of remaining-data-dependent approaches in random-subset unlearning. Moreover, many existing remaining-data-dependent approaches are still suffering from utility degradation in this setting. **We're afraid that currently, what we can do is to move, incrementally, toward that vision.**
> > >
> > >
> > > **A step towards RDF unlearning in random subset.** Nevertheless, we believe a better location of sample contribution offers a promising path to achive this goal. From our perspective, the main challenge lies in correctly identifying sample’s contribution. In our work, we find a calculable and optimizable approximation from the input-sensitivity perspective. Consequently, although our performance does not yet match that of data-dependent unlearning methods, we do improve over existing remain-data-free (RDF) approaches in the random-subset setting. We respectfully hope this improvement could be viewed as an important step forward. And closing the remaining gap in random-subset RDF unlearning will require more delicate and in-depth investigations.

---

> ### Author Response · Authors · 2025-11-22
> **Part III**
>
> > **W2.** Discussion of MIA criteria and stronger MIA should be used here as well.
>
> **Different viewpoints on MIA criteria.** Thank you for your valuable question regarding MIA criteria, and we would like to share and discuss our opinion with you. To our understanding, in machine unlearning, there are **two contrasting interpretations of MIA value**: one regard MIA as an **attacker** to measure the **residual membership signal** of forgetting data, thereby the **lower** MIA value is the better; the other one regard MIA as an **audit** of **output indistinguishbility** between unlearned and retrain model, thereby the **closer** MIA value to the retrain model is the better. We agree with the intuition behind the second viewpoint (and acknowledge the risk of “Streisand effect” or unintended signals when MIA scores become extremely low), but we argue that gap in MIA value may not serve as a strictly reliable ruler to measure the distance to the retrained model. Consequently, from the perspective of preventing privacy leakage of MU, we would prefer using a lower MIA score as an evaluation for residual membership signal.
>
> **KL divergence as a better ruler for output indistinguishbility.** Nevertheless, we fully agree that output indistinguishibility is a crucial criterion for unlearning performance. To capture this more reliably, we use KL-divergence metric（which we believe provides a more stringent and direct measure of behavioural closeness）to measure the difference between the unlearned model and retrained model in output distributions. As shown in Figure 7, MU-Mis consistently exhibits the lowest KL divergence to the retrained model than other methods during sequential unlearning, demonstrating its effectiveness in approaching the output distribution of the retrain model.
>
> **Results of stronger MIA.** Thank you for your thoughtful suggestion and we agree that the entropy-based MIA adopted in our paper is relatively simple, and a stronger MIA might provide more robust evidence for our claim. Following your suggestion, we have adopted RUM's implementation of MIA in Table 1. MU-Mis achieves the smallest MIA value, indicating a successful protection of privacy leakage. Also, its MIA value remains close to that of the retrained model, suggesting a benign output indistinguibility. Therefore, the overall conclusion is relatively consistent to our original MIA. Due to time constraints, we have not yet apply new MIA to other settings and have prioritized addressing your and the other reviewers’ remaining concerns. We would provide results of stronger MIA in full-class CIFAR-100 and sub-class CIFAR-20 (Sea) unlearning in our revised manuscript.

---

> > ### Comment · Reviewer_pcTz · 2025-11-26
> >
> > On W2:
> >
> > Thank you for understanding my concerns here. Your experiments using KL-div for measuring output indistinguishability are very welcome. And encouraging for your approach. However, if I understand correctly, these pertain to class unlearning, right? If so, in order for your claim(s) to be convincing, you would need to show this for arbitrary instance-based unlearning as well. My intuition tells me that your results there would be worse (but I do hope my intuition fails me here).

---

> ### Author Response · Authors · 2025-11-22
> **Part IV. Dicussion on input sensitivity correlations between MU-Mis and RUM (Discussion 1&2)**
>
> > **W3.** Disucssion of link between RUM and MU-Mis, including the used known memorization proxies, which are very efficient to compute.
>
> **R3.** We are very surprised and delighted that you identified the intriguing connection between MU-Mis and memorization (input loss curvature), which we considered to be quite interesting for future study, as briefly pointed in our discussion part.
>
> Before the following discussion, we would like to first clarify that MU-Mis focuses on **connection between sample contribution and input sensitivity**, while the link between **input sensitivity and memorization** (which RUM focuses) is not the key point in our research. Although our research did not undergo a deep investigation and discussion of link between input sensitivity and memorization, we really think our connections are very interesting and are very glad to discuss our current thinking with you in the following.
>
>
> **Dicussion 1. Correlations between senstivity gap and loss curvature in formula.**  The loss curvature for memorization proxy is $\\text{Curvature}(x) \propto \mathrm{tr}\Bigl(\nabla^2_x  \ell(x)\Bigr)$. For input sensitivity of loss, we have
>
> $$\nabla_x \ell（x）
> = (p - e_c)^\top \nabla_x f(x) = (1-p_c) \nabla_x f_c(x) - \sum_{c'\neq c} p_{c'} \nabla_x f_{c'}(x) \\
> = (\sum_{c'=1}^C p_{c'}) \nabla_x f_c(x) - \sum_{c'=1}^C p_{c'}\nabla_x f_{c'}(x) =\sum_{c=1}^C p_{c'} \;\bigl(\nabla_x f_c(x) - \nabla_x f_{c'}(x)\bigr).$$
>
> By comparing the formula of loss curvature and sensitivity gap used in MU-Mis, we could see that:
> $$
> \mathrm{tr}\Bigl(\nabla^2_x  \ell(x)\Bigr) = \sum_{c=1}^C p_{c'}\mathrm{tr}\Bigl(\nabla^2_x  \;\bigl(f_c(x) - f_{c'}(x)\bigr)\Bigr)  \\
> \text{v.s.} \\
> \|\nabla_x f_c(x)\|^2_F - \|\nabla_x f_{c'}(x)\|^2_F
> $$
>
> While both terms reflects sensitivity gap between target class and irrelevant classes, they refers to different order of gradient. **Specifically, loss curvature primarily aims at second-order sensitivity, while sensitivity gap in MU-Mis refers to the first-order sensitivity**. Mathematically, it seems that there is no straightforward equality or conserved bound that could directly links this two metrics. Therefore, we're afraid that we could not provide a general deterministic bound or functional equivalence without further assumptions.
>
> **Dicussion 2. Empirical investigations for sensitivity gap signatures across samples of different memorization/influence levels.**  Although there is few clue of mathematical relationship for analysis, we might empirically investigate their correlations by examining the sensitivity signatures across different samples, i.e, samples of different memorization/influence levels.
>
> **Experimental setup.** We partition training samples of **CIFAR-10** dataset according to their sample-wise **memorization score** (provided by [4] through training many models on different held-out subsets to measure each sample's self-influence on its own prediction) into low, middle, high memorization levels (following RUM [1]). We partition training samples of **CIFAR-100** dataset according to their **influence score** (provided by [4] through training many models on different held-out subsets to quantify each sample's cross-influence on test data) into the same 3 levels. The scores are available at [5].
>
> **Empirical finding: samples of different memorization/influence levels exhibit different sensitivity gap signatures.** We provide statistical metrics to demonstrate the distributions of different sample groups (low/middle/high memorization/influence levels) in Table 5 and Table 6. Surprisingly, we find that:
>
> 1. Highly memorized samples exhibit smaller sensitivity gap than low and middle memorized sample.
> 2. More influencial samples exhibit higher sensitivity gap (High > Middle > Low).
>
> Therefore, the magnitude of our sensitivity gap might implicitly reflects the sample’s level of memorization or influence score to some extent.
>
> **Table 5. Sensitivity gaps of different memorization level samples on CIFAR-10.**
> | Memorization Level | Mean | Std | 10th Percentile | 50th Percentile | 90th Percentile |
> | - | - | - | - | - | - |
> | Low | 10.7632 | 4.2386 | 6.0814 | 10.0681 | 16.4851 |
> | Mid | 10.5156 | 5.5244 | 3.9959 | 10.0496 | 17.6092 |
> | High | 6.7922 | 6.0817 | –0.7257 | 6.6459 | 14.5839 |
>
> **Table 6. Sensitivity gaps of different influence level samples on CIFAR-100.**
> | Influence Level | Mean | Std | 10th Percentile | 50th Percentile | 90th Percentile |
> | - | - | - | - | - | - |
> | Low | 36.1209 | 14.7241 | 19.3663 | 34.3506 | 56.0993 |
> | Mid | 39.3604 | 16.7700 | 21.3345 | 36.4983 | 61.0855 |
> | High | 42.5803 | 17.1963 | 23.9295 | 39.5377 | 65.7531 |

---

> > ### Comment · Reviewer_pcTz · 2025-11-26
> >
> > Thanks - this is very interesting -- although I do believe this is not a response to my W3 which asks for normalizing MIA scores instead of just reporting absolute values.
> >
> > Nonetheless, I sincerely appreciated the authors going to great lengths in order to delve into sample influence vs instance influence vs memorization etc. IMO this can serve as a foundation for really cool future results in this space. It is not clear whether the above apply to (sub)class unlearning or in general to unlearning, however.

---

> > > ### Author Response · Authors · 2025-11-28
> > > **Further Response to W3 and our Discussion**
> > >
> > > > 5.  Asks for normalizing MIA scores instead of just reporting absolute values.
> > >
> > > **R5.** Thank you again for your suggestion. By “normalize to Retrain model,” do you mean computing a gap between the MIA score of unlearned model and retrained model, i.e.  $MIA_{unlearned}-MIA_{retrain}$? Certainly, we will add this into the revised version. Since MIA value is not included in the average gap as we explained, the overall conclusion will keep unchanged.
> > >
> > >
> > > > 6.  It is not clear whether the above apply to (sub)class unlearning or in general to unlearning, however.
> > >
> > > **R6.** We sincerely thank you for recognizing our effort to investigate sample influence, instance influence, and memorization. While our above analysis is not limited to specific unlearning scenario, for suppressing sample contribution is a general mechanism that applies broadly. Thank you again for your thoughtful encouragement, and we believe it holds promise for more principled and generalizable unlearning methods.

---

> ### Author Response · Authors · 2025-11-22
> **Part IIV. Dicussion on input sensitivity correlations between MU-Mis and RUM (Discussion 3)**
>
> **Discussion 3. Essential goal of MU in terms of memorization, generalization and sample contribution.** There is an important question that might connect core idea of MU-Mis and RUM:
>
> **Q1.** What's the correlation between sample contribution and memorization？
>
> **Q2.** Is unlearning equivalent to alleviating sample's memorization level?
>
> **Brief clarification between memorization, sample influence and sample contribution.**
> * **Memorization** is defined as the change in its own prediction when a sample leaves the training set, i.e., self-influence.
> * **Sample influence** is the change on prediction of other data（test data), thereby more lies in cross-influence.
> * **Sample contribution** is the contribution of a training sample to all the model predictions, thereby comprising both the memorization (self-influence) and sample influence (cross-influence).
>
> **RQ1. Memorization level is not proportional to sample contribution.** On the one hand, high memorization can coincide with large contribution, e.g., long-tail but but genuinely informative examples, the model may need to “memorize” them to support generalization. On the other hand, high memorization does not guarantee substantial contribution: a model might over-fit noise, duplicates, or outliers, thereby exhibiting strong memory for those examples even though they contribute little or may harm the remaining data’s performance. Conversely, a training example may exert substantial influence on the model’s remaining predictions (high contribution) without having been deeply memorized (low memorization).
>
>
> **RQ2. Minimizing sample contribution is more essential for unlearning than reducing memorization.** Generally, in random subset case, to unlearn an instance, we would aim to alleviate model's memorization of it to prevent privacy leakage exploited by MIA. However, as discussed above, memorization (self-influence) and influence (cross-influence) consist sample contribution together. Therefore, in a broader sense, de-memorization is not that enough for a complete removal, while withdrawing sample contribution is more fundamental.
>
> **Optimization in MU-Mis and memorization proxy.** In light of the effectiveness of MU-Mis in full/sub-class unlearning, where removal is beyond merely reducing memorization, we think MU-Mis is not limited to directly alleviate memorization by optimizing certain curvature-based measures. Rather, we prefer to view MU-Mis as a practical and efficient proxy that could implicitly reduce memorization by suppressing a sample’s contribution. Additionally, given the less satisfactory results of MU-Mis in the random-subset unlearning setting, we suspect that a direct optimization of loss curvature (i.e., explicitly minimizing loss curvature around a speicific sample) might be sufficient to prevent privacy leakage and yield better model-utility trade-offs in this scenario.
>
> **Summary and future work.** We appreciate the insight provided by RUM, which inspires our further thinking about memorization, input sensitivity and unlearning. We fully acknowledge that our investigation is quite preliminary，but we believe that the input sensitivity might be a valuable and beneficial perspective for developing remaining-data-free unlearning, which is collectively demonstrated by MU-Mis and RUM, leaving a good starting point for future study.
>
>
> > **References**:
>
> > [1] What makes unlearning hard and what to do about it. NeurIPs, 2024.
>
> > [2] LoTUS: Large-Scale Machine Unlearning with a Taste of Uncertainty, CVPR, 2025
>
> > [3] MUNBa: Machine Unlearning via Nash Bargaining, ICCV, 2025.
>
> > [4] What Neural Networks Memorize and Why: Discovering the Long Tail via Influence Estimation, NeurIPs, 2020.
>
> > [5] https://github.com/google-research/heldout-influence-estimation.
>
> > [6] Unveiling privacy, memorization, and input curvature links, ICML, 2024.

---

> ### Author Response · Authors · 2025-11-28
> **Further response to W1 and W4 (Part I)**
>
> > 1. The authors focused on evaluating RUM only on random subset setting.
>
> **R1.** Thank you very much for your thoughtful question. We did not run RUM for class-wise unlearning is **not** because RUM is inherently limited to random subset unlearning, but mainly because we wanted to perform a completely fair and verifiable comparison under exactly the same experimental settings as the original RUM paper. In the original paper, the authors focused on evaluating RUM on random subset setting.
>
> We conjecture (but emphasize it remains conjectural) that the reason the original authors restricted to random subsets mainly because random subset is the most challenging scenario. While in the case of classwise unlearning, as you pointed out, inter-class data often exhibits relatively low sample correlation compared to random subsets drawn from a highly mixed dataset. In light of this, one would expect incorporating RUM to improve unlearning, but is likely more than SalUn alone. Also, given that our own method MU‑Mis already achieves stronger/comparable results than SalUn, including outperforming the “remaining-data-dependent” baseline, we believe that even adding a hypothetical RUM baseline would not overturn our main conclusions about MU-Mis. Nonetheless, as explained in our above R1&R4, RUM could also be combined MU-Mis as an improvement.
>
> **Summary.** According to your suggestion, we are not evaluating RUM on (sub-) class-wise unlearning task. But we cannot guarantee there are no additional obstacles when extending RUM in that direction, since there is not yet experiments about applying RUM to classwise unlearning (to our limited knowledge). Also, the true memorization score is available for CIFAR-100 and CIFAR-10, whereas in the subclass unlearning setting on CIFAR-20 we must resort to a memorization proxy, which is not fully aligned or strictly comparable with our random-subset reproduction of RUM. In light of the above reasons and the limited timeline of rebuttal, we have not yet attempted classwise RUM before. Hopefully, we can report the results by the ddl, even if we cannot reach the date, we will certainly include the results in the final version. Our expectation is that the result will likely be “better than SalUn, but not substantially different.” We hope this explanation clarifies our reasons and our conclusion of MU-Mis remains valid under fair comparison.
>
> > 2. Expectation of a solution that performs excellently for all cases, especially for the most difficult instance-based unlearning (arbitrary subsets of classes).
>
> **R2. Improvement of MU-Mis over existing RDF methods exist for all unlearning tasks, including random subset.** Surely, we sincerely agree with you that the random-subset unlearning should not be excluded. In fact, MU-Mis is designed to identify samples' contribution, which is for general machine unlearning tasks and could be applied to class, sub-class, and random subset scenarios. In all these cases, MU-Mis shows improvement from all remaining-data-free methods. In (sub)class unlearning tasks, MU-Mis could even outperform metods that access remaining data. For random subset task, we acknowledge there is a performance gap compared to the best remaining-data-dependent methods. But we do hope that the gap tot hem  in random subset setting will not lower your judgment on our improvement from all remaining-data-free methods for all scenarios.
>
> **Challenge of Reamining-data-dependent for RDF methods in random subset setting.** We remain hopeful about the vision of a perfectly RDF method that preserves model utility under random subset settings. At the same time, we think it is also important to acknowledge the daunting challenges such a vision faces. In fact, before it is truly achieved, we are even not confident whether this goal is achievable, i.e., it may be fundamentally unrealistic for an RDF method to consistently match the performance of remaining-data-dependent approaches in random-subset unlearning. Moreover, many existing remaining-data-dependent approaches are still suffering from utility degradation in this setting. **We're afraid that currently, what we can do is to move, incrementally, toward that vision.**
>
> **A step towards RDF unlearning in random subset.** Nevertheless, we believe a better location of sample contribution offers a promising path to achive this goal. From our perspective, the main challenge lies in correctly identifying sample’s contribution. In our work, we find a calculable and optimizable approximation from the input-sensitivity perspective. Consequently, although our performance does not yet match that of data-dependent unlearning methods, we do improve over existing remain-data-free (RDF) approaches in the random-subset setting. We respectfully hope this improvement could be viewed as an important step forward. And closing the remaining gap in random-subset RDF unlearning will require more delicate and in-depth investigations.

---

> ### Author Response · Authors · 2025-11-28
> **Further response to W2**
>
> > 4. KL-divergence of output distributions between the retrain and unlearned model in random subset setting.
>
> **R4. MU-Mis produces a smaller KL divergence on forget data than RUM and SalUn, implying a more reasonable unlearning mechanism.** Thank you very much for your further question. We examine the KL divergence between MU-Mis and retrained model in Table 1 (. We find that:
> 1. **MU-Mis yields the lowest KL divergence on the forget set.** Surprisingly, we find that MU-Mis achieves the smallest KL-divergence to the retrain model in forget set across different methods. We attribute this to their fundamentally different unlearning mechanisms: RUM (inherited from fine-tuning and SalUn) relies on random relabeling or catastrophic forgetting. As discussed in our paper, random relabeling might not provide a principled way to align the output distribution of the retrain model on forgetting data, leading to a sub-optimal output distribution.
> 2. **Compared with RL, MU-Mis does improve existing RDF method in random subset unlearning.** The overall KL divergence of RL is 4.3, while that of MU-Mis is 0.65, which is an substantial improvement, indicating that MU-Mis produces an output distribution much closer than other RDF methods.
>
> **Summary.** Consequently, although MU-Mis still underperforms remaining-data-dependent methods such as RUM+SalUn in terms of model utility, we think its lower KL divergence on the forgetting data is a
> meaningful step that advances unlearning towards a more faithful/reasonable unlearning.
>
> **Table 6. KL divergence between unlearned and retrain model in random subset unlearning.**
> | Method     | FA     | RA     | TA    | KL_Div_Forget | KL_Div_Test | KL_Div_Retain | KL_Div_All |
> | ---------- | ------ | ------ | ----- | ------------- | ----------- | ------------- | ---------- |
> | Pretrain   | 100.00 | 100.00 | 85.10 | 2.7131        | 0.3040      | 0.0018        | 0.1826     |
> | Retrain    | 63.93  | 100.00 | 84.45 | 0.0000        | 0.0000      | 0.0000        | 0.0000     |
> | SalUn      | 73.63  | 99.99  | 81.64 | 0.9941        | 0.3601      | 0.0548        | 0.1175     |
> | RUM        | 66.40  | 100.00 | 84.41 | 1.4462        | 0.2770      | 0.0057        | 0.0675     |
> | RL         | 58.40  | 62.32  | 47.81 | 5.9854        | 4.5657      | 4.1801        | 4.3005     |
> | MU-Mis     | 66.90  | 70.48  | 57.37 | 0.6525        | 1.1091      | 1.2391        | 0.7509     |
> | MU-Mis+RUM | 60.70  | 77.45  | 61.18 | 2.7765        | 1.0060      | 0.6421        | 0.7005     |

---

### Official Review · Reviewer_wSLf · 2025-10-31

**Soundness:** 3
**Presentation:** 3
**Contribution:** 3
**Rating:** 6
**Confidence:** 3

**Summary:**

This paper introduces MU-Mis, a novel remaining-data-free machine unlearning (MU) method based on the insight that a sample’s contribution to training is reflected in the model’s input sensitivity to that sample. The authors derive a theoretical connection between sample contribution $\frac{\partial A}{\partial x_i}$ and the model’s input Jacobian $\frac{\partial f}{\partial x}, and empirically validate that learned models exhibit higher input sensitivity to their training data. MU-Mis performs unlearning by minimizing the sensitivity gap between target and non-target logits on the forgetting samples, thus “withdrawing” their contribution without touching the remaining data. Extensive experiments show that MU-Mis achieves utility and privacy on par with or exceeding remaining-data-dependent methods, while being more efficient.

**Strengths:**

**S1**. The ability to unlearn without having access to the remaining data (fully or partially) is important and challenging, which this paper seems to be tackling.

**S1**.  The derivation linking training sample contribution to the model’s input sensitivity while not very rigorous, but it makes it very intuitive.

**S2**. The proposed method is computationally light. It requires only gradient computations on forgetting samples and uses simple stopping criteria.

**Weaknesses:**

**W1**. MU-Mis operates effectively for sample- or class-level forgetting but doesn’t directly extend to feature or attribute unlearning, or probably fine-grained cases where samples partially overlap. In that sense, the model is not domain general.

**W2**.  The method randomly selects a single irrelevant class $c'$ for the loss calculation in each step. An ablation study or justification for this random sampling (versus a deterministic average) can be clarifying.

**W3**. The stopping guideline relies on a threshold ratio $\delta$, which is a new hyperparameter. Table A1 shows that the optimal $\delta$ varies significantly across different tasks (ranging from 0.68 to 3.00). This overstates the claim of "effortless" hyper-tuning.

**W4**. Why RTE for pretrain and retrain is omitted overall? They can serve as good reference points.

**W5**. The proposed method is built on the input sensitivity of the model. In practice, training deep models usually involves some techniques to reduce such sensitivities. For example, adversarial training, augmentation, or different types of regularizations. It’s not clear how the model would perform in the presence of these practical techniques.

**Questions:**

**Q1**. How robust is the method to the choice of sensitivity metric (e.g., using $L_1$ vs Frobenius norm of $\nabla_xf$)?

**Q2**. I wonder if MU-Mis could inadvertently alter representations of semantically similar retained samples?

**Q3** Do you have any results showing larger forget-set sizes? E.g. > 30%?

---

> ### Author Response · Authors · 2025-11-22
> **Part I**
>
> > **W1.** MU-Mis doesn’t directly extend to feature or attribute unlearning, or probably fine-grained cases where samples partially overlap. In that sense, the model is not domain general.
>
> **R1.** Thank you for raising this point. We acknowledge that MU-Mis is currently limited to sample- or class-level forgetting and does not yet handle feature/attribute-level or partially-overlapping samples. We think the latter one might require more delicate design to achieve remaining-data-free, for there is higher demand of nuanced removal. But this is quite interesting and we will outline this point in our future work.
>
> > **W2.** An ablation study or justification for the random sampling of irrelevant class(versus a deterministic average) can be clarifying.
>
> **R2. Deterministic selection of irrelevant class yields a sub-optimal performance.** Thank you for your thoughtful suggestion of this ablation. We compare random label selection vs. deterministic label selection in Table 1. In the variant of deterministic selection, we fix irrelevant class c' to be the neighbor class of forgetting data, i.e., c' = c+1,  for each sample. We could see that such deterministic label selection yields a higher RA/TA drop and a non-zero FA, which suggests that a deterministic choice of a single irrelevant class when unlearning is less effective at suppressing residual information of the forgetting data. Therefore, we suspect selecting an irrelevant class at random in each step might make optimization more evenly balanced across different class logits to yield better unlearning performance.
>
>
> **Table 1. Unlearning performance Random label vs. deterministic label selection of MU-Mis on CIFAR-100 full-class unlearning.**
> | Methods | RA | FA | TA | Avg. Gap↓ | MIA |
> |---------|----|----|----|-----------|-----|
> | Pretrain | 76.41 | 79.69 | 76.47 | 26.84 | 95.80 |
> | Retrain | 76.52 $\pm$ 0.27 | 0.00 $\pm$ 0.00 | 75.76 $\pm$ 0.24 | 0.00 | 2.87 $\pm$ 0.46 |
> | Random (MU-Mis) | 76.42 $\pm$ 0.07 | 0.00 $\pm$ 0.00 | 75.64 $\pm$ 0.07 | 0.07 | 0.000 $\pm$ 0.000 |
> | Deterministic | 73.35$\pm$ 0.12 | 2.00$\pm$ 0.13 | 72.71$\pm$ 0.17 | 2.74 | 0.334$\pm$ 0.012 |
>
> > **W3.** Over-stated claim of "effortless" hyper-tuning.
>
> **R3. Our hyper-tune-friendly claim is not strict and should be limited to a fixed task/dataset.** Thank you for pointing this out. We apologize for the imprecision in our wording. Our intention was to state that the stopping threshold ratio could be determined relatively easily by observing FA and RA trends during unlearning a specific dataset/task (as illustrated in Figures A1 and A2). And we now realize that our phrasing implied a broader “effortless tuning” claim across tasks, which is inaccurate. We will revise the manuscript to clarify this limitation and adjust the wording as "effortless hyper-tuning under each dataset" to avoid overstatement.
>
> >**W4. RTE of pre-train and retrained model in our experiments.**
>
> **R4. We will incorporate RTE into our manuscript.**  Thank you for your kind suggestion. Originally, our tables reported RTE in seconds, which we felt might appear large to interpret for pre-train and retrain model. And now we agree with you and would include our RTE of pre-train and retrain model (in Table 2)  to our manuscript.
>
> **Table 2. RTE (seconds) of pretrain, retrain and MU-Mis across different datasets and models.**
> | Method   | Full-class (CIFAR-100) | Full-class (PinsFaceRecognition) | Full-class(TinyImagenet) | Sub-class (CIFAR-20) | Random (CIFAR-10) | Random (SVHN) |
> | -------- | ---------------------- | -------------------------------- | ------------------------ | -------------------- | ----------------- | ------------- |
> | Pretrain | 10880                  | 13144                            | 13600                    | 6910                 | 14322             | 16904         |
> | Retrain  | 7432                   | 11400                            | 10367                    | 4298                 | 12800             | 13890         |
> | MU-Mis   |    30                    |    83                              |      21                    |        10              |      116             |    116           |

---

> ### Author Response · Authors · 2025-11-22
> **Part II**
>
> > **W5.**  Unlearning performance under input sensitivity regularization techniques, e.g., data augmentation, adversarial training.
>
> **R5. Data augmentation has been applied in our experiments.** Thank you very much for this insightful question. In our experiments, standard data augmentation was applied, therefore MU-Mis is effective when there is data augmentation.
>
> **MU-Mis still works with Jacobian regularization.** For further investigation, we employed Jacobian regularization [1] to our pre-training and re-training. Jacobian regularization penalizes model's input-output sensitivity around training sample x, i.e., $\|J(x)\|=\|\frac{\partial f(x)}{\partial x}\|_F$ to encourage a smoother decision boundary for better robustness and generalization. For Jacobian regularized training tasks some time, we evaluated the unlearning of an full class on the CIFAR-10 dataset for a quick illustration. Specifically, we train ResNet18 for 30 epochs with lr = 0.1, bs = 256, regularization coefficient =0.01. The unlearning performance is shown in Table 3, where the first 3 rows are performances on vanilla model (with data augmentation but without Jacobian regularization). In Jacobian regularized model, it seems to be more difficult to fully forget a class by MU-Mis, but it still preserves model utility well without remaining data.
>
> | Method   | FA              | RA               | TA               | Avg. Gap↓ | MIA             |
> | -------- | --------------- | ---------------- | ---------------- | --------- | --------------- |
> | Pretrain | 88.54           | 83.59            | 84.03            | 32.54     | 0.9498          |
> | Retrain  | 0.00 $\pm$ 0.00 | 86.11 $\pm$ 0.11 | 77.56 $\pm$ 0.15 | 0.00      | 9.56 $\pm$ 0.08 |
> | MU-Mis   | 1.40 $\pm$ 0.35 | 84.13 $\pm$ 0.28 | 75.98 $\pm$ 0.52 | 1.62      | 0.98 $\pm$ 0.12 |
> | Jacob Regularization = 0.01         |                 |                  |                  |           |                 |
> | Pretrain | 88.53           | 83.33            | 83.11            | 32.58     | 0.9436          |
> | Retrain  | 0.00 $\pm$ 0.00 | 85.77 $\pm$ 0.23 | 77.35 $\pm$ 0.07 | 0.00      | 0.11 $\pm$ 0.09 |
> | MU-Mis   | 5.67 $\pm$ 0.32 | 82.37 $\pm$ 0.14 | 75.12 $\pm$ 0.07 | 3.77      | 0.21 $\pm$ 0.09 |
>
>
> > **Q1.** Robustness of performance to choice of sensitivity metric.
>
> **RQ1. L1 norm fails and L2 norms yields similar performance to F norm.** Thank you for your interesting question, which helps us better understanding our optimization on input sensitivity. We investigate performances of MU-Mis with different norm metrics in Table3. We can see that L2 norm yields similar performance to Frobenius norm (original MU-Mis) and retrain, suggesting that the method is robust as long as the norm aggregates all components of the sensitivity vector. In contrast, norms that emphasize only extreme coordinates (e.g., inf norm) tend to over-penalize the worst-case direction, which empirically hurts model utility. As for L1 sensitivity measure you mentioned, we observed that model tends to breakdown even under a  very small learning rates （1e-8). We think this might because L1 norm is non-smooth, and optimizing it induces abrupt and sparse updates in model outputs.
>
>
> **Table 3. Performances of MU-Mis under different sensitivity norm choices.**
> | Methods  | RA               | FA              | TA               | Avg. Gap↓ | MIA              |
> | -------- | ---------------- | --------------- | ---------------- | --------- | ---------------- |
> | Pretrain | 76.41            | 79.69           | 76.47            | 26.84     | 95.80            |
> | Retrain  | 76.52 $\pm$ 0.27 | 0.00 $\pm$ 0.00 | 75.76 $\pm$ 0.24 | 0.00      | 2.87 $\pm$ 0.46  |
> | Frobenius norm   | 76.42 $\pm$ 0.07 | 0.00 $\pm$ 0.00 | 75.64 $\pm$ 0.07 | **0.07**  | 0.000 $\pm$ 0.00 |
> | L2 norm  | 75.53 $\pm$ 0.05 | 1.95 $\pm$ 0.02 | 74.89 $\pm$ 0.11 | 1.27      | 0.000 $\pm$ 0.00 |
> | Inf norm | 50.34 $\pm$ 0.12 | 1.95 $\pm$ 0.04 | 50.11 $\pm$ 0.14 | 17.93     | 0.710 $\pm$ 0.02 |
>
>
>
> > **Q2.** Whether MU-Mis inadvertently alter representations of semantically similar retained samples.
>
> **RQ2. Evidence of minor embedding shift via RA/TA and KL divergence.** Thank you for this important question. Yes, we acknowledge that there is a possibility of such unintended representation drift. This might be inferred from the performance drop in our RA and TA after MU-Mis unlearning. Your question alerts us that representations of certain remaining data may indeed have shifted. But it could also be expected that such drift is not that catastrophic for model utility is largely preserved.
>
> In general, RA, TA of MU-Mis unlearned model as well as KL divergence in Figure 7 could provide certain evidence that such drifts exist, but are not that catastrophic. If required, we would be glad to conduct a dedicated empirical analysis (e.g., embedding similarity or CKA between pre-train and unlearned representations) to probe the extent of such shift.

---

> ### Author Response · Authors · 2025-11-22
> **Part III**
>
> > **Q3.** Results of larger forget-set sizes(e.g., >30%)
>
> **RQ3. It is more difficult to preserve model utility under larger forget size (50$\%$).** We conduct unlearning on a random subset comprising 50% of the training data in Table 4. Compared with the pre-trained model, the narrowed Avg. Gap for MU-Mis and the reduced MIA vulnerability indicate that MU-Mis exhibits certain effectiveness. At the same time, we observed a modest drop in RA and TA for MU-Mis, implying that as the size of the random subset increases, preserving model utility becomes more challenging for remaining-data-free method.
>
> **Table 4. MU-Mis performance in random subset (50$\%$) unlearning.**
> | Method | RA | FA | TA | Avg. Gap↓ | MIA |
> |--|----|----|----|----------|-----|
> | Pretrain | 100.00 | 99.96 | 94.68 | 5.99 | 92.64 |
> | Retrain | 100.00 $\pm$ 0.23 | 91.45 $\pm$ 0.41 | 92.63 $\pm$ 0.15 | 0.00 | 73.21 $\pm$ 0.08 |
> | MU-Mis | 97.72 $\pm$ 0.35 | 97.82 $\pm$ 0.28 | 91.18 $\pm$ 0.52 | 3.37 | 83.15 $\pm$ 0.19 |
>
> > **Reference**:
>
> > [1] Robust learning with jacobian regularization，arXiv，2019.

---

> > ### Comment · Reviewer_wSLf · 2025-11-26
> >
> > Thank you for sharing the new results.
> >
> > Overall it seems that the method performs fairly well on 'accuracy metrics' in normal unlearning settings. The MIA used is generally a weak one, which I don't think is reliable. Do you have any thoughts on how your method would perform under other attacks, like [1] or recovery attacks?
> >
> > [1] https://arxiv.org/abs/2112.03570

---

> > > ### Author Response · Authors · 2025-11-28
> > > **Further Response to LiRA as MIA**
> > >
> > > > Q4. Performance of MU-Mis under other attacks, like LiRA or recovery attacks?
> > >
> > >
> > > **RQ4.** Thank you for your thoughtful consideration. We fully agree that LiRA is a very powerful MIA for auditing privacy leakage. However, considering that online LiRA requires training a large number of shadow models, would you mind we examine our privacy leakage with a comparably good unlearning-adapted LiRA proposed by SCRUB [1] (which is also mentioned by Reviewer pcTZ)?  We report SCRUB-LiRA examined results  across 3 unlearning settings in Table 5 (MIA-Entropy refers to MIA used in our original paper).  We could see that in MU-Mis unlearned model, SCRUB-LiRA is quite close to random guessing, indicating that the forgetting data in MU-Mis unlearned model is similar to non-members.
> > >
> > > **Table 5. SCRUB-LiRA scores across different unlearning settings.**
> > > | Model                  | FA (%) | RA (%) | TA (%) | MIA-Entropy | MIA-SCRUB    |
> > > | ---------------------- | ------ | ------ | ------ | ----------- | ------------ |
> > > | Fullclass-Cifar100-Rocket                       |        |        |        |             |              |
> > > | Pretrain               | 86.00  | 76.37  | 76.56  | 95.40       | 73.40 ± 2.60 |
> > > | Retrain                | 0.00   | 76.86  | 76.07  | 6.60       | 96.40 ± 1.60 |
> > > | MUMIS                  | 0.00   | 76.37  | 75.71  | 0.00        |90.40 ± 2.00 |
> > > | Subclass-Cifar20-Sea |        |        |        |             |              |
> > > | Pretrain               | 97.00  | 85.10  | 85.14  | 91.80       | 64.40 ± 2.40 |
> > > | Retrain                | 80.00  | 84.85  | 85.00  | 53.00       | 47.40 ± 6.40 |
> > > | MUMIS                  | 80.00  | 84.26  | 84.15  | 0.60        | 53.00 ± 4.60 |
> > > | Random-Cifar10-10%     |        |        |        |             |              |
> > > | Pretrain               | 99.96  | 100.00 | 94.68  | 92.64       | 56.64 ± 0.60 |
> > > | Retrain                | 94.51  | 100.00 | 94.75  | 84.77       | 49.68 ± 0.30 |
> > > | MUMIS                  | 97.43  | 97.42  | 91.50  | 83.34       | 52.66 ± 0.40 |
> > >
> > >
> > > **Remark: LiRA-SCRUB and MIA-Entropy is complementary.** The goal of the adversary in LiRA-based MIA is to distinguish “forgotten samples” from unseen (non-member) samples. Therefore, their criteria of successful unlearning is that in the unlearned model, the adversary could not well distinguish forgotten samples from unseen samples, i.e., MIA collapse to random guessing (50$\%$) for random subset unlearning. LiRA-based MIA examines how indistinguishable the forgetting data to the non-members, thereby the closer of SCRUB-MIA value to 50$\%$ in random subset setting is the better. While MIA in our paperexamines how much forgotten samples are predicted as members, thereby the lower is the better. **That is to say, LiRA-based MIA examines the extent of *non-membership* of forgetting data, while our MIA-Entropy examines the extent of membership, i.e., the residual membership signals of forgetting data. Therefore, these 2 MIAs are not contradictory or competing, but complementary, and both results consistently support our conclusions.** Once again, we sincerely appreciate your suggestion, which strengthens the comprehensiveness of our privacy evaluation.
> > >
> > > > **References:**
> > >
> > > > [1] Towards unbounded machine unlearning,NeurIPS, 2023.

---

### Official Review · Reviewer_oXDL · 2025-11-01

**Soundness:** 2
**Presentation:** 3
**Contribution:** 2
**Rating:** 4
**Confidence:** 3

**Summary:**

This paper proposes a remaining-data-free unlearning method that treats a sample’s training "contribution" as being reflected by the model’s input sensitivity to that sample. It empirically observes that training enlarges the gap between the target-logit input gradient and irrelevant-logit gradients for seen data, and shows that a retrained-without-forget set model shrinks this gap. The method then unlearns by minimizing this sensitivity-gap on forget samples with a simple stopping rule tied to the recovery of irrelevant-class sensitivity. Experiments across several datasets/architectures claim utility comparable to strong remaining-data-dependent baselines, plus efficiency and resilience under sequential unlearning.

**Strengths:**

1. The paper proposes a remaining-data-free unlearning loss that minimizes the gap between target-logit and irrelevant-logit input sensitivities, along with a concrete stopping rule. This is simple and easy to implement, and the method itself uses only the forget set.

2. The method is evaluated on multiple image-classification settings (full-class, sub-class, random-subset; ResNet-18, ViT), and the proposed method "MU-Mis" matches or beats SOTA that does use remaining data in several cases (e.g., CIFAR-100) while being efficient, especially on ViT.

**Weaknesses:**

1. The theoretical analysis in Section 3.2, which aims to formally connect a sample's "contribution" to its "input sensitivity", is weak. The derivation relies on several convenient "simplifications" and "omissions". The central claim that the "Residual Term" is negligible is not rigorously proven and is instead justified by referencing a simple MLP example in the appendix. The theory feels more like a post-hoc rationalization for a empirical finding rather than a solid derivation that leads to the method.

2. The claim "first remaining-data-free method to outperform SOTA remaining-data-dependent" of the paper reads a bit too strong. In several tables the advantages are small or mixed (e.g., CIFAR-100/Tiny-ImageNet where SSD/BT are very close, and the performance in random-subset task is a bit weak). The novelty/significance claim could be scoped more carefully.

**Questions:**

1. Do different samples (e.g. highly memorized vs. typical ones) exhibit different sensitivity signatures? Or if the method is uniformly effective at unlearning different samples?

2. Could you provide more details of the MIA (e.g. attack family, calibration, shadow data choices, etc.) you used in the paper? The current MIA details are brief and privacy claims are sensitive to these choices.

---

> ### Author Response · Authors · 2025-11-22
> **Part  I**
>
> > **W1.** Insufficient theoretical justification for the negligibility of residual term in Section 3.2.
>
> **R1. Theoretical derivation aims to find a calculable surrogate for sample contribution to facilitate unlearning, and the residual term is more like an incalculable term.** Thanks for your questions regarding the theoretical dervation in Section 3.2 and we  interpret your concern about residual term as a shared recognition that pursing better localization of sample contribution is essential for advancing MU.
>
> **Motivation of our theoretical part.** In principle, quantifying sample's contribution is intrinsically difficult due to the incrementality of training. Our theoretical analysis is motivated by the dilemma of current measures of sample contribution. As discussed in our introduction (Line 45-50) and related work (Line 743-757), existing work mainly focus in **parameter space**, i.e., how parameters change $\Delta W$ when sample is removed. However, such parameter-space analysis are often neither reliable nor efficient for DNNs. This motivates us to shift sample contribution to **function space**, $\partial A/\partial x_i$. However, at first sight, this quantity also appears intractable.
>
> **Intention of theoretical analysis is to derive a **calculable** objective to facilitate unlearning optimization.** Given above difficulties, our goal is to find a computable surrogate that could guide practical unlearning optimization, and specifically, using only the pre-trained model. Therefore, we analyze the training dynamics theoretically in Section 3.2 to identify traces of sample contribution that are implicitly encoded in the pre-trained model. But we fully acknowledge current analysis is preliminary and imperfect and the "simplifications" and "omissions" you mentioned are quite accurate. In below, we would like to first outline and explain the faced challenges of more rigorous analysis in detail, and then provide additional empirical evidence for the established connection in the hope of alleviating your concern.
>
> **Faced challenges of more rigorous analysis:**
> 1. **Simplification during theoretical derivation is common in DNN analysis.** The key simpliciation is the first-order Taylor expansion, which simplifies the training dynamics to linear ones in a local gradient update. We acknowledge that this does not precisely capture the practical DNN training dynamics. But such linearisations are common in learning theory for DNNs, e.g., the Neural Tangent Kernel regime. Thereby, we think such simplification is acceptable.
>
> 2. **The residual term is theoretically and empirically intractble.** As can be seen, the residual term involves accumulations of inner-products among sample-wise gradient over training trajectory, which are analytically intractable. Moreover, its high dimensionality also renders direct computation prohibitively expensive. Moreover, since the true sample contribution $\partial A/\partial x_i$ itself is unavailable, we cannot obtain a reliable numerical estimate of the residual term for empirical validation.
>
> **The computable component enables a step toward remaining-data-free unlearning.** Therefore, the residual term, as you observed, is not included in our unlearning objective. In hindsight, “residual term” may be a misleading name, and it might be better viewed as an incalculable term. In other words,  **our analysis conceptually decomposes the samples' contribution to "input sensitivty" and an "incalculable term".** Our comphensive experiments have demonstrated that minimizing the calculable input sensitivty can succesuflly suppress the forgetting samples' contribution, achiving our goal of a remaining-data-free unlearning. We believe that identifying a tractable component from current “incalculable” term and developing a finer location of sample contribution could further enhance the performance of remaining-data-free unlearning.
>
> **Empirical support for our theoretical analysis.** In our following response to your first question (RQ1), we would empirically show that samples of different influence scores does exhibit different magnitudes of sensitivity gap. We think this positive correlation could confirm the effectiveness of our surrogate to some extent.
>
> **Summary.** Overall, while our theoretical section does not provide a full formal bound, we hope it nonetheless offers a meaningful intuitive justification of the connection between input sensitivity and sample contribution, as well as a new perspective of viewing sample contribution. We sincerely regret the limitation of our current explanation, and would be glad to see any improvement of this connection and remaining-data-free unlearning.
>
> **Modification.** We will (i) state limitations of our analysis more explicit, (ii) clarify the main purpose of our analysis is for a computable surrogate for sample contribution, and (iii) strengthen it with the positive trend between sample's influence score and sensitivity gap.

---

> ### Author Response · Authors · 2025-11-22
> **Part II**
>
> > **W2.** Performance claim should be more precise，rather than broadly asserting superiority.
>
> **R2. Refine our claim to specify and delineate where advantages occur.** Thank you for this thoughtful remark. We will revise our claim of performance as ''on par with top performing remaining-data-dependent methods''， and clarify that our novelty lies in moving beyond impair-then-repair regime of current unlearning and matching them in a remaining-data-free way. We will explicitly delineate that our method approaches remaining-data-dependent methods more closely in class-wise unlearning, while acknowledging that there remains clear room for further improvement in the random-subset unlearning scenario in experiment part.
>
> > **Q1.** Sensitivity signatures and performance of MU-Mis across different samples.
>
> **RQ1. Samples of different memorization level and influence score exhibit different sensitivity signatures.** Thank you for this very insightful and interesting question. We investigate the sensitivity signatures of different memorization levels and influence scores and corresponding performances of MU-Mis.
>
> **Experiment setup.** We partition training samples of CIFAR-10 dataset according to their sample-wise *memorization score* (provided by [2] through training many models on different held-out subsets to measure each sample's self-influence on its own prediction) into low, middle, high memorization levels (following RUM [3]). We partition training samples of CIFAR-100 dataset according to their *influence score* (provided by [2] through training many models on different held-out subsets to quantify each sample's cross-influence on test data) into the same 3 levels. The scores are available at [4].
>
>
> **Experiment result.** Interstingly, the distributions of sensitivity gap of different sample groups are shown in Table 1 and Table 2. We find that :
> 1. Highly memorized samples exhibit smaller sensitivity gap than low and middle memorized sample.
> 2. More influencial samples exhibit higher sensitivity gap (High > Middle > Low).
>
> Interestingly, [5] reveals that there is a profound connection among input sensitivity, generalization and memorization. In the context of unlearning, we think sample contribution comprises both the **memorization** (self-influence, i.e., contribution to prediction on its own) and **sample influence** （cross-influence, i.e., contribution to prediction of other data) part. Therefore, regardless of their magnitude of sensitivity gap, all such contributions should be withdrawn (thereby sensitivity gap minimized).
>
> **Empirical evidence for W1 (theoretical justification).** Importantly, we think above empirical findings provide meaningful support for our use of sensitivity gap as a proxy for sample contribution.
>
> **Table 1. Sensitivity gaps of different memorization level samples on CIFAR-10.**
> | Memorization Level | Mean    | Std    | 10th Percentile | 50th Percentile | 90th Percentile |
> | - | - | - | - | - | - |
> | Low|10.7632 | 4.2386 | 6.0814| 10.0681| 16.4851|
> | Mid|10.5156 | 5.5244 | 3.9959| 10.0496| 17.6092|
> | High| 6.7922  | 6.0817 | –0.7257| 6.6459| 14.5839|
>
> **Table2.  Sensitivity gaps of different influence level samples on CIFAR-100.**
> | Influence Level | Mean| Std| 10th Percentile | 50th Percentile | 90th Percentile |
> | -| - | - | - | - | - |
> | Low | 36.1209 | 14.7241 | 19.3663 | 34.3506| 56.0993|
> | Mid  | 39.3604 | 16.7700 | 21.3345| 36.4983| 61.0855|
> | High | 42.5803 | 17.1963 | 23.9295| 39.5377 | 65.7531|
>
> **Removing highly-memorized samples induced larger utility drop.**  We evaluate MU-Mis when respectively unlearning samples with low, medium, and high memorization levels in Table 3. We could see that removing those highly-memorized samples causes a more substantial utility drop in the remaining data, indicating that the performance of MU-Mis is not uniform across samples of different memorization levels. But maybe this is expected, as unlearning highly entangled samples is intrinsically difficult for any unlearning method [3,6].
>
> **Table 3. Performance of MU-Mis when unlearning low, middle, high levels of samples.**
> | Setting | Method | FA | RA | TA | Avg. Gap | MIA |
> |-|-|-|-|-|-|-|
> | Low | Original | 100.00 | 100.00 | 85.10 | 0.45 | 0.013 |
> |  | Retrain | 99.83 | 100.00 | 83.93 | 0.00 | 0.049 |
> |  | MU-Mis | 100.00 | 99.96 | 83.32 | 0.27 | 0.041 |
> | Mid | Original | 100.00 | 100.00 | 85.10 | 9.02 | 0.019 |
> |  | Retrain | 74.40 | 100.00 | 83.63 | 0.00 | 0.539 |
> |  | MU-Mis | 93.63 | 87.09 | 69.29 | 15.49 | 0.194 |
> | High | Original | 100.00 | 100.00 | 85.10 | 26.83 | 0.055 |
> |  | Retrain | 21.63 | 100.00 | 82.99 | 0.00 | 0.811 |
> |  | MU-Mis | 46.10 | 66.79 | 57.03 | 27.88 | 0.607 |
>
> **Summary.** Thank you once again for your insightful question, which helps us uncover additional and intriguing correlations that we believe are worthy of future investigation. We would incorporate these findings, limitations and discussions into our revised manuscript.

---

> ### Author Response · Authors · 2025-11-22
> **Part III**
>
> > **Q2.** More details of MIA.
>
> **RQ2.** Our MIA follows the entropy-based score attack introduced by Chundawat et al. [7]. For each sample, we compute the predictive entropy of the model’s softmax probabilities. We form an attack training set by collecting entropy values on remaining data (members) and test data (non-members), and train a logistic regression classifier to learn decision boundary between member and non-member from these statistics, replacing hand-crafted thresholds. Then we report the fraction of forgetting data predicted as members as MIA value. We will incorporate this introduction in our revision to ensure clarity.
>
>
> > **References:**
>
> > [1] Does learning require memorization? a short tale about a long tail, ACM SIGACT symposium on theory of computing, 2020.
>
> > [2] What Neural Networks Memorize and Why: Discovering the Long Tail via Influence Estimation, NeurIPs, 2020.
>
> > [3] What makes unlearning hard and what to do about it, NeurIPs, 2024.
>
> > [4] https://github.com/google-research/heldout-influence-estimation.
>
> > [5] Unveiling privacy, memorization, and input curvature links, ICML, 2024.
>
> > [6] Challenging forgets: Unveiling the worst-case forget sets in machine unlearning, ECCV, 2024.
>
> > [7] Can bad teaching induce forgetting? unlearning in deep networks using an incompetent teacher, AAAI, 2023.

---

### Author Response · Authors · 2025-12-03
**Summary to ACs**

Dear Program Chairs, Senior Area Chairs, Area Chairs,

MU-Mis addresses a central challenge in machine unlearning by estimating the forgetting data's sample contribution to minimize collateral utility loss, thereby moving beyond existing impair–then–repair unlearning regime and enabling remaining-data-free unlearning with competitive performance. Importantly, MU-Mis outperforms existing remaining-data-free methods significantly across all settings.

We sincerely appreciate the reviewers for their time and constructive feedback. While the value of remaining-data-free unlearning is well-recognized, several concerns appear to overlook its inherent difficulty, resulting in some unrealistically strict expectations.

> **1. Experimental Performance Expectation.**

**Reviewer pcTz** expected a remaining-data-free method to surpass SoTA remaining-data-dependent methods in the most challenging random-subset removal.

**Our response.**
We included 3 reviewer-recommended SoTA remaining-data-dependent methods as further comparison, and conclusions remain unchanged:

* MU-Mis **outperforms** SoTA remaining-data-dependent methods **LoTus** and **MunBa** in **full-class** and **sub-class** unlearning.
*  MU-Mis is **less competitive** than the remaining-data-dependent **RUM** in the **random-subset** setting, which is widely considered the hardest scenario.
*  Across **all settings**, MU-Mis **consistently outperforms all the existing remaining-data-free method** by a substantial margin.

Remaining-data-free unlearning is fundamentally constrained by not accessing the remaining data and MU-Mis has made substantial progress within these limits. We're afraid that expecting a nascent data-free method to exceed SoTA data-dependent approaches in the most challenging unlearning setting might imposes requirement beyond what is currently possible.

We were in active discussion with Reviewer pcTz before the interruption of rebuttal, and we wish the reviewer would consider a more calibrated expectation considering existing development of remaining-data-free unlearning.

Beyond above concerns, **Reviewer pcTz** deemed MU-Mis as interesting and engaged deeply with our analysis linking input sensitivity, sample contribution, memorization, and sample influence. The reviewer described our connections as **“very interesting”** and **“a foundation for really cool future results.”** We deeply value this recognition, which we believe to highlight a broader conceptual impact of MU-Mis and its potential to guide  more practical unlearning.

> **2. Theoretical confusion between MU-Mis and influence function.**

**Reviewer 6Bdx** conflats theoretical derivation of MU-Mis with classical influence-function, undermining the originality and key contribution of our work.

**Our response.**
1. Influence function measure sample contribution in **parameter space**, i.e., $\Delta w$, relying on convexity assumption and Hessian-inverse, which are known to break down in DNNs.
2.  MU-Mis moves beyond **parameter space** to **function space**, i.e., $\partial A/\partial x, which derives from DNN's learning dynamics to eliminate assumption and prohibitive Hessian computation.
3. To our best knowledge, influence-function–based unlearning has not yet achieved  remaining-data-free performance while MU-Mis made it.

MU-Mis is fundamentally different from, and not reducible to, influence function, and its competitive remaining-data-free performance further underscore its conceptual novelty and empirical strength.

By the way, Reviewer 6Bdx also expressed clear dissatisfaction with our decision to place the related work section in Appendix. With the extended page limit, we have moved it into the main paper.

> **3.  Theoretical Discussion and Simplifications**

**Reviewer oXDL** did realize that our measurement of sample contribution in function space is novel and challenging. The reviewer's negative score comes from a higher expectation on a rigorous guarantee of our approximation.

**Our response.**
1. Our adopted simplifications are common and standard in learning theory of DNNs.
2. Empirical performance of MU-Mis validates that our identified ''computable'' term serves as an effective proxy.
3. Additional experiments reveal correlations of our proxy with sample's memorization and influence score, confirming the omitted residuals are not that critical.

In deep learning theory, precise attribution of sample contribution in DNNs is known to be intractable. The expectation for more rigorous theoretical guarantees overlooks the underyling limitation. Within such limits, we have made to identify and provide a principled, computable proxy that can meaningfully guide unlearning to achieve competitive remaining-data-free performance.

Apart from these core misunderstandings, we addressed all reviewer comments point-by-point and integrated the revisions into our manuscript. We appreciate your careful evaluation under this year’s exceptional reviewing conditions.

---

### Meta-Review · Area_Chair_keNw · 2026-01-04

**Summary:**

This paper introduces MU-Mis, a novel remaining-data-free machine unlearning (MU) method based on the hypothesis that a sample’s contribution to training is reflected in the model’s input sensitivity to that sample.
The paper provides theoretical results to establish this connection, under some simplifications. It also provides empirical results showing that models have higher input sensitivity to their training data. Furthermore, it demonstrates that the sensitivity gap, i.e. the difference between the gradient norm of the target-class logit with respect to the input and the gradient norm of the non-target class (irrelevant) logits, increases for training examples during the learning process.
The authors therefore design an unlearning algorithm that minimizes the sensitivity gap between target and non-target logits on the forget set, as a means of removing the contribution of these forget set examples from the model’s weights, without requiring access to the remaining data.


The main concerns that the reviewers raised are the following
- *C1*. The theoretical analysis that connects a sample’s contribution to its input sensitivity is weak, relies on simplifications (Reviewer oXDL).
- *C2*. The paper overstates the performance improvements of the method. In several cases, the results are mixed. (Reviewer oXDL). Overstates “effortless finetuning” since the optimal hyperparameter value for the stopping criterion varies substantially across tasks (Reviewer wSLf).
- *C3*. MU-Mis is not domain general, e.g not suitable for feature or attribute unlearning (Reviewer wSLf)
- *C4*. Missing justification for redirecting to a randomly selected irrelevant class c’ (Reviewer wSLf).
- *C5*. Unclear how input sensitivity-based unlearning would be affected by commonly-used training regimes that attempt to reduce sensitivity, like different regularizers (Reviewer wSLf).
- *C6*. Missing comparisons to more advanced baselines, including the newer and highly-relevant RUM algorithm that outperforms the baselines used in this work (Reviewer pcTz). Reviewer 6Bdx also suggests comparing to newer methods.
- *C7*. The MIA used for evaluation is simplistic (Reviewer pcTz).
- *C8*. The intuition of the method is not expressed well, poor writing quality (Reviewer 6Bdz)
- *C9*. The novelty is unclear and differentiation from influence functions (Reviewer 6Bdz)

**Reviewer Concerns:**

The authors were very active in the rebuttal phase and made substantial efforts to address the feedback of the reviewers.
Several concerns were addressed via clarifications (e.g about the motivation, similarity and difference from influence functions, relationship to the related but distinct notion of memorization or self-influence), several additional experiments, against additional baselines, and using different metrics, to name some.

The authors admitted that their theoretical analysis makes some simplifications, which they view as acceptable, since they are standard in learning theory. They also admitted that their phrasing was misleading and overstated the performance improvement of MU-Mis over prior work. They have rephrased to a more suitable and accurate statement that better captures the nuances of the experimental findings, including the new findings obtained during the rebuttal phase.

The ultimate weakness is that MU-Mis does not outperform all methods in all settings. Notably, out of the settings considered, it is outperformed by RUM on the random subset unlearning setting when considering standard metrics (though, according to the forget set KL-divergence metric included in the rebuttal, MU-Mis does outperform all prior work). It should be noted, though, that RUM uses the retain set whereas MU-Mis does not. When compared against retain-set-free methods, MU-Mis performs better across settings explored, and this is true of even some methods that do use the retain set. This is a significant step forward for the significantly harder retain-set-free setting that I believe warrants acceptance.

**Reviewer Scores:**

**Reviewer oXDL**.
In response to the reviewer, the authors admitted that (i) the theoretical results do require simplifications, in particular the use of a first-order Taylor expansion, which however is standard in learning theory and therefore acceptable (C1), and (ii) the previous phrasing does overstate performance improvement, and the authors have adjusted the wording to express a weaker statement (“on par with …”) (C2).
The authors also presented extensive discussion and experiments in response to the reviewer’s question about whether different samples exhibit different sensitivity signatures and how this relates to the degree of memorization.
Citing relevant works, they discuss the connection between “memorization” (which can be seen as self-influence) to “sample influence” (which also captures cross-influence). Their empirical results show that highly memorized samples have smaller sensitivity gap than low and middle memorized examples, perhaps suggesting that sensitivity gap captures something beyond just self-influence and is therefore a more useful proxy for unlearning where we want to remove all influence that a given forget set example has on itself as well as on other examples.
Had there been a rebuttal phase, it seems possible that this reviewer could have raised their score from a weak reject to a weak accept at most, thanks to clarifications, additional analyses that strengthen the paper and adjustments to the text to make statements more accurate.


**Reviewer wSLf**.
The authors admitted that MU-Mis is limited to sample- or class-level forgetting, leaving other settings for future work (C3), and that their phrasing of “effortless tuning” overstates the ease of tuning, and will be adjusted (C2).
To address C4, they ran an experiment where the class c’ is chosen deterministically (as c+1 where c is the target class) rather than randomly. They show that this yields worse results compared to the random selection. While this goes in the direction of addressing the concern, aside from choosing c’=c+1, it would make sense to pick c’ as the “most similar” other class, for example.
To address C5, the authors report new results where different regularization was applied in training, showing that MU-Mis still works reliably.
They also present several other results, e.g. on how the performance of MU-Mis is affected by the size of the forget set, showing that it’s harder to preserve model utility for larger forget sets, given that MU-Mis does not use the retain set for utility preservation.
The reviewer interacted with the authors during the rebuttal, following  up on their clarification regarding the type of MIA that they used, and expressing concerns that the chosen MIA is a weak one (similar to C7 raised by Reviewer pcTz). In response, the authors presented new results using a variation of LiRA for unlearning which strengthens their empirical investigation.
This reviewer was already positive towards the paper, recommending a weak accept, and I imagine this interaction with the reviewers would only increase their score.

**Reviewer pcTz**.
The authors made substantial effort during the rebuttal to address pcTz’s feedback, including several new experiments with RUM (against RUM itself but also exploring the combination of MU-Mis with RUM, since they argue the two can be complementary), comparison against other recent baselines of LoTus and MunBa, experiments with a different MIA and discussion of its complementarity with the originally-reported one, as well as using KL-divergence as an additional metric for output indistinguishability from the (outputs of the) gold standard retrain-from-scratch model. The authors have also extensively discussed the relationship between memorization, sample influence and sample contribution and made extensive efforts to show mathematically the difference between the sensitivity gap that they use and “loss curvature” (which is a standard proxy used for the prohibitively-expensive-to-compute memorization score of an example). They empirically examine correlations between sensitivity gaps and memorization levels for samples in CIFAR-10 (see also the above regarding the responses to Reviewer oXDL). All these discussions unfolded across many rounds of interaction with the reviewer.
Overall, MU-Mis is shown to outperform LoTus and MunBa even though these methods do rely on the retain set. MU-Mis fails to outperform RUM in the random-subset unlearning scenario (the only scenario where they compared against RUM) according to standard metrics, but they do outperform it in terms of the forget set KL-divergence metric they added during the rebuttal.
Based on this, it is possible that Reviewer pcTz would raise their score, assuming these nuances are captured well enough in the updated paper, given that Mu-Mis is a retrain-free method that appears to be competitive in different regards with retain-set-reliant approaches.

**Reviewer 6Bdx**.
The authors elaborate on the motivation of their work in a clear way in my opinion, which will aid the exposition of the work in the revision (C8). This explanation also helps to illustrate the differentiation between this work and influence functions (C9), which is based on a distinction between parameter space (in influence functions) vs function space (in this work). The authors argue that the latter provides an easier-to-compute quantity, which motivates their approach. The authors have also made changes to the organization of the paper based on the reviewer’s feedback (e.g. moving the related work section to the main body of the paper) and clarified several questions the reviewer had, supported by experiments and discussion. The addition of experimental comparisons against LoTus and MunBa also satisfies the reviewer’s suggestion, and the results show that MU-Mis outperforms these methods. Based on this, it is possible that the reviewer would have raised their score.

---

### Decision · Program_Chairs · 2026-01-26

Accept (Poster)